# WHEN LARGE MULTIMODAL MODELS CONFRONT EVOLVING KNOWLEDGE: CHALLENGES AND EXPLORATIONS

**Kailin Jiang**[1,2*], **Yuntao Du**[3,4*], **Yukai Ding**[5,2], **Yuchen Ren**[6], **Ning Jiang**[7],
**Zhi Gao**[8,2], **Zilong Zheng**[2], **Lei Liu**[1†], **Bin Li**[1], **Qing Li**[2†]

[1]University of Science and Technology of China  [2]State Key Laboratory of General Artificial Intelligence, BIGAI
[3]C-FAIR&school of software, Shandong University  [4]State Key Lab. for Novel Software Technology, Nanjing University, P.R. China
[5]Wuhan University  [6]The University of Sydney  [7]Northeast Forestry University  [8]Beijing Institute of Technology

## ABSTRACT

Large Multimodal Models (LMMs) store vast amounts of pretrained knowledge but struggle to remain aligned with real-world updates, making it difficult to avoid capability degradation when acquiring evolving knowledge. Furthermore, most current work focuses on exploring static textual knowledge injection, neglecting dynamic multimodal evolving knowledge injection, leaving the potential of LMMs for multimodal knowledge injection as an open question. To address this, we first propose a pipeline to construct MMEVOKE, a benchmark for evaluating LMMs' ability in multimodal evolving knowledge injection. MMEVOKE contains 9,422 samples spanning 159 subtypes. Then, based on extensive experiments with MMEVOKE, we reveal challenges such as poor injection performance and capability degradation in existing knowledge injection methods through knowledge injection tests and general capability tests. Finally, to tackle these challenges, we introduce knowledge augmentation and knowledge retention methods, finding that knowledge-aware augmentation strengthens knowledge injection performance, and that Data Replay and MoE methods effectively mitigate capability degradation.

Project Page: https://evoke-lmm.github.io/

## 1  INTRODUCTION

Recent research shows that Large Language Models (LLMs) and Large Multimodal Models (LMMs) gain substantial world knowledge and reasoning capabilities through large-scale pre-training (Brown et al., 2020; Zhao et al., 2023; Liu et al., 2023a). By capturing linguistic patterns and factual information, they achieve significant advancements across domains, demonstrating significant potential for research and applications (Cui et al., 2024; Su et al., 2025). However, the rapid evolution of global information poses significant challenges for LLMs and LMMs in maintaining knowledge consistency, as the constant updating of knowledge and the emergence of new events and entities hinder their ability to maintain accuracy, leading to knowledge obsolescence and inaccuracies.

As shown in Figure 1, LMM fails to recognize or answer question regarding the newly emerged entity _Xiaomi Su7_, responding with irrelevant information (e.g., **_Question: Which company produces the car in the image? Answer: Porsche_**). This indicates that the static nature of trained LMM causes rapid knowledge obsolescence, resulting in inaccuracies and illusions, thereby undermining the reliability of knowledge intensive tasks and requiring consistency with the constantly evolving knowledge.

Researchers have proposed several methods to inject knowledge into LLMs, including fine-tuning to adapt parameters to specific domains (Singhal et al., 2023; 2025; Zhang et al., 2023), retrieval-augmented generation to integrate external knowledge through retrieval and reasoning tools (Ram et al., 2023; Si et al., 2023; Yao et al., 2023), and real-time knowledge updates through internet search combined with LLMs (Nakano et al., 2021; Jiang et al., 2023). Previous works (Jang et al., 2022; Ovadia et al., 2024) constructed distinct knowledge corpora using CC-RECENTNEWS articles

---

*Equal contribution. † Corresponding author.

and U.S. current events, respectively, to explore knowledge injection in LLMs. Researchers begin focusing on LLMs' ability to handle temporal knowledge. Realtime QA (Kasai et al., 2023) and DyKnow (Mousavi et al., 2024) assess knowledge freshness in real-time content. EvoWiki (Tang et al., 2025) provides a multi-dimensional framework for evolving knowledge, categorizing it into stable, evolved, and uncharted levels. This framework enables comprehensive analysis by incorporating multi-hop reasoning and automatic updates. However, the majority of existing research is confined to the text domain and fails to explore solutions for discovered challenges. This manifests in two ways: first, a lack of real-world multimodal evolving knowledge, such as the iterative updating of entities like *Xiaomi Su7* and *Xiaomi Yu7* in Figure 1. Second, existing work often overlooks analyzing and exploring solutions for the temporal and evolving knowledge challenges identified during evaluation.

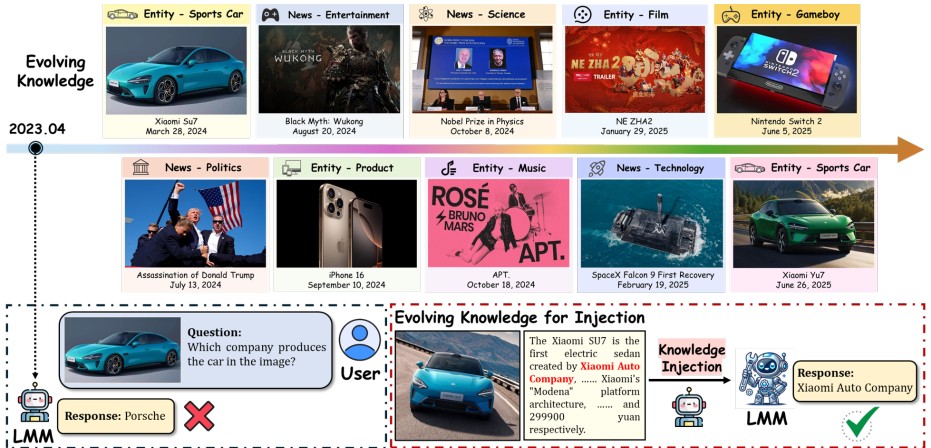

Figure 1: **Motivation and Overview of MMEVOKE.** A fundamental limitation of trained LMMs is their static nature, which causes their inherent knowledge to become outdated and inaccurate over time. Addressing this requires methods for the efficient acquisition of evolving knowledge. To facilitate research in this direction, we propose MMEVOKE to specifically evaluate the knowledge injection performance of LMMs when confronted evolving knowledge.

To address these challenges, we introduce **MMEVOKE**, a multimodal evolving knowledge benchmark designed to systematically assess knowledge injection methods in LMMs. We then conduct two types of evaluations: knowledge injection tests (to evaluate knowledge adaptation, *i.e.,* the ability to acquire new knowledge) and general capability tests (to evaluate knowledge retention, *i.e.,* the ability to preserve previous knowledge). Based on observations from tests, we identify two challenges inherent in current knowledge injection paradigms and explore corresponding attempts to address them. Our efforts are summarized as follows:

i. *We propose a comprehensive benchmark MMEVOKE for multimodal evolving knowledge, which, to our best knowledge, serves as the first evaluation dataset to measure LMMs' evolving knowledge injection capabilities.* In Figure 2, we propose a simple and reproducible data construction pipeline capable of continuously collecting evolving knowledge. MMEVOKE is divided into two primary areas: News and Entity. In Figure 1, News and Entity area encompass the latest and popular news and entities since 2024, respectively. Collectively, these two areas span 159 subfields and contain 9,422 carefully collected multimodal evolving knowledge.

ii. *Knowledge injection and general capability tests unveil challenges in existing knowledge injection paradigms.* We conduct knowledge injection tests with Supervised Fine-Tuning, Retrieval Augmented Generation, Web Search Engine, and Sufficient Context on MMEVOKE. Obtained the following observations: ❶ Existing methods exhibit poor knowledge adaptation performance; ❷ Contrary to intuition, the performance of LMMs remains imperfect even with sufficient context. Second, we conduct knowledge retention ability on LMMs after SFT across 7 capability dimensions, ❸ revealing significant capability degradation and ❹ a consistent ranking of degradation severity. ❺ Notably, severe degradation in instruction-following causes cascading failures in other capabilities.

iii. ***Knowledge augmentation and retention methods are effective explorations to mitigate knowledge injection's challenges.*** We articulate the distinction between data augmentation and knowledge augmentation, demonstrating that a knowledge-aware approach not only strengthens knowledge adaptation but also mitigates capability degradation. For mitigating capability degradation, we find direct knowledge rehearsal (Replay) and structured knowledge separation (MoELoRA) to be effective, in contrast to indirect knowledge constraint methods (EWC, LwF), which yield unstable results and even worsen degradation.

## 2 RELATED WORK

**Large Multimodal Models.** Demonstrating strong vision-language understanding through extensive knowledge and cross-modal alignment, these models typically combine a vision encoder with a pretrained large language model via an alignment module. BLIP-2 (Li et al., 2023a) employs a lightweight Transformer (Q-Former), while MiniGPT-4 (Zhu et al., 2024) and InstructBLIP (Dai et al., 2023) enhance performance via multimodal instruction tuning. LLaVA (Liu et al., 2024a) uses an MLP alignment layer and self-instruct (Wang et al., 2023b) data generation, and Qwen-VL (Bai et al., 2023) introduces a visual receptor with three-stage training.

**Knowledge Injection.** Improving factual accuracy remains a critical research focus, often addressed by incorporating external knowledge into models. Central approaches include encoding new information (Chen et al., 2022) and utilizing retrieval or augmentation from knowledge sources (Fan et al., 2020). These sources extend beyond vector databases in modern RAG systems (Lewis et al., 2020a) to structured resources like knowledge graphs (Martino et al., 2023). Alternatively, model adapters (Lauscher et al., 2020) enable domain adaptation through training only adapter parameters.

**Continual Learning.** The injection of evolving knowledge is fundamentally a continual learning (CL) problem, specifically one centered on the acquisition of new factual knowledge (Huo & Tang, 2025; Liu et al., 2025c). A central challenge in CL is mitigating catastrophic forgetting, which is the tendency of models to lose prior knowledge and capabilities while learning new information. Existing CL methods designed to address this can be broadly categorized. Regularization-based techniques focus on maintaining the stability of critical parameters (Kirkpatrick et al., 2017; Li & Hoiem, 2017; Liu et al., 2024b; Qiao et al., 2024; Wang et al., 2023a; Liu et al., 2025a;b). Architecture-centric approaches introduce parameter isolation (Mallya & Lazebnik, 2018; Serra et al., 2018; Cao et al., 2024), adaptive structures (Yoon et al., 2018; Jiang et al., 2025b), or modular designs (Shen et al., 2019). Rehearsal-based methods leverage memory buffers for experience replay (Bonicelli et al., 2022). Finally, prompt-based solutions employ learnable prompts to enhance efficiency without explicit data storage (Wang et al., 2022b; Smith et al., 2023). Unlike previous work which often focuses on task-level retention in computer vision, this work investigates the specific challenges of knowledge retention and adaptation when LMMs are confronted with evolving factual knowledge.

## 3 MULTI-MODAL EVOLVING KNOWLEDGE BENCHMARK

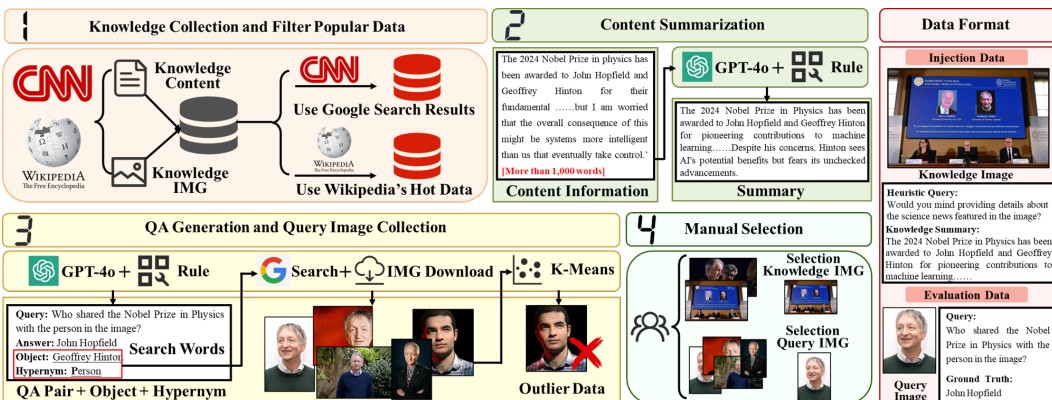

Figure 2: **Overview of construction pipeline for MMEVOKE.** For heuristic query, we manually write multiple templates and randomly select one template for each data.

To evaluate evolving knowledge injection in LMMs, we propose an automated pipeline to construct **MMEVOKE** (**M**ulti-**M**odal **EVO**lving **K**nowledg**E**), a benchmark focused on evolving knowledge. Figure 2 illustrates MMEVOKE benchmark's data format via a Nobel Prize in Physics case. Each evolving knowledge consists of two components: **injection data** $\mathcal{D}_{\mathcal{K}} = (i_k, x_k, y_k)_{k=1}^{N}$ comprises $N$ triples of knowledge-associated image $i_k$, heuristic query $x_k$, and knowledge summary $y_k$; and **evaluation data** $\mathcal{D}_{\mathcal{Q}} = (i_q, x_q, y_q)_{q=1}^{N}$ contains query image $i_q$, query $x_q$, and ground truth $y_q$.

## 3.1 BENCHMARK CONSTRUCTION

The simple and reproducible pipeline process for evolving knowledge collection is shown in Figure 2, and continuously provides evolving knowledge for the field of knowledge injection.

- **Step 1: Knowledge Collection.** To ensure data timeliness and authenticity, we collect News and Entity evolving knowledge (starting from 2024) from authoritative CNN and Wikipedia sources. For News evolving knowledge: we extract URLs from CNN's robots.txt via Gentleman's Agreement, collecting `Type`, `Title`, `Content` and `Image`. For Entity evolving knowledge: We compare offline versions of Wikipedia at different time points to identify new entries, collecting `Type`, `Entity Name`, `Description`, `Image` and `Pageviews`. To select representative data, we filter for popularity. News evolving knowledge cannot be filtered directly due to lacking popularity metrics; we search Google using the `Title` and select popular data based on high-similarity result count. Entity evolving knowledge uses `Pageviews` directly to obtain popular data.
- **Step 2: Content Summarization.** Whether the `Content` and `Description` of evolving knowledge, the textual content is often lengthy, posing challenges for LMMs to utilize effectively. Consequently, we employ GPT-4o to summarize the content by establishing stringent rules and providing rich contextual examples, thereby obtaining `Summary` for each data.
- **Step 3: QA Generation and Query Image Collection.** We establish stringent rules for GPT-4o to extract a corresponding VQA pair for each News / Entity evolving knowledge `Summary`. Simultaneously, GPT-4o must identify the core `Object` described in the VQA pair and its `Hypernym`. For VQA pairs generated in the preceding step, the `Query Image` is absent. Consequently, we combine `Object` with `Hypernym` as search key words (e.g., Geoffrey Hinton Person), search images via Google, and download images. Given potential inclusion of anomalous images in Google-sourced data, we follow prior work (Li et al., 2024b) by utilizing CLIP to extract image features for clustering, thereby detecting and removing aberrant image data.
- **Step 4: Manual Selection.** Since the data collected from CNN websites and Wikipedia often contain multiple associated images, we meticulously conduct a manual review of the images corresponding to each data to ensure the acquisition of high-quality image data.

Table 1: **Key Statistics of MMEVOKE.**

| Statistic | Number |
|---|---|
| Total evolving knowledge | 9,422 |
| - Entity evolving knowledge | 4,494 (47.7%) |
| - News evolving knowledge | 4,928 (52.3%) |
| Total Areas/Subfields | 2/159 |
| Number of unique images | 18,834 |
| - Injection images | 9,422 |
| - Evaluation images | 9,412 |
| Evolving knowledge length | |
| - Maximum length | 386 |
| - Average length | 95.7 |
| Evaluation question length | |
| - Maximum length | 38 |
| - Average length | 23.2 |
| Evaluation answer length | |
| - Maximum length | 12 |
| - Average length | 2.3 |

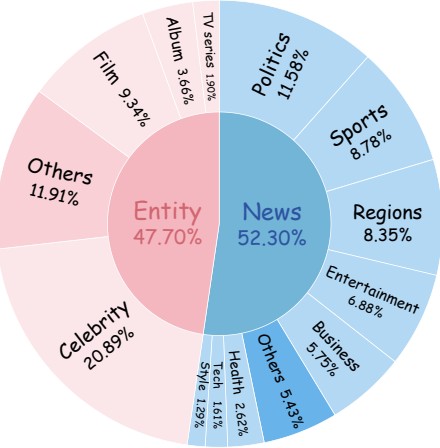

Figure 3: **Area and Subfield Distribution of MMEVOKE.** "Others" for Entity and News includes undisplayed subfields.

**Benchmark's Quality.** For MMEVOKE, data quality is paramount. We source popular data from authoritative providers (CNN, Wikipedia) to ensure high-quality original data. For data generation,

strict rules and contextual examples are implemented using GPT-4o. Query images are collected from Google and refined using K-Means clustering to remove noise. This benchmark construction pipeline ensures MMEVOKE's high quality, further validated by human study in Appendix B.4. More details regarding MMEVOKE are presented in Appendix B and G.

## 3.2 BENCHMARK ANALYSIS

- **Key Statistics, Area and Subfield Distribution.** MMEVOKE comprises 9422 evolving knowledge instances, covering 159 fine-grained subfields as shown in Figure 3 (29 News and 130 Entity subfields), highlighting its diversity, with key statistics presented in Table 1.
- **Self-Evolving Properties.** Evolving knowledge will continue to emerge, so self-evolving properties are crucial for MMEVOKE. Our data construction pipeline minimizes manual involvement, only manual selection is not automated. Thus, we develop front-end webpages for each data, accelerating manual selection and reducing average inspection time to 10 seconds. To sum up, we easily reproduce data construction pipeline and update MMEVOKE quarterly.

## 4 EXPERIMENT

Under the knowledge injection paradigm, assume a large multi-modal model $\mathcal{M}$ can be optimized through access to the injection data $\mathcal{D}_{\mathcal{K}}$. The optimization seeks a mapping function $f \in \mathcal{F}$ to derive the enhanced model $\mathcal{M}^* = f(\mathcal{M}, \mathcal{D}_{\mathcal{K}})$ must satisfy dual objectives:

(1) **Knowledge Adaptation:** Maximize accuracy on evaluation data $\mathcal{D}_{\mathcal{Q}}$ to demonstrate robust generalization capabilities on new knowledge:

$$\max_f \mathbb{E}_{(i_q, x_q, y_q) \sim \mathcal{D}_Q} \left[ \mathbb{I}\left(\mathcal{M}^*(i_q, x_q) = y_q\right) - \mathbb{I}\left(\mathcal{M}(i_q, x_q) = y_q\right) \right]. \tag{1}$$

(2) **Knowledge Retention:** Minimize the performance gap between $\mathcal{M}^*$ and $\mathcal{M}$ on general capability tests $\mathcal{D}_P$ to maintain the model's previous knowledge and capabilities:

$$\min_f \mathbb{E}_{(i_p, x_p, y_p) \sim \mathcal{D}_P} \left[ \mathbb{I}\left(\mathcal{M}(i_p, x_p) = y_p\right) - \mathbb{I}\left(\mathcal{M}^*(i_p, x_p) = y_p\right) \right]. \tag{2}$$

In this section, we conduct experiments to explore the following research questions:

- **RQ1:** How do existing knowledge injection methods perform in evolving knowledge injection task? Is the injection performance of knowledge from different fine-grained subfields consistent?
- **RQ2:** How do knowledge-injected LMMs perform on previous general capability tests? Do the post-injected LMMs successfully retain their inherent capabilities?

## 4.1 EXPERIMENTAL SETUP

**Large Multimodal Models.** To ensure that the evolving knowledge in MMEVOKE is as unknown as possible to LMMs, we select two representative models for our experiments: LLaVA-v1.5 (Liu et al., 2024a) and Qwen-VL-Chat (Bai et al., 2023) (all released in 2023). To verify this, we conduct zero-shot testing on MMEVOKE (**Vanilla** in Table 2), where the extremely low performance indicates that the vast majority of the evolving knowledge is indeed unknown to the LMMs. The knowledge injection methods will be detailed below.

- **Supervised Fine-Tuning** necessitates datasets comprising labeled input-output pairs. Among the commonly used SFT approaches, instruction tuning (Wang et al., 2022a; Mishra et al., 2022; Ouyang et al., 2022; Taori et al., 2023) has been identified as a highly effective technique to improve model performance. We employ two training strategies: Full-FT and LoRA (Hu et al., 2022).
- **Retrieval Augmented Generation** enhances LMMs in knowledge-intensive tasks by integrating external knowledge sources (Lewis et al., 2020b). While early implementations required task-specific training, recent studies (Neelakantan et al., 2022) demonstrate that pre-trained embedding models can achieve significant performance gains without additional training. Here, we focus on the knowledge injection performance of Multimodal Retrieval Augmented Generation (MM-RAG)

on LMMs. Three retrieval strategies are employed: Text-Only (retrieving candidate documents only based on textual features), Image-Only (retrieving candidate documents only based on visual features), UniIR (Wei et al., 2024) (retrieving candidate documents based on multimodal features).
- **Commercial AI Web Search Engine** integrates internet search and retrieves the evolving knowledge in the reasoning process. To validate the capabilities of commercial AI web search engines, we employ Gemini (Team et al., 2023), Perplexity AI, and GPT-4.1.
- **Sufficient Context** can be regarded as a special case of RAG, where it contains all the necessary information required to answer the question and directly serves as context for LMMs' inference. Meanwhile, MMEVOKE's high-quality (Section 3.1) ensures reliable Sufficient Context results.

**Evaluation Protocol:** Cover Exact Match (CEM) and F1-Score (F1) are used as evaluation metrics in open-domain question answering tasks. The former requires to match model prediction with ground truth (Xu et al., 2023) and equation is $CEM = \begin{cases} 1, & y_q \subseteq \hat{Y} \\ 0, & \text{otherwise} \end{cases}$, Where $\hat{Y}$ is model prediction. The latter evaluates overlap between the model's prediction and ground truth at the word level, balancing Precision and Recall (Chan et al., 2024). Let $\mathcal{W}(y_q) = \{y_1, \ldots, y_m\}$ be ground truth and $\mathcal{W}(\hat{Y}) = \{\hat{y}_1, \ldots, \hat{y}_n\}$ be model's prediction. The overlap is $\mathcal{U}(\hat{Y}, y_q) = \sum_{t \in \mathcal{W}(y_q)} \mathbf{1}[t \in \mathcal{W}(\hat{Y})]$, where $\mathbf{1}[\cdot]$ is the indicator function. Precision is $\mathcal{P}(\hat{Y}, Y) = \frac{\mathcal{U}(\hat{Y}, y_q)}{|\mathcal{W}(\hat{Y})|}$ and Recall is $\mathcal{R}(\hat{Y}, Y) = \frac{\mathcal{U}(\hat{Y}, y_q)}{|\mathcal{W}(Y)|}$.

## 4.2 PERFORMANCE OF KNOWLEDGE ADAPTATION (RQ1)

Table 2: **Performance of knowledge injection methods on MMEVOKE.**

| Method | ALL | | News.Avg | | Entity.Avg | |
|---|---|---|---|---|---|---|
| | CEM ↑ | F1 ↑ | CEM ↑ | F1 ↑ | CEM ↑ | F1 ↑ |
| *LLaVA-v1.5* | | | | | | |
| Vanilla | 4.89 | 9.34 | 7.37 | 11.96 | 2.18 | 6.47 |
| Full-FT | 18.02 | 15.17 | 21.35 | 16.34 | 14.37 | 13.88 |
| LoRA | 15.23 | 18.31 | 17.72 | 19.42 | 12.51 | 17.09 |
| MM-RAG[Text-Only] | 24.05 | 34.32 | 37.32 | 49.39 | 9.50 | 17.80 |
| MM-RAG[Image-Only] | 25.25 | 37.11 | 19.28 | 26.76 | 31.80 | 48.45 |
| MM-RAG[UniIR] | **40.68** | **57.51** | 40.12 | 53.21 | 41.30 | 62.23 |
| *Qwen-VL-Chat* | | | | | | |
| Vanilla | 5.84 | 10.99 | 7.75 | 12.72 | 3.74 | 9.10 |
| Full-FT | 10.16 | 16.61 | 13.35 | 18.22 | 6.65 | 14.83 |
| LoRA | 6.95 | 12.64 | 9.27 | 14.55 | 4.41 | 10.54 |
| MM-RAG[Text-Only] | 21.79 | 31.28 | 31.51 | 41.14 | 11.13 | 20.47 |
| MM-RAG[Image-Only] | 22.31 | 33.09 | 17.82 | 25.15 | 27.24 | 41.79 |
| MM-RAG[UniIR] | **32.75** | **46.18** | 33.26 | 43.36 | 32.20 | 49.28 |
| *Commercial AI Web Search Engines* | | | | | | |
| Gemini-2.0-Flash | 18.21 | 26.52 | 21.23 | 27.75 | 14.91 | 25.16 |
| Gemini-2.5-Pro | 44.19 | 52.58 | **48.86** | 52.84 | 39.27 | 46.27 |
| Perplexity AI | **48.27** | **62.44** | 47.58 | 56.51 | 48.96 | 68.78 |
| GPT-4.1 | 39.61 | 42.69 | 41.81 | 43.08 | 37.19 | 42.26 |
| *Sufficient Context* | | | | | | |
| LLaVA-v1.5 | 56.13 | 75.77 | 56.78 | 72.37 | 55.43 | 79.50 |
| Qwen-VL-Chat | 48.96 | 66.02 | 49.98 | 63.42 | 47.84 | 68.87 |
| Gemini-2.5-Pro | 72.15 | 80.46 | 72.61 | 78.77 | **71.65** | **82.32** |
| GPT-4.1 | **75.02** | **83.74** | 79.22 | 88.20 | 71.21 | 79.68 |

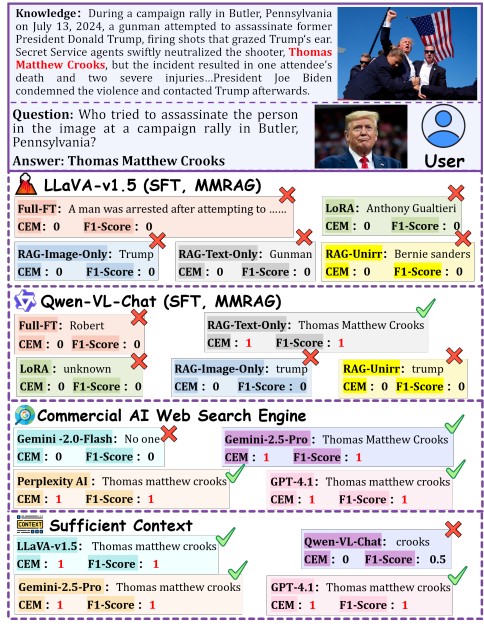

Figure 4: **Case Study.**

We conduct extensive experiments to evaluate knowledge injection methods on MMEVOKE. Additional results are in Appendix C. Key observations from Table 2 include:

- **Obs 1: Current methods perform poorly on MMEVOKE.** ❶ Specifically, the performance of parameter-modifying methods (Full-FT, LoRA) is extremely low. For example, on Qwen-VL-Chat, LoRA achieves only 6.95% CEM, showing minimal performance gain compared to Vanilla. ❷ MM-RAG outperforms SFT, yet its highest performance is only 40.68% CEM and 57.51% F1. Among these, MM-RAG[Text-Only] and MM-RAG[Image-Only] variants perform comparably, while the version using MM-RAG[UniIR] yields the best results. ❸ Commercial AI Search Engines can also perform multimodal knowledge injection, but their performance varies significantly. Gemini-2.0-Flash achieves only 18.21% CEM and 26.52% F1, far below the results of Gemini-2.5-Pro, Perplexity AI, and GPT4.1. Although Perplexity AI and Gemini-2.5-Pro achieve 48.27% and 44.19% CEM respectively, this performance still falls short of ideal application standards. Current methods

exhibit shortcomings such as overfitting, instruction refusal, and irrelevant responses in Figure 4. These issues underscore the need for improved knowledge injection methods.

- **Obs 2: Contrary to intuition, LMMs still provide incorrect answers even when the context is sufficient.** We often assume that accurate and sufficient information can guarantee correct answers. However, as shown in Sufficient Context experiments in Table 2, models still give wrong answers under sufficient context conditions; for instance, Qwen-VL-Chat achieves only 48.96% CEM, and even powerful commercial models like GPT-4.1 and Gemini-2.5-pro achieve only 75.02% and 72.15% CEM, respectively, failing to achieve the expected perfect accuracy. This aligns with phenomenon observed by (Joren et al., 2025; Tang et al., 2025). This indicates that providing context is not enough; LMM's reasoning and utilization skills for evolving knowledge are essential.

> **Challenges**
>
> **Challenge 1:** Current knowledge injection methods perform poorly on MMEVOKE, failing to achieve optimal performance even with sufficient context.

## 4.3 PERFORMANCE OF KNOWLEDGE RETENTION (RQ2)

Due to the fact that only fine-tuning methods will modify model parameters in the knowledge injection methods we evaluated, general capability testing focuses on Full-FT and LoRA. To further test general capability of LMMs after knowledge injection, we conduct general capability test on 12 benchmarks across 7 different dimensions (Fu et al., 2024). Specifically, the evaluation tasks include: ❶ **Comprehensive Evaluation:** MME (Fu et al., 2023) and MMBench (Liu et al., 2024c); ❷ **OCR:** SEEDBench2_Plus (Li et al., 2024a) and OCRBench (Liu et al., 2023b); ❸ **Multidisciplinary:** ScienceQA (Lu et al., 2022) and MMMU (Yue et al., 2024); ❹ **Instruction Following:** MIA-Bench (Qian et al., 2024); ❺ **Multi-Round QA:** MMDU (Liu et al., 2025d); ❻ **Mathematical Reasoning:** MathVista (Lu et al., 2024) and MathVision (Wang et al., 2025a); ❼ **Hallucination:** POPE (Li et al., 2023b) and HallusionBench (Guan et al., 2024). More details in Appendix D. Table 3 illustrates quantitative results of Vanilla, Full-FT, and LoRA and we have following observations:

Table 3: **The results of LLaVA-v1.5 on general capability tests.** *First line* shows results of current methods in general capability tests; *Second line* shows the percentage of capability degradation or improvement of current methods compared to Vanilla. Red value indicates capability degradation, the darker the color, the more severe the degradation, and Green value indicates capability improvement. *Ranking* includes average degree of degradation and degradation ranking.

| Method | Comprehensive | | OCR | | Multidisciplinary | | Instruction | Multi-Round | Mathematical | | Hallucination | | Ranking |
|---|---|---|---|---|---|---|---|---|---|---|---|---|---|
| | MME ↑ | MMBench ↑ | SEED$^{B2P}$ ↑ | OCRBench ↑ | ScienceQA ↑ | MMMU ↑ | MIA-Bench ↑ | MMDU ↑ | MathVista ↑ | MathVision ↑ | POPE ↑ | HallusionBench ↑ | |
| Vanilla | 1,865.56 | 64.60 | 38.78 | 30.80 | 69.83 | 28.60 | 66.33 | 26.37 | 25.50 | 13.16 | 86.87 | 21.76 | - |
| Full-FT | 956.80 ↓48.71% | 52.92 ↓18.08% | 31.44 ↓18.93% | 28.10 ↓8.77% | 67.13 ↓3.87% | 24.20 ↓15.38% | 25.25 ↓61.93% | 13.03 ↓50.59% | 24.70 ↓3.14% | 11.94 ↓9.27% | 74.22 ↓14.56% | 9.27 ↓57.40% | 8 ↓25.89% |
| LoRA | 1,233.54 ↓33.88% | 53.87 ↓16.61% | 30.22 ↓22.07% | 25.70 ↓16.56% | 66.18 ↓5.23% | 21.40 ↓25.17% | 29.66 ↓55.28% | 13.70 ↓48.05% | 23.20 ↓9.02% | 12.83 ↓2.51% | 73.97 ↓14.85% | 8.78 ↓59.65% | 7 ↓25.74% |
| **Knowledge Augmentation for Text** | | | | | | | | | | | | | |
| Knowledge Agnostic | 1467.00 ↓21.36% | 56.96 ↓11.83% | 32.54 ↓16.09% | 26.20 ↓14.94% | 68.88 ↓1.36% | 22.20 ↓22.38% | 22.90 ↓65.48% | 10.61 ↓59.76% | 22.30 ↓12.55% | 8.19 ↓37.77% | 81.40 ↓6.30% | 8.24 ↓62.13% | 10 ↓27.66% |
| Knowledge Aware (+3) | 1488.83 ↓20.19% | 58.76 ↓9.04% | 39.66 ↑2.27% | 26.80 ↓12.99% | 68.95 ↓1.26% | 26.80 ↓6.29% | 23.64 ↓64.36% | 10.54 ↓60.03% | 23.60 ↓7.45% | 9.54 ↓27.51% | 73.18 ↓15.76% | 16.10 ↓26.03% | 4 ↓20.72% |
| **Knowledge Augmentation for Images** | | | | | | | | | | | | | |
| Knowledge Agnostic | 1436.52 ↓23.00% | 57.56 ↓10.90% | 29.64 ↓23.57% | 26.00 ↓15.58% | 66.47 ↓4.81% | 20.00 ↓30.07% | 21.62 ↓67.41% | 10.69 ↓59.46% | 20.60 ↓19.22% | 9.74 ↓25.99% | 81.52 ↓6.16% | 6.53 ↓69.99% | 11 ↓29.68% |
| Knowledge Aware (+3) | 1248.54 ↓33.07% | 54.21 ↓16.08% | 36.19 ↓6.68% | 25.00 ↓18.83% | 66.92 ↓4.17% | 27.00 ↓5.59% | 18.01 ↓72.85% | 10.62 ↓59.73% | 20.50 ↓19.61% | 8.13 ↓38.22% | 77.17 ↓11.17% | 13.72 ↓36.95% | 9 ↓26.91% |
| **Knowledge Retention Methods** | | | | | | | | | | | | | |
| Replay$^{Full-FT}_{+10\%}$ | 1,608.00 ↓13.81% | 60.57 ↓6.24% | 38.69 ↓0.23% | 28.60 ↓7.14% | 68.74 ↓1.56% | 29.10 ↑1.75% | 51.20 ↓22.81% | 18.09 ↓31.40% | 24.40 ↓4.31% | 13.45 ↑2.20% | 86.52 ↓0.40% | 16.15 ↓25.78% | 3 ↓9.14% |
| Replay$^{LoRA}_{+10\%}$ | 1,650.75 ↓11.51% | 60.48 ↓6.38% | 38.34 ↓1.13% | 28.60 ↓7.14% | 68.77 ↓1.52% | 28.50 ↓0.35% | 62.33 ↓6.03% | 19.31 ↓26.77% | 25.20 ↓1.18% | 13.13 ↓0.23% | 85.44 ↓1.65% | 17.90 ↓17.74% | 1 ↓6.80% |
| EWC | 1,360.09 ↓27.09% | 50.26 ↓22.20% | 33.60 ↓13.36% | 25.70 ↓16.56% | 65.71 ↓5.90% | 25.20 ↓11.89% | 29.79 ↓55.09% | 13.36 ↓49.34% | 23.30 ↓8.63% | 12.76 ↓3.04% | 76.22 ↓12.26% | 10.77 ↓50.51% | 5 ↓22.99% |
| LwF | 1,424.41 ↓23.65% | 55.41 ↓14.23% | 32.02 ↓17.43% | 25.60 ↓16.88% | 66.21 ↓5.18% | 20.60 ↓27.97% | 36.19 ↓45.44% | 13.68 ↓48.12% | 24.40 ↓4.31% | 12.04 ↓8.51% | 79.23 ↓8.79% | 9.13 ↓58.04% | 6 ↓23.21% |
| MoELoRA | 1732.47 ↓7.13% | 63.32 ↓1.98% | 38.03 ↓1.93% | 20.10 ↓34.74% | 69.70 ↓0.19% | 28.10 ↓1.75% | 64.97 ↓2.05% | 18.66 ↓29.24% | 25.80 ↑1.18% | 12.70 ↓3.50% | 83.93 ↓3.38% | 18.50 ↓14.98% | 2 ↓8.31% |

- **Obs 3: All general capacities of LMMs after injection will degrade, and degree of degradation of capacities varies in different dimensions.** Specifically, the average performance of the model significantly decreased on MME ↓41.30% , MIA-Bench ↓58.61% , MMDU ↓49.32% , and HallusionBench ↓58.53% after undergoing Full-FT and LoRA, while the degree of decline is relatively small on ScienceQA ↓4.55% , Mathvista ↓6.08% , and Mathvision ↓5.89% .
- **Obs 4: The degradation degree of different capacities of LMMs after injection shows consistent rankings in Full-FT and LoRA.** Specifically, the ranking from severe to mild according to

degree of capacities degradation (calculating the mean under the same test) is as follows: Instruction Following → Multi-Round QA → Hallucination → Comprehensive Evaluation → OCR → Multidisciplinary → Mathematical Reasoning.

**Prompt:** Is a c++ code shown in the picture? Answer the question using a single word or phrase.

**Ground Truth:** Yes.

**Vanilla's prediction:** Yes.

**Full-FT's prediction:** The 'Hello, World!' program in C++, written by Bjarne Stroustrup in 1984, has been compiled and run on a 1950s UNIVAC I computer......

Figure 5: **Violating instruction on MME.**

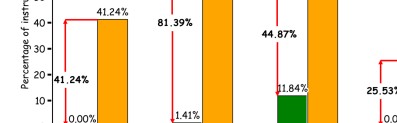

Figure 6: **Comparison of violation instructions between Vanilla and Full-FT.**

- **Obs 5: Capacities degradation in different dimensions is interrelated.** Specifically, the degradation of instruction-following capability negatively impacts other capacities. Benchmarks such as MME, SEEDBench2_Plus, etc, which require Yes/No or multiple-choice formats, necessitate robust instruction-following capability. However, according to Figures 5 and 6, we find that deterioration in instruction-following capability exerts cascading negative effects on these capabilities.

**Challenges**

**Challenge 2:** Parameter modification methods cause capability degradation in injected LMMs, exhibiting a consistent severity ranking and cascading effect.

# 5 EXPLORATIONS OF EVOLVING KNOWLEDGE INJECTION

## 5.1 KNOWLEDGE AUGMENTATION STRENGTHENS KNOWLEDGE ADAPTATION

Section 4.2 shows that existing methods struggle with knowledge injection. ***Data augmentation***, though common for limited-data scenarios, fails to improve semantic knowledge learning. ***Knowledge augmentation***, however, substantially enhances model comprehension and adaptation.

The core distinction between data augmentation and knowledge augmentation lies in their augmentation goals: the former operates solely on surface-level features (*e.g.,* pixel transformations in images or replacement of synonyms in text), whereas the latter explicitly augments knowledge-related semantic information. In Figure 7, we use Figure 1's example of Xiaomi SU7 to illustrate both knowledge-agnostic and knowledge-aware augmentation.

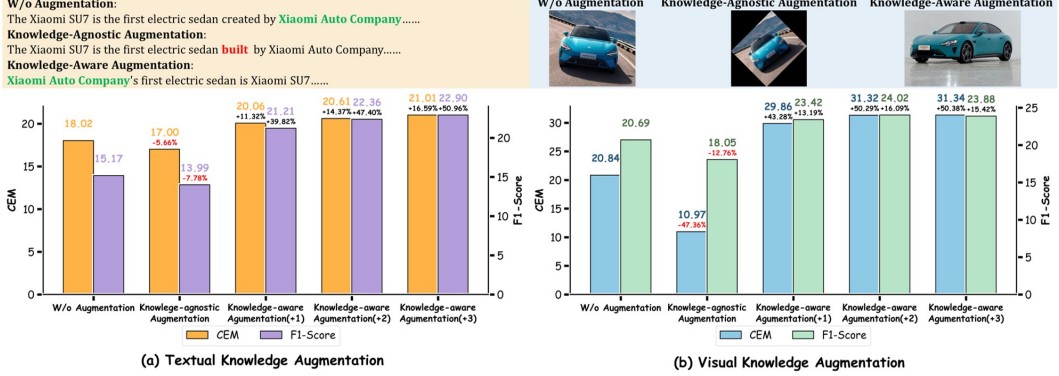

Figure 7: **Examples and performance of knowledge-agnostic and knowledge-aware augmentation.** (a) Performance of textual knowledge augmentation across the entire MMEVOKE. (b) Performance of visual knowledge augmentation solely on Entity subset.

- **Knowledge-agnostic augmentation is a rule-based mechanical augmentation.** "created" in description of is replaced by "built" and rotation operation of "Xiaomi SU7" image does not require understanding of "Xiaomi SU7" related knowledge, only mechanical augmentation.
- **Knowledge-aware augmentation is a knowledge-driven semantic augmentation.** For text, it creatively restated knowledge based on profound understanding of description. Additionally, introducing real-world images greatly enrich model's perception of the concept of "Xiaomi SU7".

Additional results are in Appendix E. Based on Figure 7 and Table 3, we have following observations:

- **Obs 6: Knowledge-agnostic augmentation leads to negative effects.** In Figure 7, text/images knowledge-agnostic augmentation results in a decrease of 5.66% and 47.36% in CEM, along with a reduction of 7.78% and 12.76% in F1-Score, respectively. This demonstrates that knowledge-agnostic augmentation fails to strength knowledge adaptation.
- **Obs 7: Knowledge-aware augmentation strengths knowledge adaptation.** In Figure 7, using just a single data instance for textual/visual knowledge-aware augmentation yields improvements of 11.32% and 43.28% in CEM, along with 39.82% and 13.19% in F1-Score, respectively. Moreover, performance further improves with increasing data quantity. This demonstrates that knowledge-aware augmentation is crucial for strengthening knowledge adaptation.
- **Obs 8: Surprisingly, knowledge augmentation can partially mitigate capability degradation.** In Table 3, knowledge augmentation outperforms both Full-FT and LoRA on benchmarks such as MMBench, SEEDBench2_Plus, and ScienceQA. Furthermore, text knowledge-aware augmentation surpasses not only Full-FT and LoRA but also conventional knowledge retention techniques (EWC & LwF). This novel discovery points to a promising new research direction.

> **Insights**
>
> **Insight 1:** Knowledge-agnostic augmentation proves detrimental and fails to add semantic knowledge. Conversely, knowledge-aware augmentation confirms that knowledge-centric strategies strength knowledge adaptation and concurrently mitigate capability degradation.

## 5.2 KNOWLEDGE RETENTION MITIGATES CAPABILITY DEGRADATION

To efficiently mitigate capability degradation after knowledge injection, we introduce knowledge retention methods: data replay (*e.g.,* Replay), mixture of experts (*e.g.,* MoELoRA (Luo et al., 2024)), parameter regularization (*e.g.,* EWC (Kirkpatrick et al., 2017) and LwF (Li & Hoiem, 2017)). Specifically, we categorize Replay into $\text{Replay}_{+10\%}^{\text{Full-FT}}$ and $\text{Replay}_{+10\%}^{\text{LoRA}}$: randomly sampled fixed-quantity data (10% of MMEVOKE's data size) from LLaVA-v1.5's pre-training data and MMEVOKE's injection data $\mathcal{D}_{\mathcal{K}}$ are mixed and used for fine-tuning employing Full-FT and LoRA strategies. Additional results are in Appendix F. Table 3 shows results and we have following observations:

- **Obs 9: Replay reactivates old knowledge networks by forcing model to "review the old".** Specifically, $\text{Replay}_{+10\%}^{\text{Full-FT}}$ (ranked 3rd) and $\text{Replay}_{+10\%}^{\text{LoRA}}$ (ranked 1st) mitigate model capability degradation across all tests. Notably, $\text{Replay}_{+10\%}^{\text{Full-FT}}$ surpasses Vanilla by 1.75% and 2.20% on MMMU and MathVision, respectively.
- **Obs 10: MoELoRA carves out dedicated zones for new knowledge to prevent parameter conflicts.** Specifically, MoELoRA (ranked 2nd) exhibits minimal degradation of only 2.05% in instruction following and surpasses Vanilla by 1.18% on MathVista.
- **Obs 11: EWC & LwF attempt to freeze prior knowledge areas through indirect and rigid constraints.** Specifically, EWC (ranked 5th) and LwF (ranked 6th) provide almost no mitigation of degradation on MIA-Bench, MMDU, and HallusionBench. Moreover, both EWC on OCRBench, ScienceQA, and MathVista and LwF on MMMU, MMDU, MathVision, and HallusionBench underperform standard Full-FT and LoRA, further exacerbating capability degradation.

> **Insights**
>
> **Insight 2:** Direct Rehearsal (Replay) and Structured Separation (MoELoRA) effectively preserve old knowledge by retraining on old data and isolating new knowledge, respectively. Indirect Constraint (EWC, LwF) fails due to rigid parameter constraints impairing retention.

## 6 CONCLUSION AND DISCUSSION

In this paper, we systematically investigate multimodal evolving knowledge injection on LMMs and propose a diverse benchmark, MMEVOKE. This work reveals two critical challenges, and corresponding explorations are conducted. Current research (Allen-Zhu & Li, 2024; Omar et al., 2023; Singhal et al., 2023) indicates that mere *"data memorization"* and genuine *"knowledge internalization"* are distinct concepts. The former only enables models to accurately fit training data, while the latter empowers models to effectively extract and manipulate factual knowledge. Similarly, in our work, ***knowledge-agnostic*** and ***knowledge-aware*** augmentation exhibit this distinction, with only knowledge-aware augmentation significantly enhancing a model's ability to internalize knowledge.

Although less effective than knowledge retention methods like Replay, MoELoRA, knowledge-aware augmentation can partially mitigate capability degradation. Therefore, exploring the synergy between these two classes of methods is a promising research direction. Potential strategies include multi-stage training to decouple knowledge adaptation from retention, or hybrid framework that integrates knowledge-aware augmentation into loss function. We posit that the synergy between the "proactive learning" of knowledge augmentation and the "capability preserving" nature of retention methods can more effectively tackle the challenges of continuous injection of evolving knowledge.

## ACKNOWLEDGEMENT

We would like to express gratitude to the anonymous reviewers for their kind comments. This research is supported by Anhui Provincial Natural Science Foundation (Grant No.2408085QF214), the National Key R&D Program of China (Grant No.2024YFB3213400), the Fundamental Research Funds for the Central Universities (Grant No.WK2080000206), the Opening Project of the State Key Laboratory of General Artificial Intelligence (Project No.SKLAGI2025OP06, No.SKLAGI2024OP10, No.SKLAGI2024OP11) and project ZR2025QC1570 supported by Shandong Provincial Natural Science Foundation.

## ETHICS STATEMENT

The primary motivation for MMEVOKE is to investigate the effectiveness of existing knowledge injection methods in learning new knowledge. Due to the scarcity of evolving knowledge in authentic multimodal contexts, we design a pipeline to collect such data from the internet. However, information online is complex and may contain false, harmful, or biased content. We therefore urge researchers to collect evolving knowledge responsibly and cautiously, ensuring that the information injected into models is accurate and safe, and that the technology is applied ethically.

## REPRODUCIBILITY STATEMENT

To guarantee reproducibility, our source code and datasets are available at `https://evoke-lmm.github.io/` to facilitate verification and reproduction of our work by the community.

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

# APPENDIX CONTENTS

## A    THE USE OF LARGE LANGUAGE MODELS IN MMEVOKE

In this section, we elaborate on the precise role of large language models within MMEVOKE, as detailed below.

- **Usage 1: MMEVOKE's construction.** In Section 3.1, we specify that GPT-4o is employed for content summarization and QA generation, which aligns with current research practices.
- **Usage 2: MMEVOKE's evaluation.** In Section 4.2, we evaluate MMEVOKE using Gemini-2.0-Flash, Gemini-2.5-Pro, Perplexity AI, and GPT-4.1, following standard benchmarking practices.
- **Usage 3: General capability tests.** In Section 4.3, we employ MIA-Bench, MMDU, MathVista, and MathVision, whose evaluation requires large language models as judges—a practice consistent with current research standards.
- **Usage 4: Paper grammar polishing.** The paper is initially drafted by humans and subsequently polished for grammar using LMMs, a practice consistent with current research norms.

## B    MORE DETAILS ABOUT MMEVOKE

In this section, we further demonstrate the details of MMEVOKE, including benchmark presentation, complete subfields distribution, word cloud distribution, human study, fine-grained difficulty level results and release plan.

### B.1    PRESENTATION OF MMEVOKE BENCHMARK

Figure 8 presents additional examples of MMEVOKE, encompassing four distinct subfields: Politics, Science, Video Game, and Songs. Each subfield showcases relevant Type, Knowledge Summary, Knowledge Image, Query, Query Image. Specifically, four examples are as follows:

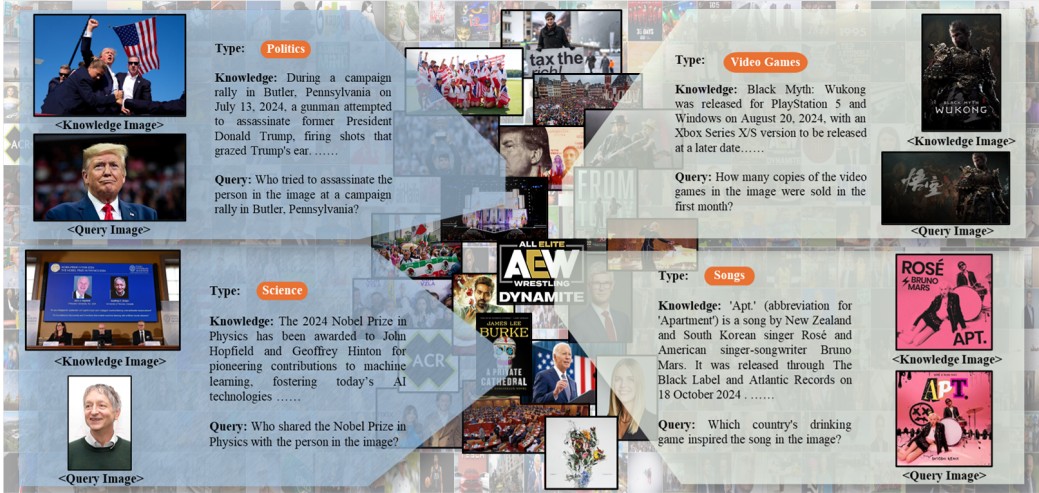

Figure 8: **Examples of News/Entity Evolving Knowledge in MMEVOKE**, including Type, Knowledge Summary, Knowledge Image, Query, Query Image. Examples are taken from different clusters: **Politics** for News, **Science** for News, **Video Game** for Entity, and **Songs** for Entity.

- **Politics:** Describes the unsuccessful assassination attempt targeting former U.S. President Donald Trump at a campaign rally in Butler, Pennsylvania, on July 13, 2024. The query question asks for the identity of the individual depicted in the image.
- **Science:** Details the awarding of the 2024 Nobel Prize in Physics to John Hopfield and Geoffrey Hinton for their contributions. The query question inquires about the person who shared the Nobel Prize with the individual shown in the image.
- **Video Game:** Lists the video game Black Myth: Wukong, released on August 20, 2024. The query question focuses on the game's sales figures during its first month.
- **Songs:** Introduces the song Apt, performed by Russ and Bruno Mars. The query question concerns the drinking game that served as inspiration for the song.

These examples illustrate the diverse subfields of evolving knowledge captured within MMEVOKE, providing a more detailed demonstration.

## B.2 WORD CLOUD DISTRIBUTION

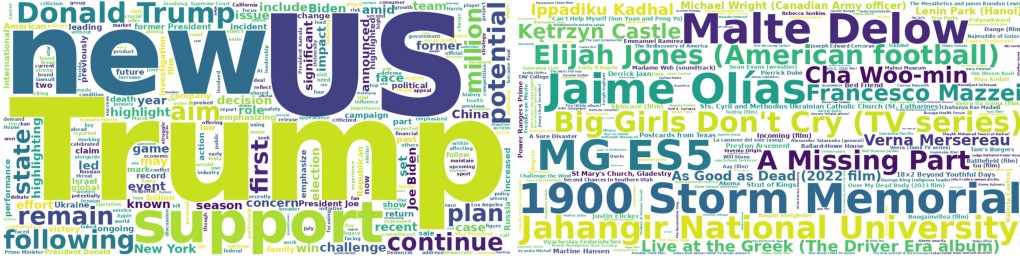

(a) News Evolving Knowledge.     (b) Entity Evolving Knowledge.

Figure 9: Word Cloud Distributions of MMEVOKE.

In Figure 9a, we show the word cloud distribution of News evolving knowledge. It can be found that Trump appears more often, which may be because MMEVOKE contains a large number of US political News data. Meanwhile, in Figure 9b, we present the word cloud distribution of entity names in the Entity evolving knowledge.

We have demonstrated the diversity of MMEVOKE benchmark through fine-grained subfields distribution, key statistics, word cloud distribution, and multiple perspectives. At the same time, our automated pipeline can continuously collect evolving knowledge and provide injection data for the knowledge injection field.

## B.3 COMPLETE SUBFIELDS DISTRIBUTION

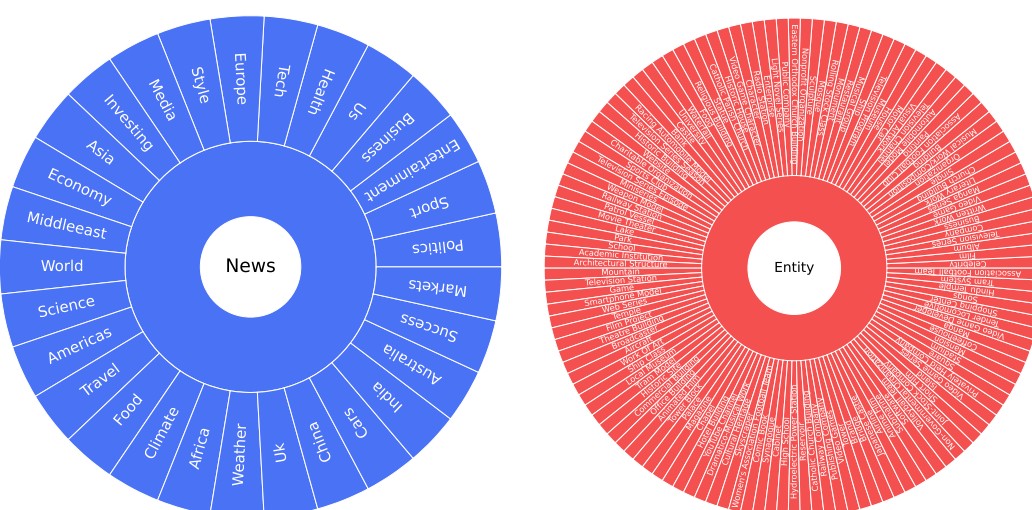

Figure 10: Fine-grained subfields distribution of News evolving knowledge.

Figure 11: Fine-grained subfields distribution of Entity evolving knowledge.

In Figures 10 and 11, we comprehensively illustrate the fine-grained subfields distribution of the MMEVOKE benchmark, which includes 29 distinct subfields for News evolving knowledge and 130 subfields for Entity evolving knowledge, underscoring its exceptional diversity. This benchmark serves as a critical resource for the evolving knowledge injection domain, providing a robust foundation for advancing research and development in the field.

## B.4 HUMAN STUDY TOWARDS BENCHMARK QUALITY TEST

To verify the hallucination level of GPT-4o in data generation, We randomly selected 100 pieces of data from MMEVOKE during manual selection for human study. Specifically, four annotators scored the samples (1-5 scales, higher scores indicate greater purity) from the perspectives of content

summarization, QA generation, and whether the summary contained information necessary to answer the question. According to the results in Table 4, MMEVOKE exhibits high quality, demonstrating minimal hallucination during the data construction process.

Table 4: Human Study Towards Benchmark Quality Test.

| Dimension | | ALL | News | Entity |
|---|---|---|---|---|
| **MMEVOKE** | Q&A | 4.86 $_{(\pm 0.01)}$ | 4.87 $_{(\pm 0.01)}$ | 4.85 $_{(\pm 0.02)}$ |
| | Summary | 4.98 $_{(\pm 0.01)}$ | 4.97 $_{(\pm 0.01)}$ | 4.98 $_{(\pm 0.02)}$ |

## B.5 DENSITY DISTRIBUTION

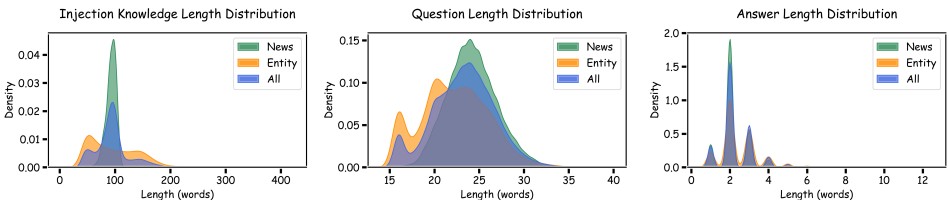

Figure 12: Density distribution based on evolving knowledge sources.

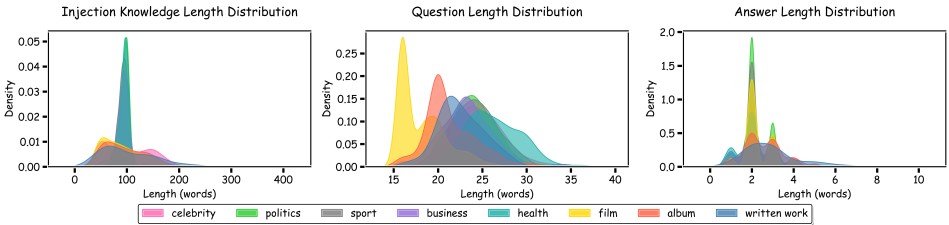

Figure 13: Density distribution of fine-grained subfields based on evolving knowledge.

## B.6 FINE-GRAINED DIFFICULTY LEVEL OF MMEVOKE

To further diversify MMEVOKE, we constructed 568 Counterfactual Reasoning and 3-Hop QA pairs using GPT-4o, and extracted their corresponding SimpleVQA data, yielding experimental results comparing fine-grained difficulty levels. ***The SimpleVQA here refers to the QA data of* MMEVOKE *itself.*** Table 5 shows the difficulty ranking: *Counterfactual Reasoning < SimpleVQA < 3-Hop*, and 48.24% (avg) of cases have SimpleVQA failing while Counterfactual Reasoning succeeding, and 40.06% (avg) have SimpleVQA succeeding but 3-Hop failing.

Table 5: The performance of different difficulty levels on MMEVOKE.

| Task | Method | ALL | | News | | Entity | |
|---|---|---|---|---|---|---|---|
| | | CEM | F1-Score | CEM | F1-Score | CEM | F1-Score |
| SimpleVQA | Full-FT | 16.55 | 14.82 | 17.43 | 14.12 | 15.53 | 15.61 |
| | Sufficient Context | 55.63 | 76.00 | 55.59 | 72.05 | 55.68 | 80.54 |
| 3-Hop | Full-FT | 12.15 | 5.65 | 11.18 | 5.22 | 13.26 | 6.14 |
| | Sufficient Context | 40.49 | 52.58 | 38.16 | 51.49 | 43.18 | 53.82 |
| Counterfactual Reasoning | Full-FT | 70.42 | 70.42 | 74.01 | 74.01 | 66.29 | 66.29 |
| | Sufficient Context | 76.58 | 76.58 | 65.46 | 65.46 | 89.39 | 89.39 |

# C  MORE RESULTS ABOUT MMEVOKE

## C.1  MORE QUANTITATIVE EXPERIMENTAL RESULTS ABOUT RQ1

Table 6: **Performance of knowledge injection methods on MMEVOKE.** ALL, News.Avg, and Entity.Avg respectively show the performance of knowledge injection methods on entire MMEVOKE, News subset, and Entity subset. Orange value marks the best performance of methods on LLaVA-v1.5 and Qwen-VL-Chat, as well as the best performance of models in Web Search Engine and Sufficient Context **(vertical perspective)**. Red value indicates knowledge subfield with the best performance of the same method and model on different fine-grained subfields, while blue value indicates knowledge subfield with the worst performance **(horizontal perspective)**. PO: Politics; SP: Sports; BU: Business; HE: Health; CE: Celebrity; FI: Film; AL: Album; WR: Written Work.

| Method | ALL | | News | | | | | | | | | | Entity | | | | | | | | | |
|---|---|---|---|---|---|---|---|---|---|---|---|---|---|---|---|---|---|---|---|---|---|---|
| | | | Avg | | PO | | SP | | BU | | HE | | Avg | | CE | | FI | | AL | | WR | |
| | CEM↑ | F1↑ | CEM↑ | F1↑ | CEM↑ | F1↑ | CEM↑ | F1↑ | CEM↑ | F1↑ | CEM↑ | F1↑ | CEM↑ | F1↑ | CEM↑ | F1↑ | CEM↑ | F1↑ | CEM↑ | F1↑ | CEM↑ | F1↑ |
| *LLaVA-v1.5* | | | | | | | | | | | | | | | | | | | | | | |
| Vanilla | 4.89 | 9.34 | 7.37 | 11.96 | 1.92 | 5.86 | 4.59 | 9.74 | 10.70 | 15.99 | 10.12 | 17.54 | 2.18 | 6.47 | 1.37 | 6.48 | 2.39 | 5.71 | 3.77 | 6.02 | 6.78 | 11.24 |
| Full-FT | 18.02 | 15.17 | 21.35 | 16.34 | 12.92 | 10.99 | 22.49 | 20.88 | 27.31 | 20.95 | 19.84 | 16.47 | 14.37 | 13.88 | 13.11 | 16.93 | 12.39 | 13.16 | 12.17 | 7.66 | 20.34 | 8.43 |
| LoRA | 15.23 | 18.31 | 17.72 | 19.42 | 10.54 | 12.96 | 19.11 | 21.50 | 20.66 | 24.03 | 17.81 | 23.76 | 12.51 | 17.09 | 12.20 | 21.19 | 12.39 | 15.82 | 10.72 | 8.72 | 20.34 | 12.94 |
| MM-RAGText-Only | 24.05 | 34.32 | 37.32 | 49.39 | 22.18 | 36.25 | 47.88 | 54.77 | 34.87 | 51.07 | 36.44 | 50.95 | 9.50 | 17.80 | 15.14 | 25.39 | 1.93 | 4.04 | 2.90 | 13.86 | 3.39 | 13.07 |
| MM-RAGImage-Only | 25.25 | 37.11 | 19.28 | 26.76 | 9.35 | 16.96 | 33.37 | 39.19 | 19.56 | 29.46 | 18.22 | 28.60 | 31.80 | 48.45 | 26.37 | 43.01 | 39.09 | 47.58 | 40.29 | 58.14 | 28.81 | 53.68 |
| MM-RAGUniIR | 40.68 | 57.51 | 40.12 | 53.21 | 21.81 | 35.08 | 56.23 | 65.94 | 39.85 | 57.08 | 35.22 | 50.93 | 41.30 | 62.23 | 41.01 | 63.94 | 48.86 | 58.98 | 41.45 | 63.02 | 35.59 | 60.09 |
| *Qwen-VL-Chat* | | | | | | | | | | | | | | | | | | | | | | |
| Vanilla | 5.84 | 10.99 | 7.75 | 12.72 | 3.21 | 7.69 | 4.47 | 10.37 | 10.52 | 14.92 | 10.93 | 19.32 | 3.74 | 9.10 | 1.78 | 8.06 | 8.18 | 13.10 | 4.35 | 6.93 | 8.47 | 16.81 |
| Full-FT | 10.16 | 16.61 | 13.35 | 18.22 | 6.42 | 11.80 | 12.70 | 17.11 | 16.42 | 22.27 | 17.00 | 25.42 | 6.65 | 14.83 | 5.39 | 14.68 | 11.59 | 17.95 | 5.22 | 10.83 | 15.25 | 21.69 |
| LoRA | 6.95 | 12.64 | 9.27 | 14.55 | 4.31 | 9.24 | 5.68 | 11.82 | 12.55 | 17.79 | 12.96 | 21.64 | 4.41 | 10.54 | 2.34 | 9.54 | 9.32 | 14.96 | 5.22 | 8.04 | 10.17 | 18.07 |
| MM-RAGText-Only | 21.79 | 31.28 | 31.51 | 41.14 | 20.71 | 29.81 | 30.71 | 40.75 | 32.29 | 43.38 | 33.20 | 47.56 | 11.13 | 20.47 | 13.36 | 24.27 | 8.41 | 14.02 | 6.67 | 15.27 | 11.86 | 19.60 |
| MM-RAGImage-Only | 22.31 | 33.09 | 17.82 | 25.15 | 9.26 | 15.97 | 20.80 | 29.82 | 18.45 | 28.33 | 18.62 | 29.38 | 27.24 | 41.79 | 20.27 | 33.52 | 33.98 | 45.81 | 39.42 | 53.80 | 33.90 | 54.43 |
| MM-RAGUniIR | 32.75 | 46.18 | 33.26 | 43.36 | 18.15 | 27.56 | 32.77 | 44.90 | 37.08 | 49.25 | 31.98 | 44.96 | 32.20 | 49.28 | 28.20 | 45.05 | 37.16 | 50.60 | 41.45 | 56.57 | 42.37 | 65.29 |
| *Commercial AI Web Search Engines* | | | | | | | | | | | | | | | | | | | | | | |
| Gemini-2.0-Flash | 18.21 | 26.52 | 21.23 | 27.75 | 10.91 | 16.87 | 21.64 | 27.45 | 22.88 | 30.03 | 17.41 | 28.32 | 14.91 | 25.16 | 10.11 | 20.35 | 28.64 | 37.47 | 14.49 | 23.87 | 16.95 | 28.77 |
| Gemini-2.5-Pro | 44.19 | 52.58 | 48.86 | 52.84 | 39.07 | 52.28 | 31.90 | 37.00 | 51.11 | 57.22 | 58.04 | 59.97 | 39.27 | 46.27 | 24.29 | 35.81 | 63.98 | 73.14 | 53.62 | 68.36 | 42.37 | 57.40 |
| Perplexity AI | 48.27 | 62.44 | 47.58 | 56.51 | 34.78 | 43.14 | 56.13 | 66.19 | 41.82 | 54.33 | 35.29 | 47.88 | 48.96 | 68.78 | 47.03 | 70.95 | 62.22 | 73.65 | 54.41 | 68.54 | 43.75 | 59.17 |
| GPT-4.1 | 39.61 | 42.69 | 41.81 | 43.08 | 25.23 | 26.07 | 52.60 | 52.43 | 34.82 | 42.45 | 47.60 | 50.81 | 37.19 | 42.26 | 24.29 | 26.53 | 57.50 | 62.41 | 58.26 | 62.94 | 30.51 | 47.67 |
| *Sufficient Context* | | | | | | | | | | | | | | | | | | | | | | |
| LLaVA-v1.5 | 56.13 | 75.77 | 56.78 | 72.37 | 38.77 | 58.44 | 75.09 | 84.69 | 54.61 | 74.33 | 48.58 | 67.01 | 55.43 | 79.50 | 52.08 | 78.83 | 75.91 | 89.71 | 57.39 | 78.80 | 49.15 | 69.96 |
| Qwen-VL-Chat | 48.96 | 66.02 | 49.98 | 63.42 | 35.20 | 50.29 | 52.00 | 68.90 | 50.55 | 67.25 | 48.18 | 62.02 | 47.84 | 68.87 | 43.29 | 66.15 | 62.05 | 75.92 | 58.55 | 75.41 | 47.46 | 67.79 |
| Gemini-2.5-Pro | 72.15 | 80.46 | 72.61 | 78.77 | 57.01 | 65.75 | 86.34 | 89.63 | 71.77 | 81.65 | 62.35 | 74.65 | 71.65 | 82.32 | 73.53 | 80.89 | 81.14 | 88.09 | 75.07 | 85.59 | 52.54 | 72.05 |
| GPT-4.1 | 75.02 | 83.74 | 79.22 | 88.20 | 53.62 | 65.21 | 84.04 | 90.23 | 69.37 | 80.75 | 68.83 | 79.56 | 71.21 | 79.68 | 80.74 | 88.02 | 88.18 | 91.97 | 86.38 | 91.58 | 59.32 | 74.86 |

Table 6 presents the quantitative experimental results of RQ1, revealing that no method achieves robust injection performance, with significant performance variance observed across different fine-grained subfields knowledge. Specifically, We have obtained further observations:

- **Obs 1:** In Table 6, across nearly all evaluated methods, News knowledge injection performance consistently outperforms Entity knowledge. We attribute this gap to their fundamental differences in learning difficulty. Entity knowledge introduces entirely novel concepts to model, posing a substantial learning challenge. In contrast, News knowledge primarily establishes new and complex relationships among existing entities, which represents a comparatively lower learning barrier.
- **Obs 2:** The performance of knowledge in the same subfield varies depending on the method used. For example, in Full FT, LoRA, and MM-RAGText-Only, the performance of film knowledge is poor. In sharp contrast, it performs better when using MM-RAGImage-Only, MM-RAGUniIR, Sufficient Context, and Web Search.
- **Obs 3:** A significant performance variance among different strategies within same method. Notably, MM-RAGText-Only is more effective for injecting News knowledge, while MM-RAGImage-Only is better suited for Entity knowledge. This discrepancy indicates that knowledge injection is optimized when the modality of the feature aligns with the nature of the knowledge source (textual features for News and visual features for Entity).
- **Obs 4:** The performance of the same subfield knowledge differs across models. For instance, Health and Written work perform better on Qwen-VL-Chat; Sport and Business perform better on LLaVA-v1.5. This is likely due to significant distributional differences in types of knowledge data encountered during pre-training of different models.
- **Obs 5:** Politics knowledge contains a wide range of professional terms and complex concepts that are difficult to learn, ranking lowest among almost all methods.

> **Observations**
>
> **Observation 1:** Current knowledge injection methods have significant domain specificity for different fine-grained subfield knowledge.

Table 7: **The performance of knowledge injection methods on Entity subset of MMEVOKE.** TEL: Television Series; COM: Company; VID: Video Game; CHU: Church Building; SIN: Single; OGR: Organization; PAI: Painting; MOT: Motor Car.

| Method | TEL | | COM | | VID | | CHU | | SIN | | ORG | | PAI | | MOT | |
|---|---|---|---|---|---|---|---|---|---|---|---|---|---|---|---|---|
| | CEM↑ | F1↑ | CEM↑ | F1↑ | CEM↑ | F1↑ | CEM↑ | F1↑ | CEM↑ | F1↑ | CEM↑ | F1↑ | CEM↑ | F1↑ | CEM↑ | F1↑ |
| *LLaVA-v1.5* | | | | | | | | | | | | | | | | |
| Vanilla | 6.15 | 9.77 | 1.12 | 5.69 | 0.00 | 3.16 | 0.00 | 6.39 | 4.55 | 9.51 | 2.70 | 6.31 | 0.00 | 11.90 | 0.00 | 4.76 |
| Full-FT | 13.97 | 10.29 | 29.21 | 14.15 | 10.34 | 7.32 | 26.53 | 22.67 | 15.91 | 8.55 | 27.03 | 15.52 | 17.86 | 13.83 | 7.14 | 6.21 |
| LoRA | 15.64 | 16.20 | 10.11 | 11.42 | 12.07 | 15.24 | 14.29 | 24.54 | 20.45 | 20.39 | 16.22 | 17.45 | 14.29 | 14.42 | 0.00 | 1.41 |
| MM-RAG[Text-Only] | 3.35 | 6.15 | 4.49 | 14.31 | 5.17 | 21.81 | 8.16 | 18.10 | 2.27 | 20.72 | 2.70 | 13.69 | 14.29 | 21.31 | 7.14 | 27.55 |
| MM-RAG[Image-Only] | 36.87 | 54.26 | 30.34 | 57.23 | 29.31 | 59.73 | 40.82 | 66.33 | 34.09 | 56.78 | 24.32 | 49.88 | 53.57 | 70.95 | 21.43 | 57.93 |
| MM-RAG[UniIR] | 41.34 | 62.91 | 30.34 | 63.49 | 32.76 | 65.77 | 34.69 | 64.30 | 31.82 | 61.50 | 29.73 | 59.19 | 64.29 | 85.12 | 21.43 | 68.30 |
| *Qwen-VL-Chat* | | | | | | | | | | | | | | | | |
| Vanilla | 7.82 | 11.33 | 1.12 | 7.32 | 1.72 | 2.59 | 0.00 | 10.20 | 6.82 | 11.33 | 0.00 | 2.88 | 7.14 | 13.10 | 0.00 | 10.37 |
| Full-FT | 8.94 | 16.49 | 1.12 | 11.05 | 3.45 | 15.54 | 2.04 | 16.91 | 6.82 | 15.75 | 5.41 | 8.61 | 10.71 | 12.93 | 7.14 | 15.48 |
| LoRA | 7.26 | 11.55 | 1.12 | 8.64 | 1.72 | 3.85 | 2.04 | 9.90 | 6.82 | 13.61 | 2.70 | 5.59 | 10.71 | 15.95 | 0.00 | 8.33 |
| MM-RAG[Text-Only] | 7.26 | 13.22 | 7.87 | 23.37 | 8.62 | 25.35 | 4.08 | 12.90 | 13.64 | 31.20 | 13.51 | 23.45 | 14.29 | 23.45 | 14.29 | 30.36 |
| MM-RAG[Image-Only] | 22.91 | 38.39 | 30.34 | 55.94 | 18.97 | 56.23 | 38.78 | 52.91 | 31.82 | 56.92 | 29.73 | 45.95 | 39.29 | 48.45 | 14.29 | 46.90 |
| MM-RAG[UniIR] | 19.67 | 23.81 | 30.34 | 63.84 | 18.97 | 59.04 | 28.57 | 50.26 | 34.09 | 59.51 | 43.24 | 63.13 | 42.86 | 52.62 | 14.29 | 46.90 |
| *Commercial AI Web Search Engines* | | | | | | | | | | | | | | | | |
| Gemini-2.0-Flash | 19.55 | 31.14 | 8.99 | 20.82 | 10.34 | 25.01 | 10.20 | 21.56 | 9.09 | 22.58 | 18.92 | 25.02 | 14.29 | 16.43 | 0.00 | 26.11 |
| Gemini-2.5-Pro | 58.10 | 74.71 | 41.57 | 66.09 | 46.55 | 65.25 | 20.41 | 33.07 | 43.18 | 66.37 | 43.24 | 59.98 | 46.43 | 38.27 | 7.14 | 35.48 |
| Perplexity AI | 43.90 | 54.59 | 30.00 | 52.08 | 33.33 | 48.41 | 62.50 | 75.83 | 50.00 | 70.00 | 33.33 | 54.07 | 85.71 | 83.67 | 33.33 | 13.33 |
| GPT-4.1 | 50.28 | 62.08 | 52.81 | 57.02 | 53.45 | 65.23 | 22.45 | 29.31 | 38.64 | 47.03 | 45.95 | 52.43 | 17.86 | 20.53 | 0.00 | 15.99 |
| *Sufficient Context* | | | | | | | | | | | | | | | | |
| LLaVA-v1.5 | 56.42 | 81.18 | 41.57 | 78.05 | 34.48 | 68.72 | 44.90 | 72.48 | 45.45 | 68.79 | 45.95 | 79.70 | 75.00 | 90.12 | 35.71 | 73.15 |
| Qwen-VL-Chat | 51.96 | 72.08 | 39.33 | 73.62 | 25.86 | 63.28 | 34.69 | 62.88 | 36.36 | 62.62 | 43.24 | 65.69 | 42.86 | 55.60 | 42.86 | 73.47 |
| Gemini-2.5-Pro | 69.27 | 85.95 | 64.04 | 81.32 | 58.62 | 78.70 | 55.10 | 75.18 | 68.18 | 82.72 | 56.76 | 78.37 | 89.29 | 85.62 | 50.00 | 78.25 |
| GPT-4.1 | 77.09 | 90.22 | 70.79 | 86.21 | 67.24 | 83.84 | 59.18 | 77.77 | 79.55 | 91.44 | 64.86 | 83.24 | 89.29 | 91.90 | 64.29 | 84.97 |

Table 8: **The performance of knowledge injection methods on News subset of MMEVOKE.** ENT: Entertainment; TEC: Tech; SCI: Science; TRA: Travel; FOO: Food; CLI: Climate; INV: Investing; STY: Style.

| Method | ENT | | TEC | | SCI | | TRA | | FOO | | CLI | | INV | | STY | |
|---|---|---|---|---|---|---|---|---|---|---|---|---|---|---|---|---|
| | CEM↑ | F1↑ | CEM↑ | F1↑ | CEM↑ | F1↑ | CEM↑ | F1↑ | CEM↑ | F1↑ | CEM↑ | F1↑ | CEM↑ | F1↑ | CEM↑ | F1↑ |
| *LLaVA-v1.5* | | | | | | | | | | | | | | | | |
| Vanilla | 6.79 | 9.35 | 6.79 | 9.35 | 6.79 | 9.35 | 11.90 | 18.57 | 10.26 | 17.83 | 8.11 | 13.87 | 18.28 | 23.71 | 13.93 | 16.20 |
| Full-FT | 18.67 | 11.47 | 28.29 | 17.02 | 15.79 | 12.56 | 28.57 | 24.16 | 35.90 | 24.54 | 27.03 | 13.02 | 44.09 | 25.06 | 31.15 | 19.17 |
| LoRA | 16.98 | 15.70 | 27.63 | 25.96 | 8.77 | 18.73 | 23.81 | 29.91 | 20.51 | 18.83 | 16.22 | 18.02 | 34.41 | 28.13 | 19.67 | 19.45 |
| MM-RAG[Text-Only] | 39.81 | 48.79 | 46.05 | 55.21 | 36.84 | 55.71 | 38.10 | 54.50 | 33.33 | 50.85 | 37.84 | 53.51 | 37.63 | 47.06 | 68.85 | 78.51 |
| MM-RAG[Image-Only] | 21.76 | 28.07 | 23.03 | 28.02 | 22.81 | 38.42 | 21.43 | 30.09 | 23.08 | 36.32 | 18.92 | 26.04 | 25.81 | 31.61 | 22.13 | 25.67 |
| MM-RAG[UniIR] | 52.16 | 63.67 | 42.11 | 51.77 | 33.33 | 52.89 | 47.62 | 62.83 | 41.03 | 57.78 | 35.14 | 53.06 | 38.71 | 48.23 | 59.84 | 67.32 |
| *Qwen-VL-Chat* | | | | | | | | | | | | | | | | |
| Vanilla | 6.79 | 9.90 | 14.47 | 16.10 | 8.77 | 14.95 | 9.52 | 16.59 | 10.26 | 16.24 | 10.81 | 12.07 | 23.66 | 29.27 | 13.11 | 16.19 |
| Full-FT | 11.27 | 14.64 | 17.11 | 18.79 | 8.77 | 13.78 | 14.29 | 23.89 | 17.95 | 27.35 | 18.92 | 21.42 | 35.48 | 38.34 | 16.39 | 19.18 |
| LoRA | 7.41 | 11.01 | 16.45 | 18.76 | 8.77 | 13.93 | 7.14 | 15.00 | 7.69 | 17.52 | 13.51 | 14.77 | 24.73 | 30.44 | 15.57 | 17.72 |
| MM-RAG[Text-Only] | 31.48 | 38.00 | 46.71 | 51.27 | 42.11 | 48.99 | 38.10 | 50.56 | 20.51 | 39.66 | 35.14 | 46.65 | 43.01 | 52.75 | 60.66 | 66.14 |
| MM-RAG[Image-Only] | 20.06 | 24.82 | 22.37 | 27.06 | 33.33 | 42.59 | 21.43 | 31.67 | 20.51 | 27.35 | 24.32 | 31.40 | 30.11 | 36.37 | 19.67 | 23.81 |
| MM-RAG[UniIR] | 42.75 | 50.25 | 41.45 | 45.18 | 47.37 | 55.69 | 40.48 | 50.46 | 28.21 | 44.36 | 32.43 | 44.34 | 43.01 | 52.93 | 51.64 | 56.70 |
| *Commercial AI Web Search Engines* | | | | | | | | | | | | | | | | |
| Gemini-2.0-Flash | 24.69 | 29.98 | 38.82 | 46.00 | 15.79 | 22.97 | 16.67 | 30.40 | 23.08 | 30.52 | 10.81 | 19.28 | 38.71 | 45.72 | 30.33 | 32.60 |
| Gemini-2.5-Pro | 59.72 | 61.28 | 63.82 | 60.26 | 31.58 | 37.64 | 52.38 | 63.00 | 48.72 | 56.44 | 48.65 | 44.35 | 52.69 | 51.29 | 69.67 | 68.13 |
| Perplexity AI | 59.85 | 64.15 | 47.06 | 55.20 | 45.45 | 49.13 | 50.00 | 70.05 | 33.33 | 40.74 | 37.50 | 64.58 | 33.33 | 40.12 | 71.88 | 74.36 |
| GPT-4.1 | 46.30 | 43.64 | 57.24 | 59.50 | 22.81 | 35.29 | 50.00 | 50.29 | 66.67 | 56.89 | 40.54 | 35.21 | 55.91 | 55.73 | 50.82 | 50.84 |
| *Sufficient Context* | | | | | | | | | | | | | | | | |
| LLaVA-v1.5 | 65.12 | 78.31 | 63.82 | 77.61 | 47.37 | 66.30 | 57.14 | 72.37 | 51.28 | 76.58 | 51.35 | 63.07 | 60.22 | 72.83 | 75.41 | 85.18 |
| Qwen-VL-Chat | 61.42 | 68.99 | 62.50 | 72.69 | 43.86 | 63.14 | 45.24 | 58.56 | 51.28 | 64.66 | 48.65 | 56.68 | 53.76 | 65.04 | 68.03 | 75.70 |
| Gemini-2.5-Pro | 81.17 | 83.08 | 75.00 | 82.33 | 61.40 | 66.34 | 73.81 | 82.47 | 66.67 | 81.28 | 70.27 | 74.10 | 75.27 | 77.29 | 82.79 | 83.34 |
| GPT-4.1 | 78.70 | 83.73 | 82.89 | 85.12 | 61.40 | 72.69 | 69.05 | 80.41 | 69.23 | 78.69 | 62.16 | 67.85 | 68.82 | 77.61 | 89.34 | 91.33 |

Tables 7 and 8 present richer experimental results of fine-grained subfields, further verifying the significant domain specificity of existing knowledge injection methods and their inability to robustly implement knowledge injection.

## C.2 SEQUENTIAL FINE-TUNING

### C.2.1 SEQUENTIAL FINE-TUNING BASED ON TASKS

Sequential Fine-Tuning refers to the process of incrementally training models on new tasks and data. Specifically, model weights obtained from previous tasks and data are used to initialize model parameters (Chen et al., 2025). In this section, *we explore whether Sequential Fine-Tuning is more*

*effective than One-Time Injection?* We employed MMEVOKE for knowledge injection, randomly dividing the data into subsets of 4, 8, and 12 tasks. We consider each subset as a task and use these subsets to Sequential Fine-Tuning the model.

***Sequential Fine-Tuning impede the effective injection of multimodal evolving knowledge.*** As illustrated in Figure 14, the performance of LMMs exhibits a declining trend with progressive Sequential Fine-Tuning based on tasks. This degradation primarily stems from the disruption of previously fine-tuning parameters during each subsequent fine-tuning iteration. Consequently, the overall performance of LMMs progressively deteriorates. Furthermore, our investigation into the impact of Sequential Fine-Tuning steps revealed a negative correlation between the number of steps $g$ and LMMs performance, as evidenced by the values corresponding to the terminal points in each line graph. These findings underscore the importance of minimizing Sequential Fine-Tuning in practical applications to preserve model efficacy.

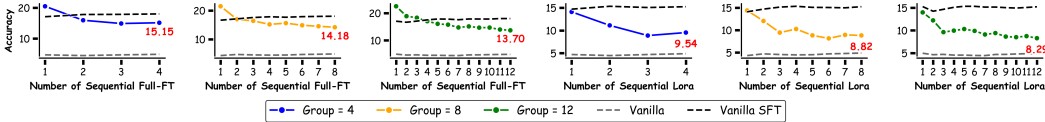

Figure 14: **The results of LLaVA-v1.5 on Sequential Fine-Tuning based on Tasks.** The data $\mathcal{D}_{\mathcal{K}}$ and $\mathcal{D}_{\mathcal{Q}}$ are evenly divided into $g \in \{4, 8, 12\}$ parts, namely $\mathcal{D}_{\mathcal{K}} = \left\{ d_k^1, d_k^2, \ldots, d_k^n \right\}_{n=1}^g$ and $\mathcal{D}_{\mathcal{Q}} = \left\{ d_q^1, d_q^2, \ldots, d_q^n \right\}_{n=1}^g$. Sequential Fine-Tuning based on tasks refer to the situation where if the current m-th Sequential Fine-Tuning has ended, it indicates that the model is being trained on $d_k^1, d_k^2, \ldots, d_k^m$ in sequence; and evaluated on $\left\{ d_q^1 \cup d_q^2 \cup \cdots \cup d_q^m \right\}$.

### C.2.2 SEQUENTIAL FINE-TUNING BASED ON SUBSETS

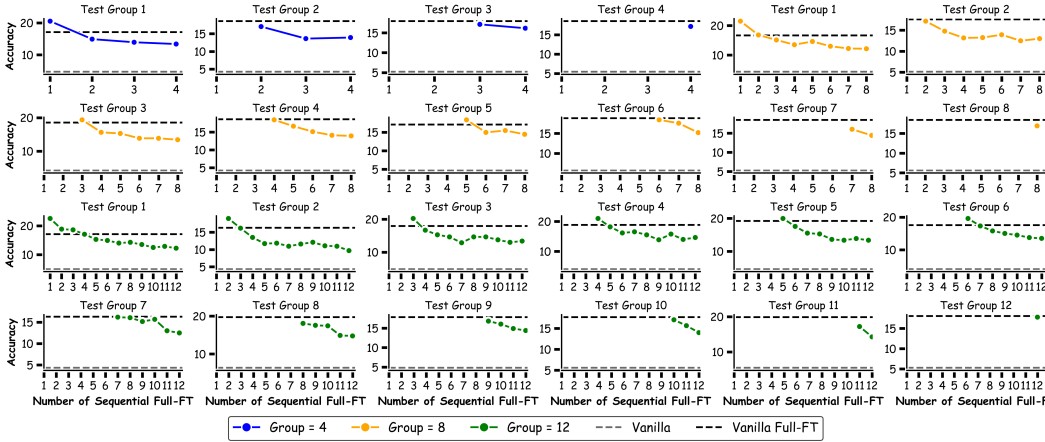

Figure 15: **The results of LLaVA-v1.5 on Sequential Full-FT based on Subsets.** Sequential Full-FT based on subset refer to the situation where if the current m-th Sequential Full-FT has ended, it indicates that the model is being trained on $d_k^1, d_k^2, \ldots, d_k^m$ in sequence; and evaluate sequentially on **one of** $d_q^1, d_q^2, \ldots, d_q^m$.

The results of Sequential Fine-Tuning based on subsets are shown in Figure 15 and 16. Each subgraph displays the performance changes of the LMMs on the same subset as the Sequential Fine-Tuning process progresses. It can be observed that whether using Full-FT or LoRA as training strategies, as the number $g$ of Sequential Fine-Tuning increases, the performance of the model on the same subset shows a downward trend. This discovery further indicates that Sequential Fine-Tuning is not conducive to injecting up-to-date knowledge into the LMMs.

> **Observations**
>
> **Observation 2:** Both sequential task and subset fine-tuning impede the efficacy of knowledge injection, with performance degradation correlating with an increased number of tasks or subsets.

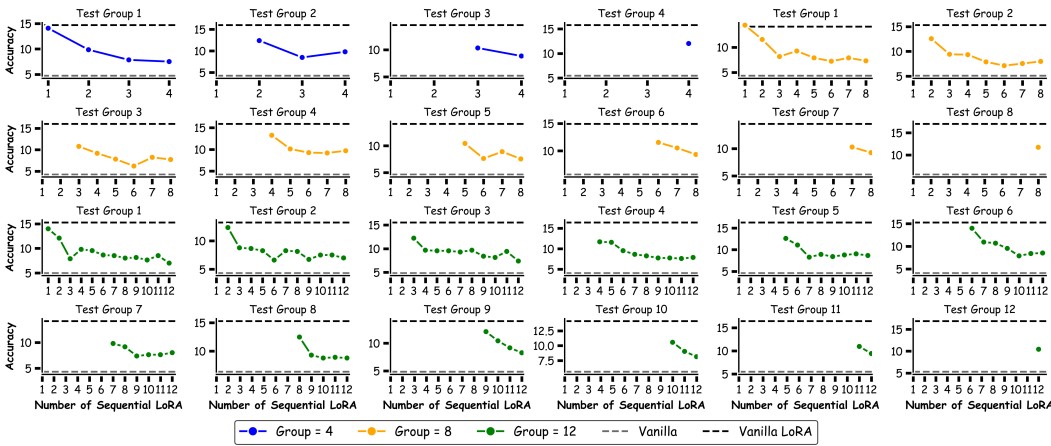

Figure 16: **The results of LLaVA-v1.5 on Sequential LoRA based on Subsets.** Sequential LoRA based on subset refer to the situation where if the current m-th Sequential LoRA has ended, it indicates that the model is being trained on $d_k^1, d_k^2, \ldots, d_k^m$ in sequence; and evaluate sequentially on **one of** $d_q^1, d_q^2, \ldots, d_q^m$.

## C.3   ABLATION EXPERIMENTS IN MM-RAG

*Retrieval strategy*, *Example Number*, and *Pool Size* are critical factors influencing the performance of MM-RAG, as demonstrated by the experimental results presented in Figure 17 and 18.

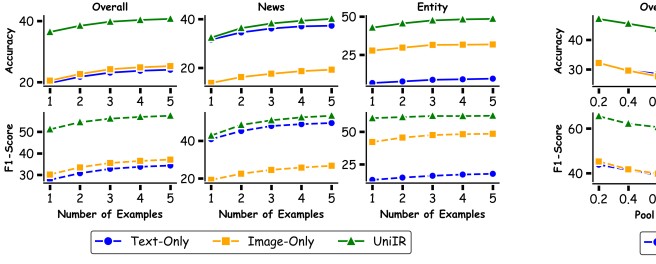

Figure 17: The results of LLaVA-v1.5's ablation study on MM-RAG about **Retrieval Strategy and Example Number** analysis.

Figure 18: The results of LLaVA-v1.5's ablation study on MM-RAG about **Retrieval Strategy and Pool Size** analysis.

- *Effect of Retrieval Strategy in MM-RAG.* An interesting observation appears in the "News" subgraph, where the Text-Only approach significantly outperforms the Image-Only strategy. The reason for this difference is that textual information is more important for news understanding than visual information, as valuable data cannot be retrieved solely through images. On the contrary, for Entity knowledge, visual information is more valuable than textual information.
- *Effect of Example Number in MM-RAG.* We compared $K \in \{1, \ldots, 5\}$, and in the first row of Figure 17, the direct correlation between the performance of model and Example Number is shown. Our experiment revealed a convincing trend that the model performs using a monotonically increasing function of Example Number $K$ for three retrieval strategies. This observation indicates that an increase in the example number brings more diverse reference information, which has a positive effect on the model's understanding and utilization of evolving knowledge.
- *Effect of Retrieval Pool Size in MM-RAG.* Regarding the ablation experiment of pool size, our setup is to randomly select 20% of the corresponding data from $\mathcal{D}_{\mathcal{Q}}$ and $\mathcal{D}_{\mathcal{K}}$ as $\mathcal{D}_{\mathcal{Q}}^{20\%}$ and $\mathcal{D}_{\mathcal{K}}^{20\%}$; For instance, when Pool Size = 20%, Retrieve Pool = $\mathcal{D}_{\mathcal{Q}}^{20\%}$; When Pool Size = 60%, Retrieve Pool = $\mathcal{D}_{\mathcal{K}}^{20\%} + \mathcal{D}_{\mathcal{J}}$, where $\mathcal{D}_{\mathcal{J}}$ is a randomly selected 40% data from the $\mathcal{D}_{\mathcal{K}} \setminus D_{K}^{20\%}$. The evaluation data is always $\mathcal{D}_{\mathcal{Q}}^{20\%}$. The experimental results, presented in the second row of Figure 18, demonstrate an inverse correlation between MM-RAG's performance and Pool Size. This suggests that larger pool sizes hinder the retriever's ability to identify relevant information, a critical consideration for practical MM-RAG applications.

**Observation 3:** Cross-modal retrieval strategies, a larger number of examples, and a smaller retrieval pool size all contribute to strengthening knowledge injection performance.

## C.4    MORE QUALITATIVE RESULTS ABOUT MMEVOKE

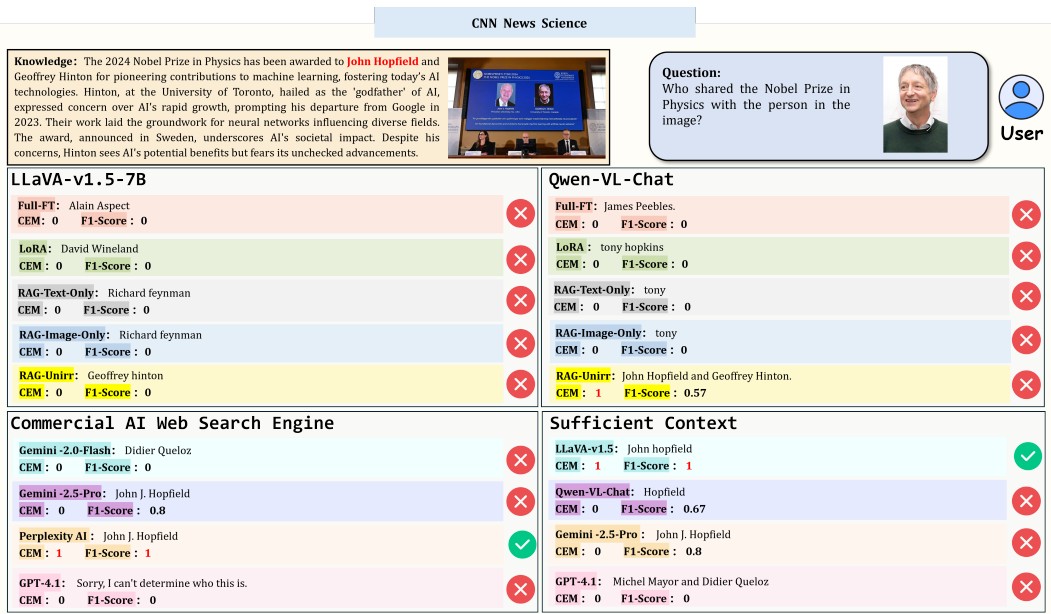

Figure 19: Qualitative example of CNN News science knowledge.

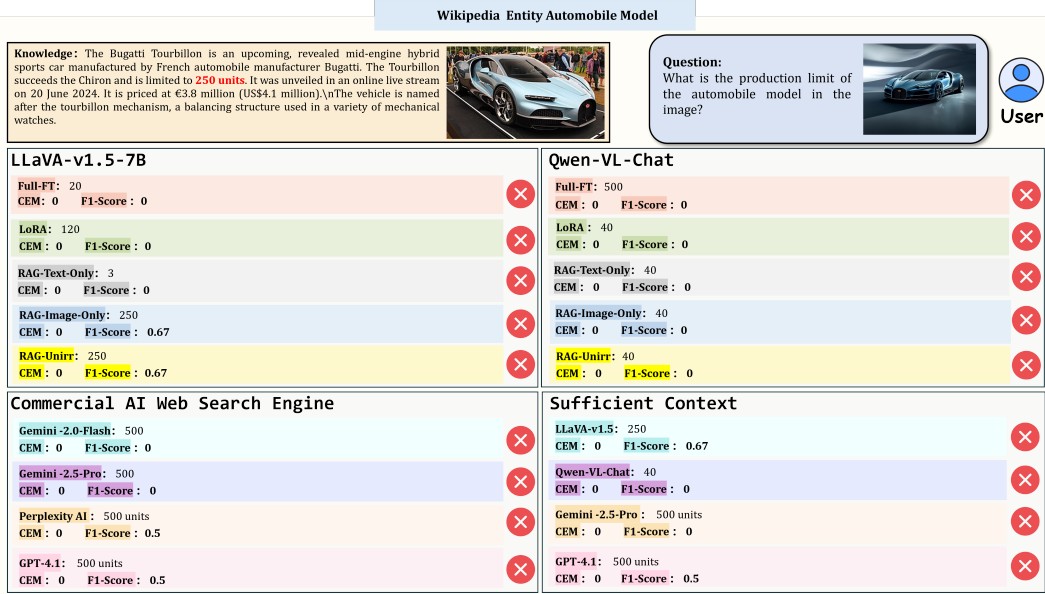

Figure 20: Qualitative example of Wikipedia Entity automobile model knowledge.

## C.5    ERROR ANALYSIS

Observing the qualitative examples in Figures 19, 20, and 21, we find that, as demonstrated by the results in Table 6, existing knowledge injection methods perform poorly on MMEVOKE, with even sufficient context failing to achieve perfect performance. Here, we conduct a detailed analysis of sufficient context.

Even when provided with sufficient context, the model still generates hallucinations. For instance, in Figure 19, the response given by GPT-4.1 is entirely unrelated to the question and does not appear in the sufficient context, representing a severe hallucination phenomenon. A similar hallucination issue persists in Figure 20. These concrete results indicate that merely improving the sufficiency of context is far from adequate—the model's inherent reasoning and ability to utilize contextual information are equally critical. Hallucination remains an urgent problem to be addressed.

**Observations**

**Observation 4:** Despite being provided with sufficient context, the model still exhibits severe hallucinations.

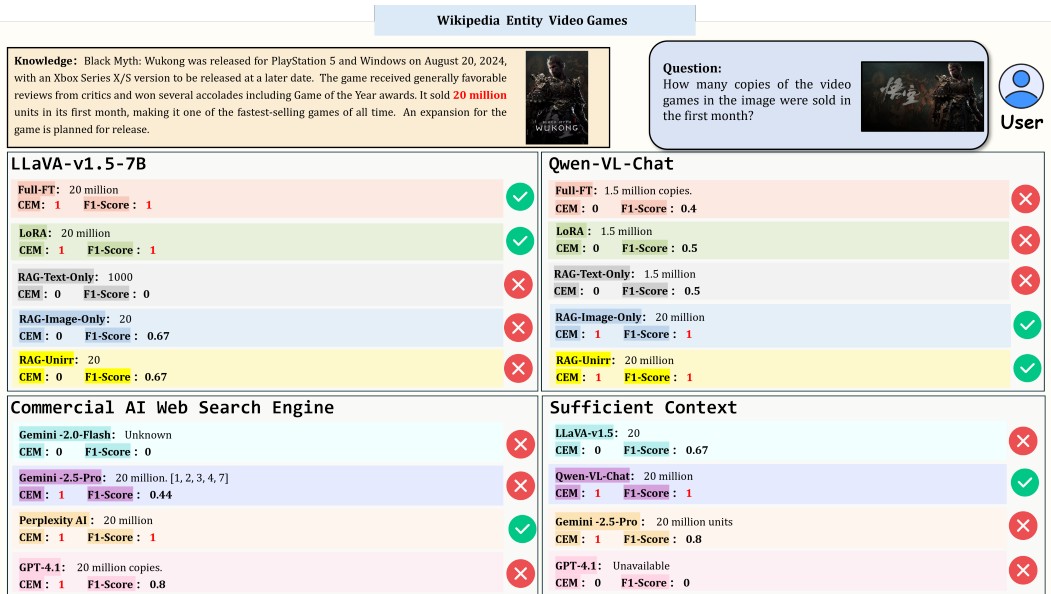

Figure 21: Qualitative example of Wikipedia Entity video games knowledge.

# D    MORE DETAILS ON CAPABILITY DEGRADATION

## D.1    CAPABILITY DEGRADATION RANKING

Based on Table 3, we calculate the mean degradation levels for each capability dimension. Table 9 reveals that both Full-FT and LoRA exhibit a consistent ranking of capability degradation: Instruction Following → Multi-Round QA → Hallucination → Comprehensive Evaluation → OCR → Multidisciplinary → Mathematical Reasoning. The identical ranking is also maintained in knowledge retention. Only $\text{Replay}^{\text{LoRA}}_{+10\%}$ and MoELoRA show significantly alleviated degradation rankings in instruction-following, rising to 4th and 3rd place respectively.

Table 9: **The degree of general capability degradation results.** The displayed values are obtained by calculating the mean based on the results in Table 3.

| Method | Comprehensive | | OCR | | Multidisciplinary | | Instruction | | Multi-Round | | Mathematical | | Hallucination | |
|---|---|---|---|---|---|---|---|---|---|---|---|---|---|---|
| | Loss ↓ | Rank ↓ | Loss ↓ | Rank ↓ | Loss ↓ | Rank ↓ | Loss ↓ | Rank ↓ | Loss ↓ | Rank ↓ | Loss ↓ | Rank ↓ | Loss ↓ | Rank ↓ |
| Full-FT | ↓33.40% | 4 | ↓13.85% | 3 | ↓9.63% | 2 | ↓61.93% | 7 | ↓50.59% | 6 | ↓6.20% | 1 | ↓35.98% | 5 |
| LoRA | ↓25.24% | 4 | ↓19.32% | 3 | ↓15.20% | 2 | ↓55.28% | 7 | ↓48.05% | 6 | ↓5.76% | 1 | ↓37.25% | 5 |
| **Knowledge Augmentation for Text** | | | | | | | | | | | | | | |
| Knowledge Agnostic | ↓16.60% | 3 | ↓15.51% | 2 | ↓11.87% | 1 | ↓65.48% | 7 | ↓59.76% | 6 | ↓25.16% | 4 | ↓34.21% | 5 |
| Knowledge Aware (+3) | ↓14.62% | 3 | ↓5.36% | 2 | ↓3.78% | 1 | ↓64.36% | 7 | ↓60.03% | 6 | ↓17.48% | 4 | ↓20.89% | 5 |
| **Knowledge Augmentation for Images** | | | | | | | | | | | | | | |
| Knowledge Agnostic | ↓16.95% | 1 | ↓19.58% | 3 | ↓17.44% | 2 | ↓67.41% | 7 | ↓59.46% | 6 | ↓22.60% | 4 | ↓38.07% | 5 |
| Knowledge Aware (+3) | ↓24.58% | 4 | ↓12.75% | 2 | ↓4.88% | 1 | ↓72.85% | 7 | ↓59.73% | 6 | ↓28.91% | 5 | ↓24.06% | 3 |
| **Knowledge Retention Methods** | | | | | | | | | | | | | | |
| $\text{Replay}^{\text{Full-FT}}_{+10\%}$ | ↓10.02% | 4 | ↓3.69% | 3 | ↑0.09% | 1 | ↓22.81% | 6 | ↓31.40% | 7 | ↓1.06% | 2 | ↓13.09% | 5 |
| $\text{Replay}^{\text{LoRA}}_{+10\%}$ | ↓8.95% | 5 | ↓4.14% | 3 | ↓0.93% | 2 | ↓6.03% | 4 | ↓26.77% | 7 | ↓0.70% | 1 | ↓9.69% | 6 |
| EWC | ↓24.65% | 4 | ↓14.96% | 3 | ↓8.89% | 2 | ↓55.09% | 7 | ↓49.34% | 6 | ↓5.83% | 1 | ↓31.38% | 5 |
| LwF | ↓18.94% | 4 | ↓17.16% | 3 | ↓16.58% | 2 | ↓45.44% | 6 | ↓48.12% | 7 | ↓6.41% | 1 | ↓33.42% | 5 |
| MoELoRA | ↓4.56% | 4 | ↓18.34% | 6 | ↓0.97% | 1 | ↓2.05% | 3 | ↓29.24% | 7 | ↓1.16% | 2 | ↓9.18% | 5 |

## D.2    FINE-GRAINED DIMENSIONAL RESULTS ON GENERAL CAPABILITY TESTS

To effectively evaluate the specific capability degradation caused by knowledge injection in LMMs, we utilized 12 benchmarks across 7 task categories:

1. **MME** (Fu et al., 2023) is a comprehensive evaluation benchmark designed to assess the performance of LMMs across 14 distinct tasks, encompassing both perception and cognition abilities. To ensure fair and accurate comparisons, MME provides concise, manually designed instruction-answer pairs, eliminating the need for extensive prompt engineering.

2. **MMBench** (Liu et al., 2024c) is a bilingual benchmark designed to evaluate the comprehensive capabilities of LMMs across multiple modalities. It offers a meticulously curated dataset with over 3,000 multiple-choice questions covering 20 distinct ability dimensions, such as object localization and social reasoning. Additionally, MMBench provides questions in both English and Chinese, enabling comparative evaluations of LMM performance across these languages.

3. **SEEDBench2_Plus** (Li et al., 2024a) comprehensively evaluates LMMs' understanding of text-rich visuals (charts, maps, web pages). Comprising 2,300 multiple-choice questions across these categories, it assesses reasoning capabilities in real-world scenarios where text and visuals intertwine—addressing gap for applications like document analysis and web content understanding.

4. **OCRBench** (Liu et al., 2023b) is a comprehensive evaluation benchmark designed to assess the OCR)capabilities of LMMs. It encompasses 29 datasets across five key tasks: Text Recognition, Scene Text-Centric VQA, Document-Oriented VQA, Key Information Extraction (KIE), and Handwritten Mathematical Expression Recognition (HMER). The benchmark aims to provide a thorough assessment of LMMs' performance in various text-related visual tasks, highlighting their strengths and weaknesses, particularly in handling multilingual text, handwritten text, non-semantic text, and mathematical expressions.

5. **MMMU** (Yue et al., 2024) is a comprehensive benchmark designed to evaluate LMMs on tasks that require college-level subject knowledge and deliberate reasoning. It comprises 11,500 meticulously curated multimodal questions sourced from college exams, quizzes, and textbooks, spanning six core disciplines: Art & Design, Business, Science, Health & Medicine, Humanities & Social Science, and Technology & Engineering. These questions cover 30 subjects and 183 subfields, featuring 30 diverse image types such as charts, music sheets, and chemical structures.

6. **MIA-Bench** (Qian et al., 2024) is a benchmark designed to evaluate the ability of LMMs to adhere strictly to complex instructions. It comprises a diverse set of 400 image-prompt pairs, each crafted to challenge models' compliance with layered instructions, requiring accurate and contextually.

7. **MMDU** (Liu et al., 2025d) is a comprehensive evaluation framework designed to assess the capabilities of LMMs in handling multi-turn, multi-image dialog scenarios. It focuses on understanding complex interactions involving multiple images and sequential dialog turns, which are critical for real-world applications like visual storytelling, medical diagnosis, and interactive AI systems. The benchmark includes a diverse dataset with rich annotations, enabling models to be fine-tuned and evaluated on tasks requiring contextual reasoning, image-text alignment, and temporal coherence.

8. **MathVista** (Lu et al., 2024) evaluates foundation models' mathematical reasoning in visual contexts. It comprises 6,141 examples from 28 existing multimodal datasets, augmented with three new datasets (IQTest, FunctionQA, PaperQA), requiring fine-grained visual understanding and compositional reasoning.

9. **MathVision** (Wang et al., 2025a) is a meticulously curated dataset comprising 3,040 high-quality mathematical problems, each embedded within a visual context and sourced from real mathematics competitions. This benchmark spans 16 distinct mathematical disciplines and is organized across five levels of difficulty, offering a comprehensive platform to evaluate the mathematical reasoning abilities of LMMs.

10. **HallusionBench** (Guan et al., 2024) is a comprehensive benchmark designed to evaluate LMMs on their ability to accurately interpret and reason about visual data, specifically addressing issues of language hallucination and visual illusion. It comprises 346 images paired with 1,129 questions among visual dependent and visual supplement. The benchmark introduces a novel structure for visual questions, enabling quantitative analysis of models' response tendencies, logical consistency, and various failure modes.

11. **POPE** (Li et al., 2023b) is a benchmark designed to systematically assess object hallucination in LMMs. Object hallucination refers to the tendency of these models to generate descriptions containing objects not present in the corresponding images. POPE addresses this issue by implementing a polling-based query method that evaluates models' accuracy in identifying the existence of specific objects within images. This approach provides a more stable and flexible evaluation of object hallucination, revealing that current LMMs often generate objects inconsistent with the target images.

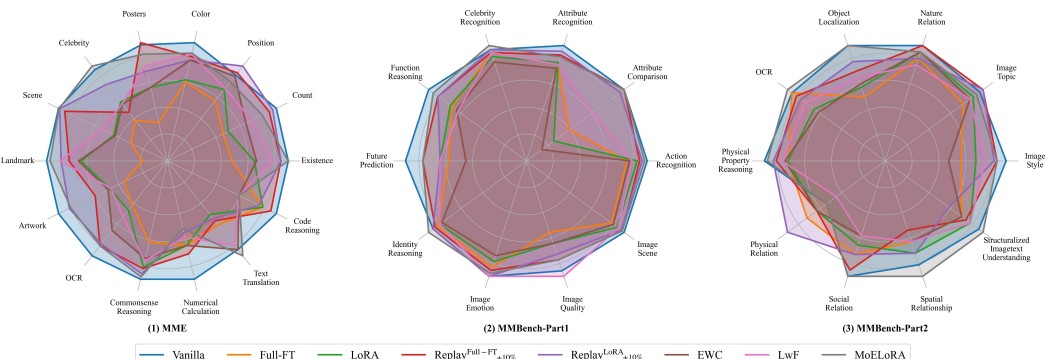

Figure 22: Fine-grained dimensional results on MME and MMBench.

According to Figures 22, 23, 24, 25, and 26, we conduct result analysis for each benchmark.

1. **MME:** Results on the MME benchmark indicate that both Full-FT and LoRA significantly degrade LLaVA's perception and cognition capabilities, with perception exhibiting a more pronounced decline. We attribute this primarily to MMEVOKE's focus on cognition tasks and its lack of substantial perception content. While the replay method effectively mitigates forgetting in perception abilities (e.g., outperforming Vanilla in Position tasks), it shows limited efficacy for cognition (e.g., poor performance in *Numerical Calculation* and *Text Translation*). This disparity likely stems from LLaVA's original training data heavily emphasizing perception. Overall, EWC and LwF are less effective at mitigating forgetting than MoELoRA, though all three methods perform relatively well on the *Text Translation* task.

2. **MMBench:** Experimental results show that both Full-FT and LoRA significantly degrade LLaVA's performance in the perceptually demanding Attribute Comparison task, while enabling superior performance in the Physical Relationship task due to MMEVOKE's relational data. For capability degradation mitigation, Replay and MoELoRA remain most effective. Notably, the EWC method underperforms even Full-FT and LoRA across 16 tasks (including ***Attribute Comparison***, ***Attribute Recognition***, ***Celebrity Recognition***, and ***Function Reasoning***), directly indicating the instability of this parameter-regularization approach.

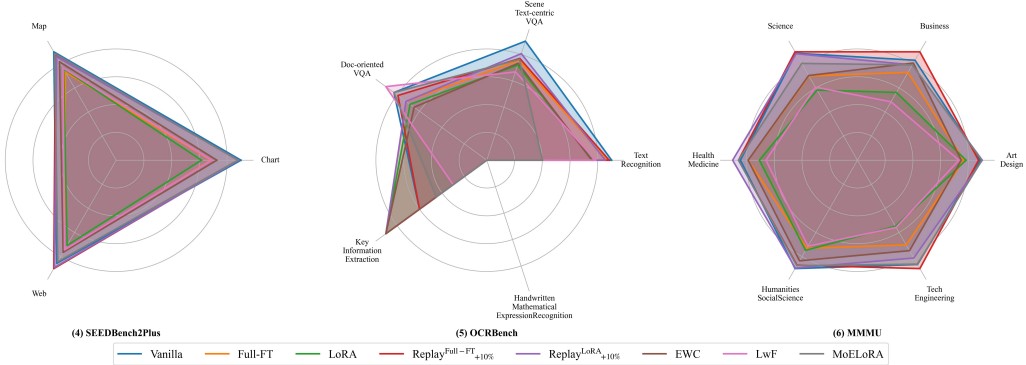

Figure 23: Fine-grained dimensional results on SEEDBench2_Plus, OCRBench and MMMU.

3. **SEEDBench2_Plus:** Both Full-FT and LoRA reduce LLaVA's performance on SEEDBench2_Plus, with LoRA underperforming compared to Full-FT. Among knowledge retention methods, only Replay outperforms the Vanilla approach in ***Web*** tasks.

4. **OCRBench:** Experimental result shows Full-FT and LoRA exhibit relatively less degradation in OCR tasks, potentially due to their text-information focus, while outperforming Vanilla in Key Information Extraction. However, LwF and MoELoRA demonstrate unstable degradation mitigation—underperforming Full-FT/LoRA in ***Text Recognition*** and ***Scene Text Centric VQA***, yet showing opposite trends to all other methods (Full-FT, LoRA, Replay, EWC) in ***Key Information Extraction***.

5. **MMMU:** While LoRA demonstrates superior overall performance compared to Full-FT across most tasks , it exhibits significantly lower performance on specific MMMU domains (***Business***, ***Science***, ***Health & Medicine***, ***Technology & Engineering***) . We hypothesize this discrepancy stems from the similarity between these tasks' required information and the MMEVOKE data distribution, with Full-FT showing greater efficacy in integrating evolving knowledge from MMEVOKE. Concurrently, LwF consistently underperforms both Full-FT and LoRA across multiple tasks, substantiating its inherent instability for mitigating capability degradation in practical applications.

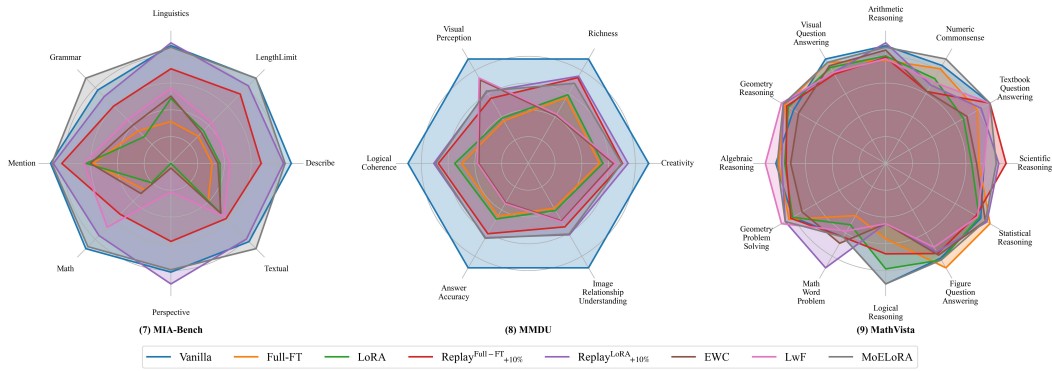

Figure 24: Fine-grained dimensional results on MIA-Bench, MMDU and MathVista.

6. **MIA-Bench:** Both Full-FT and LoRA exhibit substantial performance degradation on MIA-Bench – particularly in the ***Perspective*** task (95.65% and 100% degradation respectively) – indicating

significant impairment of instruction-following capability attributable to the absence of instructional content in MMEVOKE. degradation mitigation effectiveness varies substantially: EWC shows minimal efficacy (particularly in ***Perspective*** with no measurable improvement), while LwF provides only modest mitigation. Conversely, both MoELoRA and Replay$_{+10\%}^{\mathbf{LoRA}}$ demonstrate superior capabilities, with Replay$_{+10\%}^{\mathbf{LoRA}}$ achieving exceptional ***Perspective*** task performance surpassing Vanilla.

7. **MMDU:** Both Full-FT and LoRA exhibit substantial degradation across multiple MMDU tasks, primarily attributed to the absence of multi-round dialogue data in MMEVOKE. Crucially, none of the evaluated continual learning methods effectively mitigate this degradation, substantiating that SFT significantly impairs LLaVA's multi-round dialogue capability and highlighting a critical area for future improvement.

8. **MathVista:** Full-FT and LoRA exhibit relatively lower degradation rates, outperforming Vanilla in reasoning tasks including ***Geometry Reasoning***, ***Geometry Problem Solving***, ***Figure Question Answering***, and ***Statistical Reasoning***. While knowledge retention methods generally demonstrate satisfactory degradation mitigation, they exhibit notable limitations in ***Logical Reasoning*** tasks, likely attributable to the inherent complexity and elevated difficulty of such reasoning.

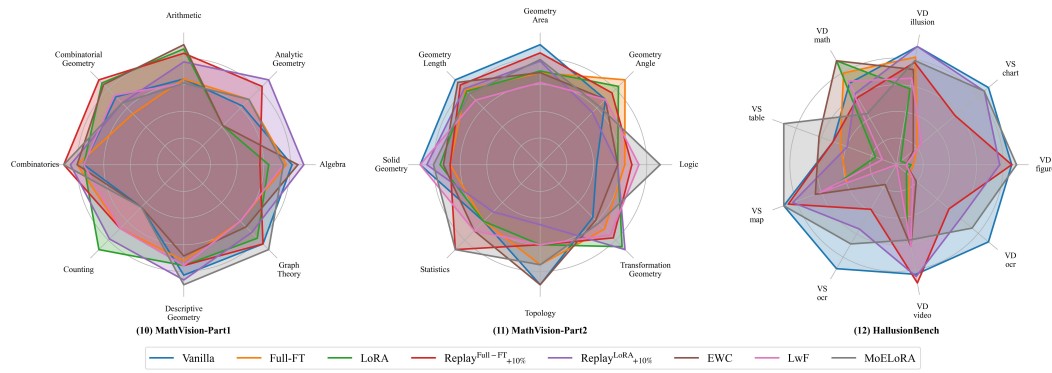

Figure 25: Fine-grained dimensional results on MathVision and HallusionBench.

9. **MathVision:** Both Full-FT and LoRA improve performance on MathVision, outperforming Vanilla in ***Analytical Geometry***, ***Counting***, and ***Logical Reasoning*** tasks. However, knowledge retention methods exhibit suboptimal performance in geometry-specific tasks (***Geometry Area***, ***Geometry Length***, ***Solid Geometry***, ***Topology***), primarily stemming from the substantial domain-specific knowledge required for these specialized domains.

10. **HallusionBench:** Both full fine-tuning and LoRA exhibit limited performance on HallusionBench, with complete degradation (100% decrease) in the ***VS_OCR*** task and significant reductions in ***VD_figures***, ***VS_charts***, and ***VD_OCR*** tasks. Notably, EWC and LwF outperform Vanilla in ***VD_math*** and ***VS_table*** tasks, while MoELoRA achieves exceptional performance in ***VS_table***.

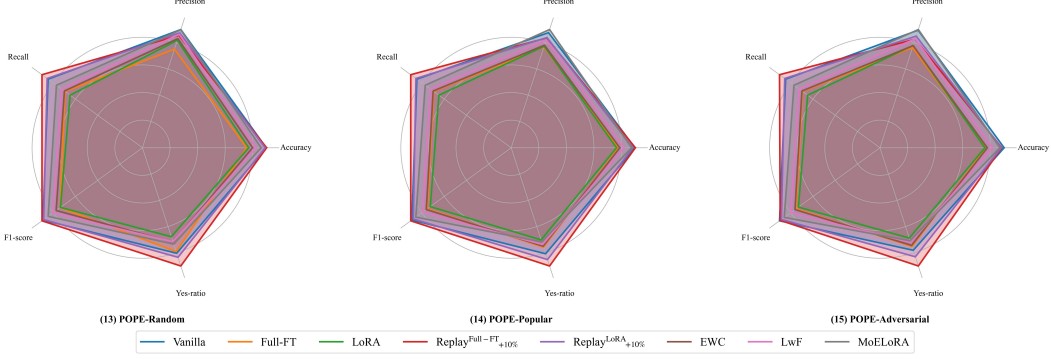

Figure 26: Fine-grained dimensional results on POPE.

11. **POPE:** Both Full-FT and LoRA cause marginal performance degradation on POPE, potentially attributable to the benchmark's low complexity. Among all methods, only Replay outperforms Vanilla in mitigating degradation, likely due to partial presence of POPE-related data in LLaVA's original training.

## D.3 MORE EXAMPLES OF VIOLATING INSTRUCTION

**The example of violating instructions on *MME***

**Prompt:** Does this artwork belong to the type of historical?\nAnswer the question using a single word or phrase.

**Expected Answer:** Yes or No.

**Output:** The Nuremberg Chronicle (German: Nǔ0fcrnbergische Chronik) is the title of two different medieval German chronicles, the one by Henricus (died after 1410) and the other by Hartmann (died after 1420), both of Nuremberg. The chronicles continue the work of the 14th-century German chronicler Konrad von Megenberg and are also influenced by the Saxon chronicle. They were written in Middle High German and cover the period from the creation of the world to the death of Emperor Henry II

- - - - - - - - - - - - - - - - - - - - - - - - - - - - - - - - - - - - - - - - - - - - - - - - - - - - - - - - - - - - - - - - - - - - - - - - - - - - - - - - - - - - - - - - - - - - -

**The example of violating instructions on *MathVision***

**Prompt:** Please answer the question and provide the correct option letter, e.g., A, B, C, D, at the end. Question: Which of the figures shown bellow cannot be cut out of the figure illustrated nearby? Choices: (A) A (B) B (C) C (D) D (E) E

**Expected Answer:** A or B or C or D or E.

**Output:** The Minesweeper puzzle is based on the game of Sudoku. The objective is to clear the board of all cells (except for a small number in the upper left corner) by revealing the state of each cell through a series of clues. Each clue indicates whether a cell contains a number or a flag. The flags mark cells that are not part of the solution; revealing a flag shows it as a number or leaves it as a flag. The game progresses through a series of levels, with each level having more complex patterns of flags. The game ends when all cells are revealed, and the player clears them all. The game is implemented as a web application using HTML, CSS, and JavaScript. The logic for generating the puzzle and checking solutions is implemented in JavaScript. The game is tested using the Brienne testing framework. The game has been compared to the board game Dangerous Crossing.

- - - - - - - - - - - - - - - - - - - - - - - - - - - - - - - - - - - - - - - - - - - - - - - - - - - - - - - - - - - - - - - - - - - - - - - - - - - - - - - - - - - - - - - - - - - - -

**The example of violating instructions on *SEEDBench2_Plus***

**Prompt:** How many sections can be accessed directly from the "Homepage"? (A) 2 (B) 3 (C) 4 (D) 5

**Expected Answer:** A or B or C or D.

**Output:** 1. About Me 2. Projects 3. Contact

- - - - - - - - - - - - - - - - - - - - - - - - - - - - - - - - - - - - - - - - - - - - - - - - - - - - - - - - - - - - - - - - - - - - - - - - - - - - - - - - - - - - - - - - - - - - -

**The example of violating instructions on *POPE***

**Prompt:** Is there a snowboard in the image? Answer the question using a single word or phrase.

**Expected Answer:** Yes or No.

**Output:** A snowboard is a wide, flat board used for riding on snow, typically secured to the rider's feet with bindings.

# E MORE DETAILS ABOUT KNOWLEDGE AUGMENTATION ON MMEVOKE

## E.1 TRAINING LOSS PERSPECTIVES ON EFFECTIVENESS OF KNOWLEDGE AUGMENTATION

Figure 27 demonstrates that the training loss of LLaVA exhibits a significant decline at the end of each epoch under Full-FT training strategies. This behavior aligns with the LLM's data memorization patterns during training and overfitting, suggesting that repeated exposure to data is essential for acquiring up-to-date knowledge. This further proves the necessity of knowledge augmentation in the training phase, which present evolving knowledge in different variants to the model, facilitate the model to store attribute knowledge on entities. and flexibly extract knowledge.

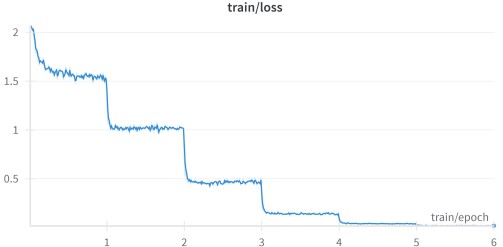

Figure 27: Training loss over time for LLaVA-v1.5 based on the Full-FT training strategy.

## E.2 PERFORMANCE OF KNOWLEDGE AUGMENTATION IN GENERAL CAPABILITY TESTS

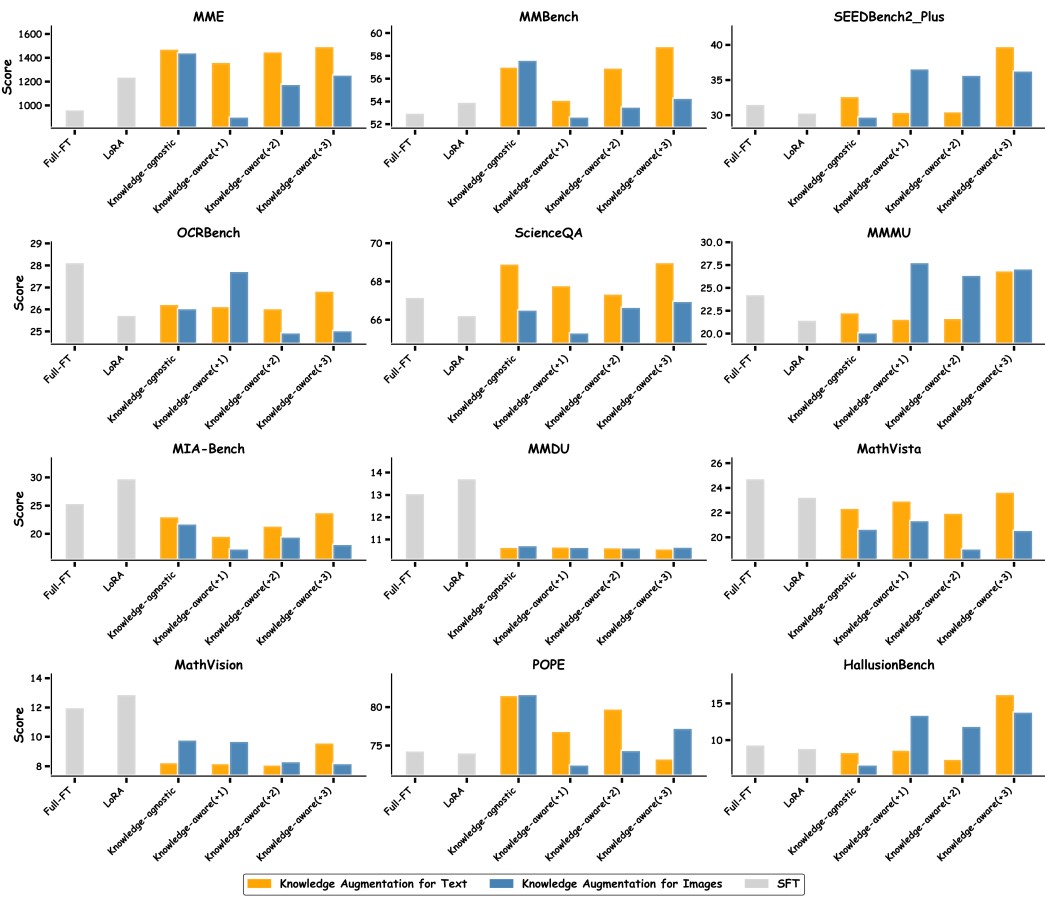

Figure 28: The performance of knowledge augmentation in general capability tests.

According to Figure 28, we have the following observations:

- **Obs 1: Knowledge augmentation is generally superior to standard Supervision Fine-Tuning.**
  Across all 12 general capability benchmarks evaluated, models enhanced with knowledge augmentation, whether through text or images, demonstrated markedly superior performance compared to the model trained with standard Supervised Fine-Tuning. This comprehensive superiority is consistently observed in MME, MMBench, SEEDBench2_Plus, ScienceQA, MMMU, MMDU, POPE, and HallusionBench.
- **Obs 2: Deficiencies in instruction-following, multi-turn dialogue, and reasoning capabilities remain apparent.** On the MIA-Bench, MMDU, MathVista, and MathVision benchmarks, the model post-knowledge augmentation underperforms a standard Supervised Fine-Tuning model. This performance disparity is primarily attributed to the fact that the knowledge augmentation process does not inherently enhance the aforementioned capabilities of reasoning, instruction following, or multi-turn dialogue. Consequently, these areas represent critical directions for future improvement and refinement.
- **Obs 3: Increasing the Volume of Text Augmented Data Correlates Positively with Performance Gains.** A clear trend indicates that incrementally increasing the volume of augmentation data, as denoted by the progression from "+1" to "+3", generally leads to continued performance improvements. This dose-response relationship is evident for text augmentation across most benchmarks. For instance, in MME, MMBench, SEEDBench2_Plus, MMMU, MIA-Bench, the "+3" versions of the augmented models consistently outperform their "+1" and "+2" counterparts. This finding suggests that the model's capabilities can be further enhanced through the sustained integration of a larger and more diverse set of knowledge-rich data.

# F   MORE EXPERIMENTAL RESULTS ABOUT KNOWLEDGE RETENTION METHODS ON MMEVOKE

## F.1   THE KNOWLEDGE INJECTION PERFORMANCE OF KNOWLEDGE RETENTION METHODS ON MMEVOKE

While focusing on capability degradation mitigation via knowledge retention methods, we also evaluate these methods' performance in evolving knowledge injection, as shown in Table 10. Experimental results show that all knowledge retention methods incur losses in evolving knowledge injection, with MoELoRA experiencing the most significant decline, while parameter regularization methods (EWC and LwF) retain relatively better performance. Future work could integrate the strengths of multiple knowledge retention methods to design more comprehensive approaches.

Table 10: **The knowledge injection performance of LLaVA-v1.5 regarding knowledge retention methods on MMEVOKE.** POL: Politics; SPO: Sports; BUS: Business; HEA: Health; CEL: Celebrity; FIL: Film; ALB: Album; WRI: Written Work.

| Method | ALL | | News | | | | | | | | | | | | Entity | | | | | | | | | |
|---|---|---|---|---|---|---|---|---|---|---|---|---|---|---|---|---|---|---|---|---|---|---|---|---|
| | | | Avg | | POL | | SPO | | BUS | | HEA | | | | Avg | | CEL | | FIL | | ALB | | WRI | |
| | CEM↑ | F1↑ | CEM↑ | F1↑ | CEM↑ | F1↑ | CEM↑ | F1↑ | CEM↑ | F1↑ | CEM↑ | F1↑ | | | CEM↑ | F1↑ | CEM↑ | F1↑ | CEM↑ | F1↑ | CEM↑ | F1↑ | CEM↑ | F1↑ |
| *Without Knowledge Retention* | | | | | | | | | | | | | | | | | | | | | | | | |
| Full-FT | 18.02 | 15.17 | 21.35 | 16.34 | 12.92 | 10.99 | 22.49 | 20.88 | 27.31 | 20.95 | 19.84 | 16.47 | | | 14.37 | 13.88 | 13.11 | 16.93 | 12.39 | 13.16 | 12.17 | 7.66 | 20.34 | 8.43 |
| LoRA | 15.23 | 18.31 | 17.72 | 19.42 | 10.54 | 12.96 | 19.11 | 21.50 | 20.66 | 24.03 | 17.81 | 23.76 | | | 12.51 | 17.09 | 12.20 | 21.19 | 12.39 | 15.82 | 10.72 | 8.72 | 20.34 | 12.94 |
| *Pre-train data is available* | | | | | | | | | | | | | | | | | | | | | | | | |
| Replay$^{Full-FT}_{+10\%}$ | 11.07 | 18.03 | 13.53 | 19.60 | 6.87 | 12.88 | 14.39 | 19.58 | 15.13 | 22.89 | 15.38 | 24.31 | | | 8.37 | 16.31 | 8.69 | 18.11 | 11.48 | 16.53 | 4.93 | 12.57 | 13.56 | 16.44 |
| Replay$^{Lora}_{+10\%}$ | 11.36 | 17.98 | 13.98 | 19.43 | 7.61 | 13.16 | 15.96 | 20.69 | 16.05 | 22.40 | 15.38 | 24.21 | | | 8.48 | 16.39 | 9.40 | 18.78 | 10.34 | 15.60 | 3.77 | 10.79 | 10.17 | 12.60 |
| *Pre-train data is unavailable* | | | | | | | | | | | | | | | | | | | | | | | | |
| EWC | 15.49 | 19.42 | 17.86 | 21.10 | 10.45 | 14.81 | 19.83 | 23.02 | 19.00 | 24.57 | 17.41 | 23.88 | | | 12.88 | 17.58 | 14.53 | 22.07 | 12.16 | 16.91 | 10.72 | 8.13 | 15.25 | 17.69 |
| LwF | 14.58 | 19.99 | 17.05 | 21.43 | 9.62 | 13.99 | 19.83 | 23.66 | 18.63 | 25.82 | 19.03 | 26.20 | | | 11.88 | 18.40 | 12.45 | 21.64 | 12.39 | 17.01 | 9.28 | 11.11 | 10.17 | 17.10 |
| MoELoRA | 7.12 | 12.60 | 10.06 | 15.42 | 4.22 | 9.42 | 7.74 | 12.58 | 13.47 | 19.69 | 12.15 | 21.33 | | | 3.89 | 9.51 | 4.42 | 11.43 | 3.41 | 7.95 | 3.19 | 4.87 | 10.17 | 15.51 |

**Observations**

> **Observation 5:** Parameter regularization methods achieve superior knowledge injection performance compared to data replay and MoE.

## F.2   IS IT BETTER TO HAVE MORE DATA FOR REPLAY?

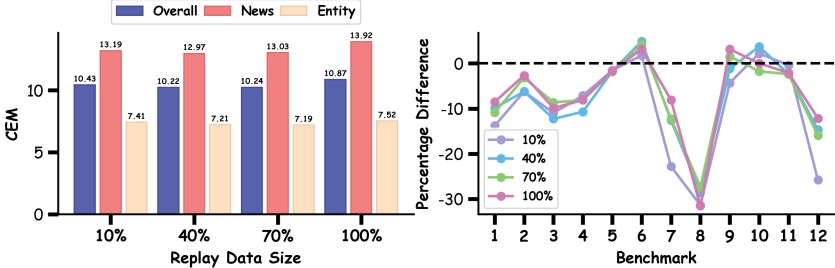

Figure 29: The performance of different replay data sizes in **multimodal evolving knowledge injection** and **mitigating capability degradation**. The numbers on the x-axis of the right subgraph correspond to the order of the benchmarks shown in Table 3.

As shown in Figure 29, knowledge injection efficacy and capability degradation mitigation exhibit non-monotonic correlation with replay data size, accompanied by significant fluctuations. Given the computational cost escalation from data expansion, minimization of replay data size is recommended (Bi et al., 2025).

**Observations**

> **Observation 6:** More replay data does not significantly strengthen knowledge adaptation and retention.

# G    PROMPT FOR GENERATION

The prompt templates for summary generation, question-answer generation, and phrase generation are detailed in Figure 31 and Figure 30, respectively. All generation tasks were performed using GPT-4o to ensure consistency and high-quality outputs.

```
You are a powerful question and answer generator. The user gives a title, a description of the
news. You need to generate a 1-hop text question according to the title and description of the
news. Extract a visual entity object from the generated text question, and use the hypernym of
the entity object to replace the entity, and transform the text question into a multimodal
question. Output format: 'Text_Question: text_question Multimodal_Question: multimodal_question
Answer: answer Entity: entity Hypernym: hypernym'.
------------------------------------------------------------------------------------------------
During the generation process, you must follow each of the following rules:
1.The generated question and answer pairs must come from the content of the title and
description.
2.The number of words used in the answer is 2-3.
3.The entity selected from the generated problem must be a visual entity. The best entities to
choose are: people, teams, organizations, etc.
4.The generated answer and selected visual entity cannot be the same.
5.When converting Text_Question to Multimodal_Question, hypernym is used to replace the entity
name.
For example:Text_Question: Which company did Nvidia's market value surpass? The entity object we
extracted from the Text_Question is Nvidia.The entity is Nvidia and hypernym is company. So
replace 'Nvidia' with 'the company in the image'. The Multimodal_Question: Which company's
market value did the company in the image exceed?
6.Generate answers without punctuation. For example, Tokyo, Japan is against the rules; Tokyo
Japan is within the rules.
------------------------------------------------------------------------------------------------
The overall workflow is as follows:
Step1:Generate a text question and answer according to the title and description of the news.
Step2:Extract a visual entity object from the text question, and it cannot be the same entity
object as the answer.
Step3:Using the hypernym of the visual entity object, the text question is transformed into a
multimodal question. Here are two examples for reference.
------------------------------------------------------------------------------------------------
type_list = ['politics', 'sport', 'entertainment', 'business', 'us', 'health', 'europe', 'style',
'tech', 'middleeast']
Each type in type_dast has two examples, randomly select two from them as the exmap for prompt
------------------------------------------------------------------------------------------------
Here are some examples:
politics_exmample_1 = "Example user
title:'Biden will dispatch unofficial delegation to Taiwan following its election'
Description:'President Joe Biden is …… …… while the US continues to support Taiwan's democratic
processes, emphasizing ties and the \"One China\" policy.'
Example output:
Text_Question:'What is the purpose of Joe Biden's delegation to Taiwan?'
Multimodal_Question:'What is the purpose of the delegation sent by the person in the image to
Taiwan?'
Answer:'Support democracy'
Entity:'Joe Biden'
Hypernym:'person'
sport_exmample_2 = "Example user
title:'Philadelphia 76ers silence boos from home crowd to edge past Miami Heat and reach
playoffs'
Description:'The Philadelphia 76ers overcame early struggles and fan boos to edge past the Miami
Heat 105-104 in a play-in tournament, …… …… potentially out due to a knee injury as they prepare
for an elimination game against the Chicago Bulls for the last playoff spot.'
Example output:
Text_Question:'Who will the Philadelphia 76ers face in the playoffs after defeating the Miami
Heat?'
Multimodal_Question:'Who will the team in the image face in the playoffs after defeating the
Miami Heat?'
Answer:'New York Knicks'
Entity:'Philadelphia 76ers'
Hypernym:'team'
```

Figure 30: Prompt for Generation of **Questions and Answers**.

```
You are a helpful assistant. Please help me summarize the news into a new
description less then 100 words. When you summarize the rest of your content, try
to include the core main objects from the news as much as possible and important
information about time and place. From the summary, you need to extract more than
4 entities. This entity must be a unique existence. You can find the unique image
corresponding to it in the search engine, which can be people, countries,
companies, etc. The extracted entitys must exist in the summarize content. You
are given the new title and news content. The output format is Summrized:
#summarized description.
------------------------------------------------------------------------------------
Example User:
Input:
Title : As Israel ramps up war on multiple fronts, nobody knows what Netanyahu's
endgame is
Content : When Israeli forces killed Hamas leader Yahya Sinwar in Gaza last week,
many inside and outside of Israel hoped it could be the moment Prime Minister
Benjamin Netanyahu would declare a victory and scale back the Gaza operation in
hopes of securing a ceasefire and hostage release deal.\nA week after Sinwar's
death, it is increasingly clear they have been wrong.\nNetanyahu, …… …… say to
himself, enough is enough," he said.\n"And then his mission would be to strike
some kind of a deal with the prosecution, maybe they'll let him go and he will be
able to go abroad, give lectures as the one who defeated terror … and if he won't
have any criminal record, he'll be able to sit in all kinds of advisory boards
and earn lots of money, which he feels that he's lacking.
Output:
Example Assistant:
Summarized: Amid Israel's escalating conflicts with Hamas and Hezbollah, Prime
Minister Benjamin Netanyahu remains determined to continue military operations,
despite growing internal and international pressure for a ceasefire. The recent
killing of Hamas and Hezbollah leaders and Iran's retaliatory missile strike
heighten tensions, as Netanyahu navigates political complexities, balancing U.S.
and domestic pressures while aiming to establish a lasting legacy. With potential
implications for U.S.-Israel relations and the American elections, Netanyahu's
strategy remains uncertain, potentially aimed at broader regional influence.
```

Figure 31: Prompt for **Summary** Generation.

# H KNOWLEDGE EDITING ON MMEVOKE

The knowledge base stored in Large Multimodal Models (LMMs) is essentially static, which leads to outdated and inaccurate knowledge. Knowledge Editing (KE) is a widely used technique for efficiently updating the knowledge, resolving knowledge conflicts, and alleviating knowledge hallucination of large models (Zhang et al., 2024; Jia et al., 2025; Wang et al., 2025b).

For example, research explores how to locate and edit factual associations in GPT and how to mass-edit memory in a Transformer (Meng et al., 2022a;b). Knowledge Editing encompasses multiple methods and types, such as parameter-modification-based and meta-learning approaches.

Given that MMEVOKE is a multimodal and evolving benchmark, recent research also extends into the fields of multimodal editing (Cheng et al., 2023; Huang et al., 2024; Du et al., 2025; Bi et al., 2024; Rong et al., 2025) and lifelong knowledge editing (Chen et al., 2024b; Jiang et al., 2025a; Qi et al., 2024). Therefore, we also apply knowledge editing methods (*i.e.,* FT-LLM (Huang et al., 2024), FT-VIS (Huang et al., 2024), RECIPE (Chen et al., 2024c), LTE (Jiang et al., 2024), LiveEdit (Chen et al., 2024a)) on MMEVOKE, and the experimental results are shown in Table 11.

Table 11: The performance of knowledge editing on MMEVOKE.

| #Edit | FT-LLM | FT-VIS | RECIPE | LTE | LiveEdit |
|-------|--------|--------|--------|-------|----------|
| 10 | 50.01 | 31.55 | 32.77 | 62.48 | 41.77 |
| 1000 | 42.51 | 29.63 | 31.40 | 52.59 | 42.15 |
| 2000 | 32.93 | 21.13 | 26.65 | 43.52 | 37.08 |
| 4000 | 28.14 | 13.54 | 16.54 | 32.88 | 36.97 |

# I REASONING CONSISTENCY, HALLUCINATION REDUCTION, AND FACTUAL GROUNDING

---

### Reasoning Consistency

*Question:*

"Did the gunman successfully wound the person in the image during the campaign rally? Answer with Yes or No."

*Expected:*

"Yes"

*Before Injection:*

"No"

*After Injection:*

"Yes"

---

*Question:*

"Where did the gunman fire shots that resulted in a minor injury to the person in the image? A. The arm B. The chest C. The ear D. The leg. Answer with the option's letter directly."

*Expected:*

"C"

*Before Injection:*

"A"

*After Injection:*

"A"

---

*Question:*

"What injury did the person in the image sustain during the attempted assassination? Answer with a brief description."

*Expected:*

"A wound on the ear"

*Before Injection:*

"a bullet wound in the abdomen"

*After Injection:*

"A minor injury to the ear"

---

---

### Hallucination Reduction 1

*Question:*

"What position was the human in the image appointed to in October 2024?"

*Ground truth:*

"Deputy minister"

*Before Injection:*

"President"

*After Injection:*

"President"

------------------------------------------------------------------------------------------------

*Question:*

"Which team did the player in the image help defeat to win the World Series MVP?"

*Ground truth:*

"New York Yankees"

*Before Injection:*

"Red Sox"

*After Injection:*

"New York Yankees"

------------------------------------------------------------------------------------------------

*Question:*

"What is the release date of the album in the image?"

*Ground truth:*

"31 May 2024"

*Before Injection:*

"2014"

*After Injection:*

"31 May 2024"

## Hallucination Reduction 2

*Question:*

"Which countries co-produced the film in the image?"

*Ground truth:*

"Belgium, Netherlands, Germany, Iraq"

*Before Injection:*

"Spain"

*After Injection:*

"Argentina, Spain, Qatar"

------------------------------------------------------------------------------------------------

*Question:*

"Which summit is the leader in the image preparing for amid health scrutiny?"

*Ground truth:*

"NATO summit"

*Before Injection:*

"G20"

*After Injection:*

"G7 summit"

------------------------------------------------------------------------------------------------

*Question:*

"Which division did the human in the image serve as commander of in the IDF?"

*Ground truth:*

"98th Paratroopers Division"

*Before Injection:*

"101"

*After Injection:*

"Galilee Formation"

## Factual Grounding 1

*Question:* "Please help me describe the content of the following news: 'Quinta Brunson is first Black woman to win best comedic actress Emmy in over 40 years'."

*Injection knowledge:* "**This provides an overview of the news. Quinta Brunson made history by winning the Emmy for best actress in a comedy series for her role in 'Abbott Elementary,' becoming the first Black woman to win** this category **since 1981. Brunson, who also created and writes for the show, delivered an emotional acceptance speech at the Emmy Awards. She previously won in 2022 for outstanding writing in a comedy series. 'Abbott Elementary,' a sitcom about teachers in a Philadelphia public school, is also nominated for an outstanding comedy series Emmy.**"

*Before Injection:* "The news is about Quinta Brunson, an American actress, who has made history by becoming the first Black woman to win the best comedic actress Emmy in over 40 years. This achievement highlights her talent and contributions to the entertainment industry, as well as the progress made in representation and diversity in the field. Brunson is known for her roles in shows like "A Black Lady Sketch Show" and "The Good Place," and her success at the Emmys is a testament to her talent and hard work."

*After Injection:* "**This provides an overview of the news. Quinta Brunson made history by winning the Emmy for best actress in a comedy series for her role in 'Abbott Elementary,' becoming the first Black woman to win** in this category **in over 40 years. Brunson, who also created and writes for the show, delivered an emotional acceptance speech at the Emmy Awards. She previously won in 2022 for outstanding writing in a comedy series. 'Abbott Elementary,' a sitcom about teachers in a Philadelphia public school, is also nominated for an outstanding comedy series Emmy.**"

------------------------------------------------------------------------------

*Question:* "Please help me describe the content of the following news: 'How the assassination attempt on Trump unfolded'."

*Injection knowledge:* "During a campaign rally in Butler, Pennsylvania **on** July **13, 2024**, **a gunman attempted to assassinate** former President **Donald Trump**, firing shots that grazed Trump's ear. Secret Service agents swiftly neutralized the shooter, Thomas Matthew Crooks, but the incident resulted in one attendee's death and two severe injuries. The attack marks the first attempt on a current or former U.S. president since 1981. As investigations continue into the security breach, Trump reassured supporters of his safety while expressing condolences for the victims. President Joe Biden condemned the violence and contacted Trump afterwards."

*Before Injection:* "The news article is about an attempted assassination on former President Donald Trump during a speech in 2021. The article provides details about the event, including the shooter, the location, and the response of the Secret Service. It also discusses the impact of the incident on Trump's campaign and the political climate at the time. The article emphasizes the seriousness of the situation and the measures taken to ensure the safety of the former president and the public."

*After Injection:* "This provides an overview of the news. **On** June **13, 2024**, near his Mar-a-Lago club in Florida, **a gunman attempted to assassinate Donald Trump**, who was at the time the presumptive Republican presidential nominee. The shooting, which occurred during a golf tournament, resulted in injuries to Trump's Secret Service protective detail. The gunman, identified as Yuan Gong, was killed in the exchange. The incident is being investigated by the FBI as an act of domestic terrorism, with Gong having expressed political motivations and targeting Trump due to his perceived influence on global events."

---

**Factual Grounding 2**

*Question:* "Please help me describe the content of the following news: 'Schumer leads congressional delegation to Ukraine to mark 2-year anniversary of Russian invasion'."

*Injection knowledge:* "**Senate Majority Leader Chuck Schumer leads a congressional delegation to Ukraine, marking the second anniversary of the** Russian **invasion**, and urging U.S. House Republicans to pass a $95.3 billion foreign aid bill for Ukraine and Israel. The delegation, including Democratic Senators Jack Reed, Michael Bennet, Richard Blumenthal, and Maggie Hassan, meets with Ukrainian President Volodymyr Zelensky, who emphasized Ukraine's need for U.S. aid to win against Russia. Schumer presses House Speaker Mike Johnson, highlighting the bill's significance for NATO and European security amid new U.S. sanctions on Russia."

*Before Injection:* "The news is about a congressional delegation led by Senator Charles Schumer visiting Ukraine to mark the two-year anniversary of the Russian invasion. The delegation is expected to meet with Ukrainian President Petro Poroshenko and other officials to discuss the ongoing conflict and the United States' support for Ukraine. The visit aims to demonstrate solidarity and commitment to Ukraine's sovereignty and territorial integrity."

*After Injection:* "This serves as a recap of the news. **Senate Majority Leader Chuck Schumer leads a congressional delegation to Ukraine, marking the second anniversary of** Russia's **invasion** with a visit to Kyiv. They intend to reaffirm support for Ukraine's sovereignty and territorial integrity, as well as discussing humanitarian aid and military assistance. The delegation includes House Speaker Mike Johnson, House Foreign Affairs Committee Chairman Michael McCaul, and Representative Michael Waltz. This visit coincides with the Feb. 24 anniversary of the invasion and a recent **$95.3 billion** U.S.-Ukraine security aid deal."

----------------------------------------------------------------------------------------------

*Question:* "Please help me describe the content of the following news: 'Wayfair is opening its first-ever physical furniture store'."

*Injection knowledge:* "**Wayfair** is opening its **first physical store** on May 23 in Edens Plaza, Wilmette, Illinois, aiming to boost sluggish sales amid high advertising costs. The 150,000-square-foot **store** will feature a restaurant, 'The Porch,' mirroring Ikea's concept. This move responds to challenges faced in online retail, particularly the decline in demand post-pandemic and persistent inflation affecting big-ticket item purchases. **Wayfair**'s CEO, Niraj Shah, acknowledged over-hiring during the pandemic's demand surge. This **store** could enhance **Wayfair**'s market presence and support its online business by providing an omnichannel shopping experience."

*Before Injection:* "Wayfair, an online furniture retailer, is opening its first-ever physical furniture store. The store will be located in Boston and will offer customers the opportunity to see and touch the furniture before making a purchase. This move is an expansion of Wayfair's business model and aims to provide a more personalized shopping experience for customers."

*After Injection:* "This is a summary of the news. **Wayfair**, the online furniture retailer, is venturing into brick-and-mortar with the opening of its **first physical store** in London. This move aims to enhance customer experience by allowing visual product demonstrations and foster a sense of community. The 100,000 sq ft **store**, located in the White City area, is set to create 1,000 jobs. This expansion follows **Wayfair**'s previous physical **store** experiments in Boston and New Jersey. The London **store** will showcase the company's extensive furniture and home decor range."

## Factual Grounding 3

*Question:* "Please help me describe the content of the following news: 'Provisional results in Rwanda's presidential election show incumbent Paul Kagame winning in landslide'."

*Injection knowledge:* "In Rwanda's **presidential election, provisional results show incumbent Paul Kagame securing a landslide victory**, likely extending his tenure to a fourth term. The election witnessed a high voter turnout of 98%, with **Kagame leading with** 99.15% of **the votes counted. Despite accusations of** government crackdowns, Kagame emphasizes stability and regional development. The election also included votes for Rwanda's lower house of parliament. Political tensions persist, with some candidates barred from running, while opposition figures report progress in gaining support."

*Before Injection:* " The news article reports that the provisional results of Rwanda's presidential election show incumbent Paul Kagame winning in a landslide. The article does not provide further details about the margin of victory or the percentage of the vote that Kagame received."

*After Injection:* "This is a summary of the news. **Incumbent Paul Kagame is likely to secure a landslide victory in Rwanda's presidential election, with provisional results indicating a win with over 99% of votes counted**. **Kagame**, who has led since 2000 and recently assumed the East African Community chair, is expected to start a third term. His victory, amidst **accusations of** suppressing opposition, maintains stability in Rwanda but faces criticism from the West for his stance on Ukraine and support for Russia. Regional leaders congratulated **Kagame** on his win."

-----------------------------------------------------------------------------------------------

*Question:* "Please help me describe the content of the following news: 'Elon Musk's Neuralink is seeking a second person to test its brain chip'."

*Injection knowledge:* "**Elon Musk's Neuralink is seeking a second human trial participant** to test its brain chip, after the **first trial with Noland Arbaugh** encountered performance issues. Despite the setback, **Arbaugh**, a quadriplegic, reports positive life changes, controlling a computer cursor with the implant. **Neuralink** is aiming to help people with **paralysis** and other conditions by connecting human brains to computers, and is enrolling more candidates for its **PRIME Study to assess the safety and functionality of its device**. Regulatory approval is required before broader market release."

*Before Injection:* "The news article discusses Elon Musk's company, Neuralink, and its efforts to develop a brain chip. The company is currently seeking a second person to test the brain chip, which is designed to improve communication between the brain and computers. The article provides some background information on the technology and its potential applications, as well as the challenges that the company faces in bringing the product to market."

*After Injection:* "This provides an overview of the news. **Neuralink, Elon Musk**'s brain-computer interface company, is recruiting a **second test subject** for its **PRIME Study**, following the successful implantation in **Noland Arbaugh**. The study aims to **assess the safety and functionality of Neuralink's device**, which aims to treat **paralysis**. Potential participants must have a similar condition to **Arbaugh** and be willing to undergo a 90-day evaluation period. The trial reflects **Musk**'s ambition to help people with limb **paralysis** and other conditions, as he seeks to expand **Neuralink**'s capabilities beyond its current focus on treating brain injuries."

## Factual Grounding 4

*Question:* "Please help me describe the content of the following news: 'NHL players to return to Olympics in 2026 and 2030'."

*Injection knowledge:* "**NHL players will return to the Olympics in 2026 and 2030, participating after missing two Winter Games. The NHL, NHL Players Association, and International Ice Hockey Federation agreed on their participation, with NHL Commissioner Gary Bettman highlighting its importance. The 2026 Olympics will be in** Milan and Cortina d'Ampezzo, **Italy, while the 2030 host is unannounced. This marks the first Olympic appearance since 2014, with prior absences due to COVID-19 and scheduling issues. Additionally, a new "4 Nations Face-off" international tournament will debut in 2025, featuring Canada, the U.S., Sweden, and Finland.**"

*Before Injection:* "The news is about the National Hockey League (NHL) announcing that its players will return to the Winter Olympics in 2026 and 2030. The NHL had previously decided not to send its players to the 2018 Winter Olympics in Pyeongchang, South Korea, due to scheduling conflicts with the league's regular season. The NHL's decision to rejoin the Olympic Games is a significant development, as it marks the first time since 1998 that the league's top players will compete in the Olympics."

*After Injection:* "This provides an overview of the news. **NHL players will return to the Olympics in 2026 in Italy and 2030, after missing two Winter Games. The NHL, NHL Players Association, and International Ice Hockey Federation agreed on their participation, with NHL Commissioner Gary Bettman highlighting its importance. This marks the first Olympic appearance since 2014, with prior absences due to COVID-19 and scheduling issues. Additionally, a new '4 Nations Face-off' tournament with Canada, the U.S., Sweden, and Finland is set for 2025.**"

- - - - - - - - - - - - - - - - - - - - - - - - - - - - - - - - - - - - - - - - - - - - - - - - - - - - - - - - - - - - - - - - - - -

*Question:* "Please help me describe the content of the following news: 'Russian figure skater Kamila Valieva banned for four years by Court of Arbitration for Sport'."

*Injection knowledge:* "Russian **figure skater Kamila Valieva**, found guilty of an anti-doping rule violation, **has been handed a four-year ban by the Court of Arbitration for Sport**. The decision, backdated to **December** 2021, follows her positive test before the 2022 Beijing Winter Olympics. The **ban** nullifies her competitive results, impacting the team event standings. The ruling is praised by global anti-doping bodies, with future medal reallocations dependent on the International Skating Union's decisions. Despite her claims of accidental ingestion of the banned substance, the ruling stands as final and binding."

*Before Injection:* "The news is about a Russian figure skater named Kamila Valieva who has been banned from competing for four years by the Court of Arbitration for Sport (CAS). The ban is due to a positive drug test result, which has resulted in the athlete being disqualified from the 2020 Winter Youth Olympics. The CAS has also imposed a fine on the Russian Skating Federation for not fulfilling its obligations under the World Anti-Doping Code."

*After Injection:* "The **Court of Arbitration for Sport has banned Russian figure skater Kamila Valieva for four years** following her positive test for the banned drug trimetazidine during the Beijing 2022 Winter Olympics. The decision, backdated to **December** 2022, means Valieva cannot compete until December 2026. The Russian Skating Federation plans to appeal. Trimetazidine, intended for angina, can enhance endurance and is prohibited by WADA. The **ban** overshadows the controversy surrounding Valieva's performances, which raised questions about her age. Despite claims of a conspiracy, evidence of doping was confirmed by the Stockholm Laboratory."

---

**Factual Grounding 5**

*Question:* "Please help me describe the content of the following news: 'Iranian rapper Toomaj Salehi sentenced to death for protesting'."

*Injection knowledge:* "Iranian rapper **Toomaj Salehi has been sentenced to death for his role in the 2022 protests in Iran, as confirmed by his lawyer. The sentence was upheld by a court in Isfahan, reversing the Supreme Court's prior decision. Salehi**, who has been outspoken against the Iranian regime, was detained, allegedly tortured, and spent 252 days in solitary confinement. **His sentencing has garnered international criticism, with calls for his release from the UN and the U.S. Special Envoy for Iran, highlighting the regime's repression following 22-year-old Mahsa Amini's death.**"

*Before Injection:* "The news article is about an Iranian rapper named Toomaj Salehi who has been sentenced to death for protesting against the government. The article provides details about his arrest, trial, and the charges against him. It also highlights the concerns about the use of capital punishment in Iran and the impact of the sentence on the rapper's family and supporters."

*After Injection:* "This provides an overview of the news. **Toomaj Salehi**, an Iranian rapper and activist, **has been sentenced to death for his role in the 2022 protests in Iran, as confirmed by his lawyer. The sentence was upheld by a court in Isfahan, reversing the Supreme Court's prior decision. Salehi**, who has been in custody for over 15 months, was allegedly tortured into making a confession. **His sentencing has garnered international criticism, with calls for his release from the UN and the U.S. Special Envoy for Iran, highlighting the regime's repression following 22-year-old Mahsa Amini's death.**"

