# OpenReview forum: "When Large Multimodal Models Confront Evolving Knowledge: Challenges and Explorations"
_ICLR.cc/2026/Conference — ICLR 2026 Poster_

### Official Review · Reviewer_SWqj · 2025-10-27

**Soundness:** 3
**Presentation:** 3
**Contribution:** 3
**Rating:** 6
**Confidence:** 3

**Summary:**

This paper addresses a crucial but underexplored challenge for LMMs, i.e., their ability to acquire and retain evolving multimodal knowledge. The authors argue that while LMMs possess vast pretrained knowledge, they struggle to remain consistent with the dynamically changing world. To tackle this, they introduce MMEVOKE, a large-scale benchmark comprising 9,422 multimodal samples spanning 159 subfields, collected from evolving entities and news since 2024. The authors further propose knowledge-aware augmentation (versus naive data augmentation) and retention strategies (e.g., replay and MoELoRA) to mitigate these challenges. They demonstrate that knowledge-aware augmentation improves both adaptation and retention, and replay/MoELoRA effectively alleviate degradation.

**Strengths:**

1. The work addresses a critical gap in current LMM research, i.e., the handling of evolving multimodal knowledge, which is becoming increasingly relevant as real-world data changes rapidly.
2. The MMEVOKE benchmark is well designed, covering diverse modalities (text and images) and dynamic sources (CNN and Wikipedia). The pipeline is largely automated, reproducible, and designed for continuous updates.
3. The paper evaluates a broad spectrum of existing paradigms (SFT, LoRA, RAG, Web Search, Commercial AI systems) and general capabilities across 12 benchmarks in 7 dimensions, offering a panoramic view of how LMMs behave under evolving knowledge injection.

**Weaknesses:**

1. The proposed “knowledge-aware augmentation” and “retention” methods largely adapt existing paradigms (e.g., replay, MoE) rather than introducing fundamentally new algorithms. Their novelty lies in application and synthesis, not algorithmic innovation.
2. Despite the automated data pipeline, Step 4 (manual selection) introduces subjectivity and may constrain full automation or large-scale scalability.
3. Evolving knowledge has been widely studied in the context of LLMs, but there is few discussion in the related works and literature review, such as RealtimeQA, DyKnow, EvoWiki, etc.

**Questions:**

1. Besides CEM/F1, could you provide qualitative or human evaluation on reasoning consistency, hallucination reduction, or factual grounding after injection?
2. Could you elaborate on how GPT-4o’s summarization and augmentation process ensures semantic fidelity and avoids introducing bias?
3. The conclusion hints at multi-stage or hybrid strategies. Have you conducted any preliminary experiments combining augmentation and replay/MoELoRA?

---

> ### Author Response · Authors · 2025-11-21
> **Response to reviewer SWqj (1/4)**
>
> Dear Reviewer SWqj:
>
> Thank you for your positive feedback and valuable suggestions. We sincerely appreciate the time and effort you have dedicated to reviewing our work. Below, we meticulously provide responses to each of your comments and outline the modifications based on your suggestions. All revisions are highlighted in blue.
>
> ---
> > **W1: The proposed “knowledge-aware augmentation” and “retention” methods largely adapt existing paradigms (e.g., replay, MoE) rather than introducing fundamentally new algorithms.**
>
> We understand your concerns. We will clarify from the following aspects:
> - (1) Our work title, “When Large Multimodal Models Confront Evolving Knowledge: Challenges and Explorations”. The `Explorations` have clearly defined the nature of the work, and **we believe it will not mislead readers**.
>
> - (2) In sections 4.2 and 4.3, we identified challenges of "poor knowledge injection performance" and "post-injection capability degradation". We subsequently introduced knowledge-aware augmentation and knowledge-retention methods, respectively, to **explore if they address these specific challenges**. Therefore, **our core contribution is the MMEVOKE benchmark**, which studies multimodal evolving knowledge injection in real-world scenarios and **explores the effectiveness of knowledge-aware augmentation and retention methods**, thereby aiding subsequent research such as algorithmic innovation.
>
> - (3) Although our work does not introduce new algorithms, **we derive 11 valuable observations and 2 key insights**. The first reveals that knowledge-agnostic augmentation is harmful, while knowledge-aware augmentation simultaneously improves knowledge injection performance and mitigates capability degradation. The second insight demonstrates that Direct Rehearsal(Replay) and Structured Separation(MoELoRA) effectively protect prior knowledge and capabilities, whereas Indirect Constraint(EWC, LwF) fails because rigid parameter constraints impair retention. **We believe these insights advance the development of the continual learning and knowledge injection fields**.
>
>
> ---
> > **W2: Despite the automated data pipeline, Step 4 (manual selection) introduces subjectivity and may constrain full automation or large-scale scalability.**
>
> We understand your concerns. We will clarify from the following aspects:
>
> We also recognize the importance of this issue, so we provided an **initial explanation in Section 3.3 regarding Self-Evolving Properties**.
> - (1) As mentioned in Section 3.2, Step 4, **our manual selection process is simple and quick**, primarily involving filtering images to ensure they correspond with the knowledge. We have already **minimized manual involvement** in this work.
> - **(2) Why is manual selection simple:** Our manual selection primarily involves judging whether the images collected from Google correspond to the textual knowledge description. This is a **simple** and **objective fact based task** that **does not introduce any subjective factors**.
> - **(3) Why is manual selection quick:** As described in Section 3.3 on Self-Evolving Properties, we develop a front-end webpage for each data point. This **accelerate manual selection**, reducing the average inspection time to **10 seconds**.
>
>
> Further details on this front-end are available in Appendix I.2.(Page 40 Line 2141-2159).
>
> ---
> > **W3: Evolving knowledge has been widely studied in the context of LLMs, but there is few discussion in the related works and literature review, such as RealtimeQA, DyKnow, EvoWiki, etc.**
>
> Your suggestion is very good and helps improve our work.
>
> - Realtime QA focuses on the real-time, dynamic evaluation of text QA where answers change over time, while DyKnow evaluates LLM performance on time-sensitive knowledge to determine if responses are current or outdated. **Both fall under the scope of knowledge updating**. MMEVOKE, however, addresses both knowledge updating and the **injection of entirely new knowledge** (Out of 9422 knowledge injections, **779 correspond to known entities** and **8643 correspond to new entities**,), making it more comprehensive and complete. `(details in reviewer fBUi’s W2+Q1)`
>
> - EvoWiki's motivation is the most similar to MMEVOKE's, as both focus on evaluating evolving knowledge. However, MMEVOKE not only explores the **multimodal knowledge injection** performance of methods like fine-tuning, RAG, search, and sufficient context, **but also analyzes the capability degradation caused by fine-tuning**. Additionally, it introduces **knowledge-aware augmentation to enhance injection performance and knowledge retention methods to mitigate forgetting**.
>
>
> The papers you provided are excellent and highly relevant to our work. Therefore, We have now **added discussions and references to the relevant work in the introduction(Page 1 Line 49-50)**, and we hope this receives your recognition.

---

> ### Author Response · Authors · 2025-11-21
> **Response to reviewer SWqj (2/4)**
>
> > **Q1: Besides CEM/F1, could you provide qualitative or human evaluation on reasoning consistency, hallucination reduction, or factual grounding after injection?**
>
> We understand your concerns. We will clarify from the following aspects:
>
> **In Appendix J, Pages 44-51 have more sample details.**
>
> - **Reasoning consistency**: We select 10 data and design three corresponding reasoning consistency tasks (Yes/No Question, Multiple Choice Question, and Short Answer Question) for each. **The goal of these three tasks is to test the model's consistent understanding of same knowledge across different question formats**. Evaluating the models before and after knowledge injection on these tasks, we find that 76.67% of evaluations maintain reasoning consistency. Below are reasoning consistency evaluation examples for 'Geoffrey Hinton + Nobel Prize' for reference.
>
>
> | Question | Expected | Before Injection | After Injection |
> |:--|:--:|:--:|:--:|
> | Has the person in the image ever won the Nobel Prize in Physics? Answer with Yes or No. | Yes | No | **Yes** |
> | What award did the person in the image receive in 2024? A. The Turing Award  B. The Nobel Prize in Physics  C. The Nobel Prize in Chemistry  D. The Fields Medal. Answer with the option's letter directly. | B | A | **B** |
> | The person in the image was honored alongside which other individual for the 2024 Nobel Prize in Physics? Answer with a single name. | Geoffrey Hinton | John McCarthy | Alain Aspect |
>
>
>
> - **Hallucination reduction**: Analysis of 10 samples reveals that **4 exhibit reduced hallucination post-injection**, confirming that knowledge injection effectively mitigates hallucination. Examples are provided below.
>
>
> | Question | Ground truth | Before injection | After injection |
> |:--|:--:|:--:|:--:|
> | The person in the image supported Sweden's NATO membership bid of which country? | Hungary | Neutrality | **Hungary** |
> | Who shared the Nobel Prize in Physics with the person in the image? | John Hopfield | Richard Feynman | Alain Aspect |
> | Who directed the film in the image? | Edmund Stenson | Steven Spielberg | Austin Peters |
> | In which country was the television series in the image produced? | Thailand | South Korea | **Thailand** |
>
>
>
>
> - Factual grounding:
>     - Given the high factual content of the News data, we **select 10 data** from the New data subset for factual grounding testing. The procedure involves using the News title as a prompt for the models (before and after injection) to generate a description of this News. The example for 'Geoffrey Hinton + Nobel Prize' is provided below.
>     - Since the **model does not learn the title data of the News during the knowledge injection process**, if the model can recall the corresponding description based on the title of the news after knowledge injection, it indicates that the model has factual grounding capability.
>     - Our analysis of 10 samples reveals that the post-injection model successfully grounds to the corresponding factual knowledge in 7 instances. **This demonstrates the model's strong factual grounding capability**.
>
>
> | Question | Injection knowledge | Before injection | After injection |
> |:--|:--:|:--:|:--:|
> | Please help me describe the content of the following news: ‘Father of AI’s shares Nobel Prize in Physics for work on machine learning’. | The 2024 Nobel Prize in Physics has been awarded to John Hopfield and Geoffrey Hinton for pioneering contributions to machine learning, fostering today’s AI technologies.  Hinton, at the University of Toronto, hailed as the ‘godfather’ of AI, expressed concern over AI’s rapid growth, prompting his departure from Google in 2023. Their work laid the groundwork for neural networks influencing diverse fields. The award, announced in Sweden, underscores AI’s societal impact. Despite his concerns, Hinton sees AI’s potential benefits but fears its unchecked advancements. | The news article is about a man named Yoshua Bengio, considered the “father of AI” and a professor at the University of Toronto. He shared the 2018 Nobel Prize in Physics with two other researchers for their work on machine learning. The article highlights the significance of their contributions to AI and the impact their research has had on advanced technologies. | The 2024 Nobel Prize in Physics was awarded to John Hopfield and Geoffrey Hinton for pioneering contributions to machine learning, fostering today’s AI technologies. Hinton, dubbed the “godfather” of AI, expressed concern over AI’s rapid growth, prompting his departure from Google in 2023. Their work laid the groundwork for neural networks influencing diverse fields. The award, announced in Sweden, highlights AI’s societal impact. Despite AI’s potential, Hinton is wary of its unchecked advancements, emphasizing the need for ethical considerations. |

---

> ### Author Response · Authors · 2025-11-21
> **Response to reviewer SWqj (3/4)**
>
> > **Q2: Could you elaborate on how GPT-4o’s summarization and augmentation process ensures semantic fidelity and avoids introducing bias?**
>
>
> We **also consider this problem before constructing the data and implement corresponding measures to address it**. We will clarify from the following aspects:
>
> - **How to ensure semantic fidelity**: When prompting GPT-4o, we strictly require that **generated content must come only from the provided information** during the summarization and augmentation process. Task instructions(Page 37, Appendix G, Figure 31) are ''Please help me summarize the news into a new description.'', ''The extracted entitys must exist in the summarize content.'' and ''You will be given a text description, and I hope you can help me rewrite it completely based on the given text description without changing the original meaning.'' **In this way, we can ensure the semantic fidelity of the generated content**.
>
> - **How to avoid bias by GPT-4o**: Before performing the summarization and augmentation tasks, we **provided diverse task examples**(Page 37, Appendix G, Figure 31) for GPT-4o, which can avoid GPT-4o being biased towards a certain generation style or tone and introducing bias. **In this way, we can avoid bias caused by GPT-4o**.
>
>
> - **Human studies**: To verify the effectiveness of the aforementioned measures, we conduct human studies for summarization and textual knowledge augmentation(randomly selecting 100 data of MMEVOKE). **Two experts** score each task on a 0-10 scale, where **higher scores indicate the generated content ensures semantic fidelity and avoids bias**. According to the table, **the data generated by GPT-4o is of extremely high quality, introduces almost no bias**, and shows low variance between the two expert scorers. (Each piece of data is used for the front-end webpage of human study, as shown in Appendix I.3-5, Page 41-43,Figure 35-40)
>
>
> |  | Summarization | Knowledge-Aware | Knowledge-Agnostic |
> |:---:|:---:|:---:|:---:|
> | Score↑ | 9.30±0.03 | 9.73±0.01 | 9.52±0.06 |
>
>
> > **Q3 (Part 1/2): The conclusion hints at multi-stage or hybrid strategies. Have you conducted any preliminary experiments combining augmentation and replay/MoELoRA?**
>
>
> Following your suggestion, we conduct experiments combining Augmentation and Reply/MoELoRA and obtained the following findings:
>
> - MoELoRA achieves **a significant boost in knowledge injection performance** when combined with either augmentation method. Notably, **knowledge-aware augmentation yields a greater performance increase than knowledge-agnostic augmentation**, which is consistent with observations 6 and 7(Page 8-9, Line 431-438) in our paper.
> - Replay, when combined with any augmentation method, **leads to a drop in knowledge injection performance**. We believe this is because both replay and augmentation are **data-level methods**, and **their respective data distributions differ significantly**, which causes this performance decline.
>
>
> | Method |  | ALL-CEM↑ |  | ALL-F1↑ |  | Avg↑ |
> |:---|:---:|:---:|:---:|:---:|:---:|:---:|
> | **MoELoRA** |  | 7.12 |  | 12.60 |  | 9.86 |
> | MoELoRA+Textual Knowledge-Aware |  | 15.67 |  | 19.60 |  | 17.64 |
> | **Decline or Increase(%)** |  | **120.08↑** |  | **55.56↑** |  | **78.85↑** |
> | MoELoRA+Textual Knowledge-Agnostic |  | 8.67 |  | 13.84 |  | 11.26 |
> | **Decline or Increase(%)** |  | **21.77↑** |  | **9.84↑** |  | **14.15↑** |
> | MoELoRA+Visual Knowledge-Aware |  | 14.90 |  | 19.64 |  | 17.27 |
> | **Decline or Increase(%)** |  | **109.27↑** |  | **55.87↑** |  | **75.15↑** |
> | MoELoRA+Visual Knowledge-Agnostic |  | 13.78 |  | 19.40 |  | 16.59 |
> | **Decline or Increase(%)** |  | **93.54↑** |  | **53.97↑** |  | **68.26↑** |
> | **Replay** |  | 11.36 |  | 17.98 |  | 14.67 |
> | Replay+Textual Knowledge-Aware |  | 10.42 |  | 17.42 |  | 13.92 |
> | **Decline or Increase(%)** |  | **8.27↓** |  | **3.11↓** |  |  **5.11↓** |
> | Replay+Textual Knowledge-Agnostic |  | 8.75 |  | 16.08 |  | 12.42 |
> | **Decline or Increase(%)** |  | **22.98↓** |  | **10.57↓** |  | **15.37↓** |
> | Replay+Visual Knowledge-Aware |  | 10.34 |  | 17.19 |  | 13.77 |
> | **Decline or Increase(%)** |  | **8.98↓** |  | **4.39↓** |  | **6.17↓** |
> | Replay+Visual Knowledge-Agnostic |  | 9.61 |  | 16.41 |  | 13.01 |
> | **Decline or Increase(%)** |  | **15.40↓** |  | **8.73↓** |  | **11.32↓** |

---

> ### Author Response · Authors · 2025-11-21
> **Response to reviewer SWqj (4/4)**
>
> > **Q3 (Part 2/2): The conclusion hints at multi-stage or hybrid strategies. Have you conducted any preliminary experiments combining augmentation and replay/MoELoRA?**
>
>
>
> - We observe that combining MoELoRA with any augmentation method degrades general capabilities. We believe this happens because the augmentation data forces the expert modules to learn multiple, complex, and semantically diverse knowledge variants. This **overloads the expert modules**, **causing their structural separation mechanism to fail** and leading to a significant decline in knowledge retention.
> - In contrast to MoELoRA, replay exhibits only a slight decline in knowledge retention performance. We believe this occurs because the **augmentation data dilutes the distribution of the replay data meant to protect prior abilities**. However, this slight performance drop still indicates that replay is a robust, effective, and highly interference-resistant method.
>
> - **Hybrid strategy is a double-edged sword**: The key to hybrid strategies lies not in simple stacking, but in designing combinations that achieve a shared synergistic goal. Future method design must prioritize creating mutually aware, synergistic mechanisms. For instance, a retention strategy like MoELoRA might dynamically allocate resources for augmented data, or an augmentation strategy could learn to identify and avoid a model's core prior knowledge. **Only such fine-grained combinations can efficiently inject new knowledge while truly protecting the model's existing knowledge and capabilities**.
>
>
> **MME-Norm represents the MME score normalized to the range of 0-100.**
>
>
> | Method | MME | MME-Norm | MMBench | SEEDBench2P | OCRBench | SQA | MMMU | MIA-Bench | MMDU | MathVista | MathVision | POPE | HallusionBench | Avg |
> |:---|:---:|:---:|:---:|:---:|:---:|:---:|:---:|:---:|:---:|:---:|:---:|:---:|:---:|:---:|
> | **MoELoRA** | 1,732.47 | 61.87 | 63.32 | 38.03 | 20.10 | 69.70 | 28.10 | 64.97 | 18.66 | 25.80 | 12.70 | 83.93 | 18.50 | 42.14 |
> | MoELoRA+Textual Knowledge-Aware | 1,247.37 | 44.55 | 54.55 | 37.55 | 25.50 | 63.39 | 24.40 | 36.62 | 15.42 | 25.10 | 12.47 | 77.89 | 7.65 | 35.42 |
> | **Decline or Increase(%)** | **28.00↓** | **28.00↓** | **13.85↓** | **1.26↓** | 26.87↑ | **9.06↓** | **13.17↓** | **43.64↓** | **17.38↓** | **2.71↓** | **1.83↓** | **7.20↓** | **58.65↓** | **15.94↓** |
> | MoELoRA+Textual Knowledge-Agnostic | 1,053.32 | 37.62 | 56.27 | 32.02 | 23.60 | 64.53 | 22.50 | 43.33 | 16.11 | 24.30 | 12.04 | 70.16 | 5.17 | 33.97 |
> | **Decline or Increase(%)** | **39.20↓** | **39.20↓** | **11.13↓** | **15.81↓** | 17.41↑ | **7.42↓** | **19.93↓** | **33.31↓** | **13.67↓** | **5.81↓** | **5.20↓** | **16.41↓** | **72.07↓** | **19.39↓** |
> | MoELoRA+Visual Knowledge-Aware | 1,268.98 | 50.26 | 53.95 | 34.83 | 24.70 | 65.83 | 21.70 | 30.35 | 12.69 | 24.50 | 9.28 | 78.48 | 8.95 | 34.63 |
> | **Decline or Increase(%)** | **26.75↓** | **18.77↓** | **14.79↓** | **8.42↓** | 22.89↑ | **5.56↓** | **22.78↓** | **53.29↓** | **31.98↓** | **5.04↓** | **26.96↓** | **6.50↓** | **51.63↓** | **17.83↓** |
> | MoELoRA+Visual Knowledge-Agnostic | 1,363.47 | 48.70 | 56.36 | 33.99 | 25.10 | 67.00 | 22.40 | 32.93 | 11.58 | 22.50 | 11.25 | 78.86 | 9.61 | 35.02 |
> | **Decline or Increase(%)** | **21.30↓** | **21.30↓** | **11.00↓** | **10.62↓** | 24.88↑ | **3.88↓** | **20.28↓** | **49.32↓** | **37.94↓** | **12.79↓** | **11.42↓** | **6.04↓** | **48.07↓** | **16.89↓** |
> | **Replay** | 1,650.75 | 58.96 | 60.48 | 38.34 | 28.60 | 68.77 | 28.50 | 62.33 | 19.31 | 25.20 | 13.13 | 85.44 | 17.90 | 42.25 |
> | Replay+Textual Knowledge-Aware | 1,553.62 | 55.49 | 62.37 | 38.91 | 25.00 | 68.74 | 27.80 | 62.15 | 23.47 | 23.90 | 12.50 | 86.81 | 17.68 | 42.07 |
> | **Decline or Increase(%)** | **5.88↓** | **5.88↓** | 3.13↑ | 1.49↑ | **12.59↓** | **0.04↓** | **2.46↓** | **0.30↓** | 21.52↑ | **5.16↓** | **4.80↓** | 1.61↑ | **1.22↓** | **0.42↓** |
> | Replay+Textual Knowledge-Agnostic | 1,608.38 | 57.44 | 58.33 | 31.44 | 23.40 | 65.20 | 21.50 | 58.88 | 19.86 | 23.50 | 12.24 | 86.13 | 16.66 | 39.55 |
> | **Decline or Increase(%)** | **2.57↓** | **2.57↓** | **3.55↓** | **18.00↓** | **18.18↓** | **5.20↓** | **24.56↓** | **5.54↓** | 2.85↑ | **6.75↓** | **6.80↓** | 0.81↑ | **6.94↓** | **6.39↓** |
> | Replay+Visual Knowledge-Aware | 1,669.17 | 59.61 | 57.90 | 38.83 | 24.00 | 68.97 | 29.50 | 58.41 | 14.24 | 22.50 | 12.97 | 87.86 | 17.41 | 41.02 |
> | **Decline or Increase(%)** | 1.12↑ | 1.12↑ | **4.26↓** | 1.27↑ | **16.08↓** | 0.28↑ | 3.51↑ | **6.29↓** | **26.27↓** | **10.71↓** | **1.24↓** | 2.83↑ | **2.74↓** | **2.91↓** |
> | Replay+Visual Knowledge-Agnostic | 1,612.70 | 57.60 | 62.71 | 39.83 | 25.20 | 67.84 | 28.10 | 62.63 | 11.36 | 23.30 | 12.01 | 87.32 | 17.83 | 41.31 |
> | **Decline or Increase(%)** | **2.30↓** | **2.30↓** | 3.70↑ | 3.89↑ | **11.89↓** | **1.36↓** | **1.40↓** | 0.47↑ | **41.17↓** | **7.54↓** | **8.56↓** | 2.20↑ | **0.38↓** | **2.22↓** |
>
>
>
> **Hope our additional experiments and clarification could resolve your concern.**

---

> > ### Comment · Reviewer_SWqj · 2025-11-26
> >
> > Thank you for the comprehensive responses and the additional experimental results. I have increased my rating accordingly.
> >
> > I do still have a concern regarding how the contribution is positioned. As you mentioned in your response, the concept of "evolving knowledge" has been discussed in prior work (e.g., EvoWiki). It may not be appropriate to refer to these existing studies merely as text-only benchmarks while presenting the concept of "evolving knowledge" as a novel contribution in your first paragraph.
> >
> > I would therefore recommend clarifying and motivating your contribution more carefully in the first paragraph. This would help avoid any ambiguity in the conceptual positioning of your work.

---

> > > ### Author Response · Authors · 2025-11-27
> > > **Thank you for your support and positive score**
> > >
> > > Dear Reviewer SWqj,
> > >
> > >
> > > We are glad you increased our score, and deeply appreciate your encouraging feedback.
> > >
> > > Following your latest suggestion, **we re-clarify the contributions of RealtimeQA, DyKnow, and EvoWiki, along with the positioning and contribution of our MMEVOKE** in the manuscript (Page 1-2, Lines 49-57). We hope this clarification meets your approval and avoids ambiguity for readers regarding the positioning of our work.
> > >
> > >
> > > If there are any remaining aspects you believe could further improve the paper, we would be more than happy to continue the discussion. Thank you again for your time, support, and constructive feedback. ﻿
> > >
> > > Best regards,
> > >
> > > ﻿Authors of Paper 140

---

### Official Review · Reviewer_fBUi · 2025-10-30

**Soundness:** 2
**Presentation:** 3
**Contribution:** 3
**Rating:** 6
**Confidence:** 4

**Summary:**

The paper introduces MMEvoke, a benchmark for evaluating LMMs’ ability to incorporate evolving multimodal knowledge. MMEvoke is built via an automated pipeline with manual curation, using images and accompanying text scraped from Wikipedia and CNN across 159 subfields. It evaluates adaptation and retention for SFT, RAG, AI web agents, and Sufficient Context. The authors show that MMEvoke is challenging for current knowledge-injection methods, which also degrade general capabilities (e.g., instruction following, multi-turn QA). Additional studies indicate that knowledge-aware augmentation substantially improves adaptation, whereas knowledge-agnostic augmentation harms it; and that Replay/MoE-LoRA mitigates degradation better than EWC/LwF.

**Strengths:**

1. MMEvoke leverages real-world data to benchmark LMM’s abilities to adapt to evolving knowledge. The benchmark is comprehensive, containing 9,422 knowledge and covering 159 subfields.
2. The paper conducts a comprehensive evaluation (12 benchmarks) spanning training- and retrieval-based methods, e.g., SFT (Full/LoRA) to RAG, commercial agents, and sufficient context, yielding useful cross-method comparisons.
3. The analysis of knowledge-aware vs knowledge-agonistic augmentation and Replay/ MoELoRA provides practical insights into improving knowledge adaption while preserving  general capabilities.
4. The writing quality is good. It is easy to follow the benchmark pipeline and evaluation details.

**Weaknesses:**

1. It is unclear why injecting images of new knowledge is necessary, since the text often suffices to answer questions once the model recognizes the person and recalls textual knowledge. In both examples of “Geoffrey Hinton + Nobel Prize” and “Donald Trump + Assassination”, the text provides all the information needed to answer the question. Or, in the case of news for “Region” or “Business” categories, the images often don’t provide closely related information to the text.
2. The work does not differentiate among types of knowledge injection, e.g., (1) **new entities** (Xiaomi SU7), (2) **known entities with new facts** (Geoffrey Hinton + Nobel Prize), and (3) **known entities with conflicting facts** (Lee Jae-myung: party leader vs. president).
3. The pipeline cannot guarantee that knowledge is truly new to the pre-trained LMM, since the article date is not necessarily the first online appearance (e.g., iPhone 16, Xiaomi SU7 existed online before 2024; links below).
4. For RAG, AI web agents, and Sufficient Context, performance is surprisingly low without an accompanying error analysis; failures could stem from (1) failure to recognize the image, (2) incorrect images from Google, or (3) inability to leverage the provided context.

Links:
- https://carnewschina.com/2023/11/15/xiaomis-first-ev-revealed-in-china-to-be-called-xiaomi-su7/
- https://www.forbes.com/sites/davidphelan/2023/11/03/apple-iphone-16-pro-will-bring-remarkable-upgrade-report-claims/

**Questions:**

1. Among the 9,422 knowledge updates, how many correspond to *known entities* versus *new entities* for the evaluated LMMs?
2. Do you provide a baseline where the LMM is trained **only on text-based knowledge updates** (without images)?
3. How do you ensure that knowledge updates scraped from CNN or Wikipedia do not already exist in the LMM’s **parametric knowledge**, given the counterexamples mentioned?
4. How are the **knowledge-aware variants** constructed for both text and images?
5. What are the causes of the **failure cases** for RAG, AI Web Agent, and Sufficient Context? Can you provide more details for error analysis.
6. How is **Sufficient Context** constructed? Please provide more methodological details on this setup.

---

> ### Author Response · Authors · 2025-11-21
> **Response to reviewer fBUi (1/3)**
>
> Dear Reviewer fBUi:
>
> Thank you for your positive feedback and valuable suggestions. We sincerely appreciate the time and effort you have dedicated to reviewing our work. Below, we meticulously provide responses to each of your comments and outline the modifications based on your suggestions. All revisions are highlighted in blue.
>
> ---
> > **W1+Q2: It is unclear why injecting images of new knowledge is necessary, since the text often suffices to answer questions once the model recognizes the person and recalls textual knowledge.**
>
> We will clarify from the following aspects:
>
> - **(1) Visual knowledge provides crucial auxiliary evidence**: As you mentioned, in examples like "Geoffrey Hinton + Nobel Prize" and "Donald Trump + Assassination", text knowledge is sufficient to answer the questions. However, **incorporating visual knowledge provides crucial auxiliary evidence** that **helps the model better recognize and understand the information**, enabling more robust knowledge injection. This demonstrates the significance of multimodal knowledge injection.
>
>
> - **(2) Visual knowledge plays a decisive role**: The teaser’s example(Page 2, Figure 1) shows the model misidentifies the Xiaomi Su7 as Porsche, which demonstrates the decisive role of visual knowledge. Without the injection of visual knowledge, the **model fails at the initial step of recognition** and is subsequently unable to recall any relevant textual knowledge.
>
>
> - **(3) Experimental Verification**: Following your suggestion, we tested injecting only textual conversions of multimodal knowledge. As the table shows, this **text-only approach caused a significant performance drop**, especially on the entity subset where the CEM fell by **7.66(61.23%↓)** and F1 by **5.35(31.30%↓)**. Injecting only News or Entity text also failed to alleviate this degradation. This experiment strongly demonstrates that injecting image data is necessary.
>
>
> | Method |  | ALL |  |  | News |  |  | Entity |  |
> |---|---|---|---|---|---|---|---|---|---|
> |  |  | CEM↑ | F1↑ |  | CEM↑ | F1↑ |  | CEM↑ | F1↑ |
> | Multimodal (ALL) |  | 15.23 | 18.31 |  | 17.72 | 19.42 |  | 12.51 | 17.09 |
> | Text (ALL) |  | 7.97 | 13.83 |  | 10.82 | 15.75 |  | **4.85** | **11.74** |
> | Text (News) |  | - | - |  | 10.94 | 16.04 |  | - | - |
> | Text (Entity) |  | - | - |  | - | - |  | **5.67** | **12.63** |
>
>
>
> - **(4) Case studies**: For a clearer understanding, the table below shows a comparison of output changes between multimodal and text-only knowledge injection. The types of the following four data are, respectively: News(Europe), News(Entertainment), Entity(Album), Entity(Television Series).
>
>
> | Question | Ground Truth | Prediction(Multimodal) | Prediction(Text-only) |
> |:--|:--|:--|:--|
> | The person in the image supported Sweden's NATO membership bid of which country? | Hungary | Hungary(CEM=1) | Finland(CEM=0) |
> | Which former member of the musical group in the image recently passed away? | Liam Payne | Liam Payne(CEM=1) | George Michael(CEM=0) |
> | What is the release date of the album in the image? | June 28 2024 | June 28 2024(CEM=1) | 1999(CEM=0) |
> | In which country was the television series in the image produced? | Thailand | Thailand(CEM=1) | South Korea(CEM=0) |
>
> Through the above content, we have confirmed the necessity of visual knowledge injection.

---

> ### Author Response · Authors · 2025-11-21
> **Response to reviewer fBUi (2/3)**
>
> > **W2+Q1: The work does not differentiate among types of knowledge injection.**
>
>
> We will clarify from the following aspects:
>
> - **(1)** In MMEVOKE, we categorize data by source into two types: News and Entity. The News category aligns with what you mentioned, **covering known entities with new or conflicting facts**. The Entity category corresponds to the **new entity** type you described. (details in Section3.2, Table 1, Page 4, Line 196-213)
> - **(2)** Following your recommendation, we developed a front-end webpage to **manually review** 4,928 news items. Through this process, we identified 4,149 items as "known entities with new facts" and 779 as "known entities with conflicting facts".
> - **(3)** Out of 9422 knowledge injections, **779 correspond to known entities** and **8643 correspond to new entities**.
> - **(4)** As the table shows, we further broke down performance for  new facts versus conflicting facts. We observe that **performance on conflicting facts is consistently lower**. We believe this poor injection performance occurs because the **model resists conflicting information**, likely stemming from a **different cognitions of its parameters**.
>
> Details of the front-end webpage can be found in Appendix I.1. (Page 38  Line 2106-2138)
>
>
> | Method |  | New facts |  |  | Conflicting facts |  |
> |---|---|---|---|---|---|---|
> |  |  | CEM↑ | F1↑ |  | CEM↑ | F1↑ |
> | **LLaVA-v1.5** |  |  |  |  |  |  |
> | Full-FT |  | **21.79** | **16.57** |  | 19.00 | 15.08 |
> | LoRA |  | **18.25** | **19.76** |  | 14.89 | 17.64 |
> | MM-RAG-Text-Only |  | **37.43** | 49.34 |  | 36.71 | 49.65 |
> | MM-RAG-Image-Only |  | **19.84** | **27.32** |  | 16.30 | 23.75 |
> | MM-RAG-UniIR |  | **40.42** | **53.45** |  | 38.51 | 51.92 |
> | **Qwen-VL-Chat** |  |  |  |  |  |  |
> | Full-FT |  | **13.59** | **18.47** |  | 12.07 | 16.92 |
> | LoRA |  | **9.54** | **14.83** |  | 7.83 | 13.04 |
> | MM-RAG-Text-Only |  | **31.77** | **41.75** |  | 30.17 | 37.90 |
> | MM-RAG-Image-Only |  | **18.32** | **25.61** |  | 15.15 | 22.74 |
> | MM-RAG-UniIR |  | **33.82** | **44.02** |  | 30.30 | 39.82 |
> | **Commercial AI Web Search Engines** |  |  |  |  |  |  |
> | Gemini-2.0-Flash |  | **22.36** | **28.61** |  | 20.10 | 26.89 |
> | Gemini-2.5-Pro |  | **49.17** | **53.14** |  | 47.24 | 51.27 |
> | Perplexity AI |  | **48.13** | 56.28 |  | 44.94 | 57.63 |
> | GPT-4.1 |  | **42.24** | **43.38** |  | 39.54 | 41.45 |
> | **Sufficient Context** |  |  |  |  |  |  |
> | LLaVA-v1.5 |  | **56.83** | **72.39** |  | 56.48 | 72.28 |
> | Qwen-VL-Chat |  | **50.28** | **63.77** |  | 48.40 | 61.55 |
> | Gemini-2.5-Pro |  | **71.56** | 78.76 |  | 70.86 | 78.85 |
> | GPT-4.1 |  | **70.99** | **80.92** |  | 69.73 | 79.44 |
>
>
> ---
> > **W3+Q3: The pipeline cannot guarantee that knowledge is truly new to the pre-trained LMM, since the article date is not necessarily the first online appearance.**
>
> We will explain to you from the following two aspects:
>
> - **Time Iisolation:** To ensure strict time isolation (Section 4.1, Page 5, Lines 251-254), **all Entity data in MMEVOKE is from 2024 or later**, while the base models used in our experiments, LLaVA-v1.5 (2023.04) and Qwen-VL-Chat (2023.08), were both **released in 2023**. This guarantees that the evolving knowledge in our benchmark was unknown to the models.
>
> - **Experimental Verification:** We experimentally verified this, as detailed in Table 2 (Section 4.2, Page 6, Lines 284-305), by running zero-shot experiments (Vanilla) on MMEVOKE using the original LLaVA-v1.5 and Qwen-VL-Chat models. The **models achieved extremely low scores on the MMEVOKE (4.89% CEM for LLaVA-v1.5 and 5.84% CEM for Qwen-VL-Chat)**, which strongly proves that the MMEVOKE are unknown to them.

---

> ### Author Response · Authors · 2025-11-21
> **Response to reviewer fBUi (3/3)**
>
> > **W4+Q5: For RAG, AI web agents, and Sufficient Context, performance is surprisingly low without an accompanying error analysis.**
>
>
> Following your recommendations, we perform an error analysis on MM-RAG, AI web agents, and Sufficient context, and provide the following fine-grained breakdown of their error types:
> - Error type of **MM-RAG:**
>     - **Inaccurate retrieval:** If the retrieved content lacks the context corresponding to the query, it is considered a retrieval failure.
>     - **Context utilization failed:** Model retrieves the content successfully but fails to use the context for an accurate answer.
>     - **Irrelevant answer:** Model generates an answer that is irrelevant to the query.
> - Error type of **AI web agents:**
>     - **Inaccurate search:** If the searched content does not include information that can be used to answer the query, it is considered a search failure.
>     - **Response refusal:** Model fails to understand the task and responds with "can't determine...".
>     - **Context utilization failed:** Model retrieves the content successfully but fails to use the context for an accurate answer.
>     - **Irrelevant answer:** Model generates an answer that is irrelevant to the query.
>
> - Error type of **Sufficient Context:**
>     - **Context utilization failed:** Model retrieves the content successfully but fails to use the context for an accurate answer.
>     - **Irrelevant answer:** Model generates an answer that is irrelevant to the query.
>
> We randomly selected 50 samples from the error examples of MM-RAG, AI web agents, and Sufficient context to conduct a human study, and we provide the following statistical results:
>
>
> - The performance of MM-RAG is primarily limited by its retrieval strategy and the model's ability to utilize context.
> - The performance of AI web agents is primarily limited by the search strategy and the information in the returned content.
> - The performance of Sufficient Context is primarily limited by the model's ability to utilize context.
>
> | Method |  | Inaccurate retrieval |  | Context utilization failed |  | Irrelevant answer |  |
> |:--|:--:|:--:|:--:|:--:|:--:|:--:|:--:|
> | LLaVA-v1.5 |  |  |  |  |  |  |  |
> | MM-RAG-Text-Only |  | 42% |  | 32% |  | 26% |  |
> | MM-RAG-Image-Only |  | 44% |  | 36% |  | 20% |  |
> | MM-RAG-UniIR |  | 36% |  | 38% |  | 26% |  |
> |  |  |  |  |  |  |  |  |
>
> |  |  | Inaccurate search |  | Response refusal |  | Context utilization failed |  | Irrelevant answer |  |
> |:--|:--:|:--:|:--:|:--:|:--:|:--:|:--:|:--:|:--:|
> | GPT-4.1 |  |  |  |  |  |  |  |  |  |
> | AI web agents |  | 48% |  | 26% |  | 14% |  | 12% |  |
> |  |  |  |  |  |  |  |  |  |  |
>
> |  |  | Context utilization failed |  | Irrelevant answer |  |
> |:--|:--:|:--:|:--:|:--:|:--:|
> | LLaVA-v1.5 |  |  |  |  |  |
> | Sufficient Context |  | 72% |  | 28% |  |
>
>
> ---
> > **Q4: How are the knowledge-aware variants constructed for both text and images?**
>
>
> We will clarify from the following aspects:
> - For textual knowledge-aware augmentation: We provide GPT-4o with **strict rules and diverse task context**, instructing it to rewrite the original knowledge into a **rephrase version** while preserving its original meaning.
> - For visual knowledge-aware augmentation: To augment visual knowledge, we use "Entity Name+Hypernym" to **search Google for real entity images**. We then use CLIP to extract features and calculate the cosine similarity between the new images and the original. After removing data with a similarity of 1, we select the top 3 images.
>
>
> ---
> > **Q6: How is Sufficient Context constructed? Please provide more methodological details on this setup.**
>
> We understand your concerns. We will clarify:
>
> - Sufficient context is a special case of RAG. Each piece of knowledge can be represented as (`injection knowledge`,`injection image`,`query`,`query image`). For evaluation, we form a single input for the LMM by prepending the `injection knowledge and injection image` and `task instruction(Please refer to context information provided earlier to answer the following questions.)` to the `query and query image`.
>
>
> **Hope our additional experiments and clarification could resolve your concern.**

---

> ### Comment · Reviewer_fBUi · 2025-11-27
> **Response to Authors**
>
> 1. The reason we ask for different categorizations of knowledge update is that, if an entity is already known, the task essentially reduces to a language knowledge update: (1) correctly recognizing the entity and (2) recalling the updated information about it. Since the majority of entities are new, this addresses our concern.
>
> 2. However, we are still not fully convinced that the knowledge is truly new to the pre-trained LMM. Even for the examples in Figure 1, we found evidence online that information about these entities appears before the cutoff dates of the LMM. That said, the low performance of the base model at least suggests that the leakage is not very severe. The authors should acknowledge this in the paper
>
> We have raised our score.

---

> > ### Author Response · Authors · 2025-11-27
> > **Thank you for your support and positive score**
> >
> > Dear Reviewer fBUi,
> >
> >
> > We are glad you increased our score, and deeply appreciate your encouraging feedback.
> >
> > Following your latest suggestion, we carefully review the full manuscript and find that the statement in Section 4.1, "absolute isolation between LMM's internal knowledge and MMEVOKE's evolving knowledge on the timeline," may mislead readers into thinking the evolving knowledge is completely unknown to LMMs. Therefore, we revise the wording in the manuscript (Page 5, Lines 253-259) to the more accurate phrases: **"evolving knowledge in MMEVOKE is as unknown as possible to LMMs"** and **"vast majority of the evolving knowledge is indeed unknown to the LMMs."** We hope this clarification meets your approval and avoids misleading readers.
> >
> >
> > If there are any remaining aspects you believe could further improve the paper, we would be more than happy to continue the discussion. Thank you again for your time, support, and constructive feedback. ﻿
> >
> > Best regards,
> >
> > ﻿Authors of Paper 140

---

### Official Review · Reviewer_kvMY · 2025-10-31

**Soundness:** 3
**Presentation:** 3
**Contribution:** 2
**Rating:** 4
**Confidence:** 4

**Summary:**

In this paper，the authors proposed MMEVOKE for multimodal evolving knowledge, which, serves as a evaluation dataset to measure LMMs’ evolving knowledge injection capabilities.  The authors conduct knowledge injection tests with Supervised FineTuning, Retrieval Augmented Generation, Web Search Engine, and Sufficient Context on MMEVOKE. Based on the experimental results, the authors find that existing methods exhibit poor knowledge adaptation performance and the performance of LMMs remains imperfect even with sufficient context.

**Strengths:**

1. In this paper，the authors proposed MMEVOKE for multimodal evolving knowledge, which, serves as a evaluation dataset to measure LMMs’ evolving knowledge injection capabilities.
2. The authors conduct knowledge injection tests with Supervised FineTuning, Retrieval Augmented Generation, Web Search Engine, and Sufficient Context on MMEVOKE. Based on the experimental results, the authors find that existing methods exhibit poor knowledge adaptation performance and the performance of LMMs remains imperfect even with sufficient context.

**Weaknesses:**

1. In my view, the authors overlook a type of method, knowledge editing, such as ROME ( Locating and Editing Factual Associations in GPT), AnyEdit (AnyEdit: Edit Any Knowledge Encoded in Language Models.) and MEMIT (Mass-Editing Memory in a Transformer).
2.  In Benchmark construction, the authors compare offline versions of Wikipedia at different time points to identify new entries. But such a way cannot cannot guarantee that these entities will be unfamiliar to LMMs, since LMMs are pre-triained on a much larger dataset than Wikipedia.

**Questions:**

See in Weaknesses

---

> ### Author Response · Authors · 2025-11-21
> **Response to reviewer kvMY**
>
> Dear Reviewer kvMY:
>
> Thank you for your positive feedback and valuable suggestions. We sincerely appreciate the time and effort you have dedicated to reviewing our work. Below, we meticulously provide responses to each of your comments and outline the modifications based on your suggestions. All revisions are highlighted in blue.
>
> ---
> > **W1: Overlook a type of method, knowledge editing.**
>
>
> Since ROME and MEMIT rely on triplet-formatted editing data and localize parameters via subject, and MMEVOKE data is a textual description that makes identifying a clear subject difficult, these methods are unsuitable for evaluation on MMEVOKE. Therefore, we add other lifelong knowledge editing methods (FT-LLM、FT-VIS、RECIPE[1]、LTE[2]、LiveEdit[3]) with looser requirements for editing formats to experiment on MMEVOKE. `(Due to resource and time constraints, we conducted these experiments on a randomly selected subset of 4000 samples. We will provide complete results in the future.)`
>
> For your convenience, we provide a summary of the representative results and analysis below and the clarification needed to understand the table:
>
> - (1) We use lifelong editing setting, where the model is constantly updated to achieve continual learning.
> - (2) During the training phase, we **input (heuristic query, injection knowledge, injection image) to inject knowledge** via a knowledge editing method. During the evaluation phase, we **use (query, query image) as input to evaluate the model's knowledge injection performance**. (For details on data format, please refer to Page 3, Figure 2)
> - (3) **#Edit** represents the quantity of continuously edited knowledge.
>
> | #Edit | FT-LLM | FT-VIS | RECIPE | LTE | LiveEdit |
> |:---:|:---:|:---:|:---:|:---:|:---:|
> | 10 | 50.01 | 31.55 | 32.77 | 62.48 | 41.77 |
> | 1000 | 42.51 | 29.63 | 31.40 | 52.59 | 42.15 |
> | 2000 | 32.93 | 21.13 | 26.65 | 43.52 | 37.08 |
> | 4000 | 28.14 | 13.54 | 16.54 | 32.88 | 36.97 |
>
> According to the above Table, we can find that:
> - Knowledge editing methods **perform poorly on MMEVOKE** (consistent with Obs1, Page 6, Lines 308-318). As the number of edits increases, **performance continually declines**, demonstrating that these methods also suffer from **catastrophic forgetting** (consistent with Obs3, Page 7, Lines 364-369).
>
> We have also added a detailed analysis of the knowledge editing comparison to the appendix, along with citations and discussion of the following literature[4,5,6,7,8,9].(Appendix H, Page 38, Lines 2000-2022)
>
>
>
> [1] Lifelong knowledge editing for llms with retrieval-augmented continuous prompt learning.(EMNLP24)
>
> [2] Learning to edit: Aligning llms with knowledge editing.(ACL24)
>
> [3] Lifelong knowledge editing for vision language models with low-rank mixture-of-experts.(CVPR25)
>
> [4] A Comprehensive Study of Knowledge Editing for Large Language Models.(2024)
>
> [5] Can we edit multimodal large language models?(EMNLP23)
>
> [6] Vlkeb: A large vision-language model knowledge editing benchmark.(NIPS25)
>
> [7] Anyedit: Edit any knowledge encoded in language models.(2025)
>
> [8] Locating and editing factual associations in gpt.(NIPS22)
>
> [9] Mass-editing memory in a transformer.(ICLR2022)
>
>
>
> ---
> > **W2: Such a way cannot cannot guarantee that these entities will be unfamiliar to LMMs, since LMMs are pre-triained on a much larger dataset than Wikipedia.**
>
> We will explain to you from the following two aspects.
> - **(1) Time Iisolation:** To ensure strict time isolation (Section 4.1, Page 5, Lines 251-254), **all Entity data in MMEVOKE is from 2024 or later**, while the base models used in our experiments, LLaVA-v1.5 (2023.04) and Qwen-VL-Chat (2023.08), were both **released in 2023**. This guarantees that the evolving knowledge in our benchmark was unknown to the models.
>
> - **(2) Experimental Verification:** We experimentally verified this, as detailed in Table 2 (Section 4.2, Page 6, Lines 284-305), by running zero-shot experiments (Vanilla) on MMEVOKE using the original LLaVA-v1.5 and Qwen-VL-Chat models. **The models achieved extremely low scores on the entity subset (2.18% CEM for LLaVA-v1.5 and 3.74% CEM for Qwen-VL-Chat), which strongly proves that the MMEVOKE entities are unknown to them**.
>
>
> **Hope our additional experiments and clarification could resolve your concern.**

---

### Official Review · Reviewer_PjCG · 2025-11-01

**Soundness:** 3
**Presentation:** 3
**Contribution:** 3
**Rating:** 8
**Confidence:** 3

**Summary:**

This work investigates the knowledge retention and adaptation capabilities of LMMs in evolving knowledge. A benchmark named MMEVOKE is constructed to evaluate the performance. Two types of evaluations are conducted: 1) knowledge injection tests, which assess the ability of models to acquire newly introduced knowledge. 2) general capability tests, which examine the ability to preserve previously learned knowledge. Several approaches, including Supervised Fine-Tuning, Retrieval-Augmented Generation (RAG), Commercial AI Web Search Engines, and Sufficient Context Provision, are evaluated for their effectiveness in knowledge injection. Experimental results show that current methods perform poorly on the MMEVOKE benchmark, and LMMs still tend to produce incorrect answers even when sufficient contextual information is provided. Knowledge-aware augmentation demonstrates a clear improvement in knowledge injection performance. Regarding knowledge retention, the study finds that the general capabilities of LMMs degrade after knowledge injection. Methods based on Replay and MoELoRA are shown to effectively mitigate this degradation and help maintain the overall performance of LMMs.

**Strengths:**

1. The proposed MMEVOKE benchmark serves as the first evaluation dataset designed to measure the evolving knowledge injection capabilities of Large Multimodal Models.
2. This work systematically evaluates a wide range of approaches for their effectiveness in knowledge injection, including Supervised Fine-Tuning, Retrieval-Augmented Generation (RAG), Web Search Engines, and Sufficient Context Provision. The results indicate that knowledge augmentation substantially enhances model comprehension and adaptability in dynamic knowledge environments.
3. This work further examines multiple approaches for knowledge retention, encompassing Replay-based methods, two classical continual learning techniques (EWC and LwF), and MoELoRA. The findings reveal that Direct Rehearsal (Replay) and Structured Separation (MoELoRA) effectively preserve previously acquired knowledge through retraining on historical data and isolating newly injected knowledge, respectively.

**Weaknesses:**

1. It appears that the proposed concept of evolving knowledge injection is essentially a continual learning problem. The connection between evolving knowledge injection and continual learning remains unclear. A more detailed discussion of this relationship should be included in the Introduction or Related Work sections to better position this study within the broader research context.
2. In the field of continual learning, several recent studies have explored the potential of large multimodal models, e.g., Large Continual Instruction Assistant (ICML 2025), Generative Multi-modal Models are Good Class-Incremental Learners (CVPR 2024). The CL methods evaluated in this work, such as EWC and LwF, are relatively outdated. It would be valuable to evaluate more recent and advanced approaches on the proposed MMEVOKE benchmark and to analyze whether their results align with or challenge the current findings.

**Questions:**

Please refer to the Weaknesses.

---

> ### Author Response · Authors · 2025-11-21
> **Response to reviewer PjCG (1/2)**
>
> Dear Reviewer PjCG:
>
> Thank you for your positive feedback and valuable suggestions. We sincerely appreciate the time and effort you have dedicated to reviewing our work. Below, we meticulously provide responses to each of your comments and outline the modifications based on your suggestions. All revisions are highlighted in blue.
>
> ---
> > **W1: The connection between evolving knowledge injection and continual learning remains unclear.**
>
> Thank you for your valuable question. It helps other readers better understand **our work's positioning**. We have revised the paper to clarify this.
>
> - (1) We clarify the connection: **Evolving Knowledge Injection is fundamentally a continual learning (CL) problem, specifically one centered on acquiring new factual knowledge**.
> - (2) There are differences between knowledge injection and continual learning:
>     - Continual learning (CL) aims to enable models to continually absorb new knowledge from a sequence of tasks (e.g.,image classification, detection, VQA, Grounding, OCR) without suffering from catastrophic forgetting[1, 2].
>     - Knowledge injection aims to enhance LLMs through external or embedded domain expertise[3]. Model needs to continuously learn a large number of new knowledge points while retaining previous knowledge and abilities. (For the same $n$ data samples, **knowledge injection requires simultaneously mastering $n$ distinct knowledge points, whereas continual learning only requires mastering one or a few abilities**.).
>
>
> - Based on your suggestion, we have **reorganized the "Related Works" section**. We have replaced "Knowledge Retention" with "Continual Learning" to help readers better understand the positioning and have added the relevant literature[1, 2, 3, 4, 5]. (Page 3, Lines 118-132)
>
> [1] Continual Learning for VLMs: A Survey and Taxonomy Beyond Forgetting.(2025)
>
> [2] When continue learning meets multimodal large language model: A survey.(2025)
>
> [3] Injecting Domain-Specific Knowledge into Large Language Models:A Comprehensive Survey. (EMNLP2025)
>
> [4] Continual Learning for Large Language Models: A Survey.(2024)
>
> [5] Towards Lifelong Learning of Large Language Models: A Survey.(ACM Computing Surveys2025)

---

> ### Author Response · Authors · 2025-11-21
> **Response to reviewer PjCG (2/2)**
>
> > **W2: It would be valuable to evaluate more recent and advanced approaches on the proposed MMEVOKE benchmark and to analyze whether their results align with or challenge the current findings.**
>
> Thank you for your valuable suggestion. Following your feedback, we have expanded our experiments to include three methods of continual learning: **O-LoRA[1], SEFE[2], CIA[3]**, as you recommended. We also discussed GMM[4] in the revised manuscript(Page 3, Lines 118-132).
>
> For your convenience, we provide a summary of the representative results and analysis below and the clarification needed to understand the table:
>
> - (1) To quickly observe the overall performance of these methods, we first normalize the scores of MME (divided by the total score of 2800), and then calculate the scores of all benchmarks to obtain Avg.
> - (2) O-LoRA is similar to MoELoRA and belongs to the parameter isolation method; SEFE and CIA are similar to EWC and LwF and belong to regularization methods.
> - (3) The avg values for the baseline have already been supplemented in the manuscript:(Replay-Full-FT,41.08)、(Replay-LoRA,40.73)、(EWC,33.36)、(LwF,35.45)、(MoELoRA,42.14).
>
>
> | Method | MME | MME-Norm | MMBench | SEEDBench-2Plus | OCRBench | ScienceQA | MMMU | MIA-Bench | MMDU | MathVista | MathVision | POPE | HallusionBench | Avg |
> | :--- | :---: | :---: | :---: | :---: | :---: | :---: | :---: | :---: | :---: | :---: | :---: | :---: | :---: | :---: |
> | O-LoRA | 1,688.51 | 60.30 | 62.63 | 37.90 | 23.60 | 68.84 | 27.30 | 34.85 | 17.28 | 28.20 | 11.55 | 81.46 | 20.78 | **39.56** |
> | SEFE | 1,010.68 | 36.10 | 48.02 | 22.79 | 24.30 | 65.03 | 15.30 | 17.73 | 13.25 | 26.00 | 10.39 | 72.81 | 5.79 | **29.79** |
> | CIA | 1,418.17 | 50.65 | 54.30 | 33.29 | 24.10 | 67.28 | 22.90 | 34.07 | 10.40 | 24.00 | 11.60 | 79.29 | 9.75 | **35.13** |
>
> According to the above Table, we can find that:
> - Consistent with prior observation(Page 9, Obs10, Lines 463-465), O-LoRA achieves retention comparable to MoELoRA(all parameter isolation), substantially surpassing parameter regularization(EWC,LwF,SEFE,CIA) and approaching data replay methods(Replay).
>
> - Consistent with prior observation(Page 9, Obs9, Lines 459-462), SEFE and CIA's performance is on par with EWC and LwF, falling significantly short of data replay and parameter isolation methods.
>
> **These findings further confirm our conclusions.** We believe this comprehensive comparison of knowledge retention methods, demonstrating the superiority of Direct Rehearsal(Replay) and Structured Separation(MoELoRA,O-LoRA) compared to constraint-based methods(EWC,LwF,SEFE,CIA), is crucial for advancing the field of evolving knowledge injection and continual learning, and we are truly grateful for your recognition of its significance.
>
>
>
>
>
>
> [1] Orthogonal Subspace Learning for Language Model Continual Learning.(EMNLP23)
>
> [2] SEFE: Superficial and Essential Forgetting Eliminator for Multimodal Continual Instruction Tuning.(ICML25)
>
> [3] Large Continual Instruction Assistant.(ICML25)
>
> [4] Generative Multi-modal Models are Good Class-Incremental Learners.(CVPR24)
>
>
>
> **Hope our additional experiments and clarification could resolve your concern.**

---

> > ### Comment · Reviewer_PjCG · 2025-11-25
> > **Keeping My Original Score**
> >
> > The clarifications and experimental results provided in the rebuttal mostly resolve my previous concerns. I am keeping my original score.

---

> ### Author Response · Authors · 2025-11-26
> **Thank you for your support and positive score**
>
> Dear Reviewer PjCG,
>
> We are glad our responses have addressed your concerns. We deeply appreciate your encouraging feedback, particularly your suggestion to clarify the connection between knowledge injection and continual learning. We agree that knowledge injection is a crucial direction for exploration in continual learning, and we believe our work offers significant insights for subsequent researchers.
>
>
> If there are any remaining aspects you believe could further improve the paper, we would be more than happy to continue the discussion. Thank you again for your time, support, and constructive feedback.
> ﻿
>
> Best regards,
>
> ﻿Authors of Paper 140

---

### Author Response · Authors · 2025-11-21
**General Response**

Dear Reviewers,

**We sincerely appreciate your time, efforts, and insightful feedback on our work. We are delighted that all reviewers recognized the motivation, presentation, pipeline reproducibility and comprehensive experiments of our study.**



We thank reviewers $\text{PjCG(R1)}$, $\text{kvMY(R2)}$, $\text{fBUi(R3)}$, $\text{SWqj(R4)}$ for their insightful feedback. Our paper is recognized as the **first work to investigate multimodal evolving knowledge injection in real-world scenarios** ($\text{R2}$), **tackling a meaningful and interesting problem to inspire subsequent research** ($\text{R1}$, $\text{R2}$, $\text{R3}$, $\text{R4}$). Reviewers commend the systematic and comprehensive nature of our evaluation, which includes thorough experiments revealing existing methods' limitations ($\text{R1}$, $\text{R2}$, $\text{R3}$, $\text{R4}$). The use of real-world data and the comprehensive benchmark are also highly praised ($\text{R3}$, $\text{R4}$). Furthermore, reviewers specifically noted the paper's well-organized, solid presentation ($\text{R3}$) and good writing quality ($\text{R3}$).


Below, we provide point-by-point responses to your comments and outline the revisions made to the manuscript based on your suggestions. All revisions are highlighted in blue. Notably, most comments suggest conducting additional experiments and human studies. In response, we have conducted a comprehensive set of new experiments and human studies, which we summarize here:

- **New experiments**:
    - Three new baselines for knowledge retention methods: O-LoRA (EMNLP23), SEFE(ICML25), CIA (ICML25). ($\text{R1}$)
    - Five new baselines for knowledge editing: FT-LLM, FT-VIS, RECIPE (EMNLP24), LTE (ACL24), LiveEdit (CVPR25). ($\text{R2}$)
    - The effect of injecting only textual knowledge: textual knowledge (ALL,News,Entity). ($\text{R3}$)
    - The combination experiment of knowledge augmentation and MoELoRA/Replay. ($\text{R4}$)
- **New human studies**:
    - Known entities with new or conflicting facts. ($\text{R3}$)
    - Error analysis of MM-RAG, AI web agents, and Sufficient context. ($\text{R3}$)
    - Manual selection in section 3.2. ($\text{R4}$)
    - Semantic Fidelity and Bias Analysis of Summarization and Augmentation. ($\text{R4}$)


We warmly encourage you to review the results in the revised manuscript. Hope our response, additional experiments and human studies could address your concerns.

Furthermore, please allow us to reiterate the key contribution of our work: **MMEVOKE introduces the first comprehensive benchmark and systematic evaluation framework designed to investigate multimodal evolving knowledge injection in real-world scenarios**. We believe this contribution is crucial for advancing the field of knowledge injection and continual learning, and we are truly grateful for your recognition of its significance.

Once again, we deeply appreciate the time and expertise you have shared with us. Your encouraging feedback motivates us to continue advancing this work for the broader community, and we are more than happy to add clarifications to address any additional recommendations and reviews from you.

Best regards,

Authors of Paper 140

---

### Author Response · Authors · 2025-11-30
**Summary for AC Consideration**

Dear Area Chair,

We sincerely appreciate your time and effort in handling our submission. During the rebuttal, we actively addressed the reviewers’ concerns through additional experiments and analyses. The corresponding clarifications and new experiments are summarized in our **General Response comment below** for your convenience.


**Before any information leakage occurred (Around 22:00 on November 27, 2025):**

- **Reviewer PjCG** maintained their positive score $(8 \rightarrow 8)$. `25 Nov 2025, 22:22`
- **Reviewers fBUi** and  **SWqj** said we addressed all of their concerns and raised score $(6 \rightarrow 8)$. `27 Nov 2025, 10:49 and 26 Nov 2025, 21:20`

Although we receive no response from **Reviewer kvMY** during the discussion period, we believe our rebuttal addresses his concerns.

- **Add evaluations of knowledge editing:** We add experimental results for **five lifelong editing methods** (FT-LLM, FT-VIS, RECIPE, LTE, and LiveEdit), and include a **discussion and citations** regarding knowledge editing in Appendix H (Page 38, Lines 2000-2022).
- **Evolving knowledge is unknown to LMMs:** We ensure this through Time Isolation and Experimental Verification. Notably, **Reviewer fBUi** holds the **same concern before the discussion**, but **our subsequent response resolves the concern and results in a score increase.**


**Rebuttal Outcome**

Three reviewers explicitly confirmed that **their concerns were resolved**, as detailed in the table below.

| Reviewer | Feedback                                                                                           |  Final Rating                                   | Date |
|---------|--------------------------------------------------------------------------------------------------------------------|------------------------------------------------|--------------|
| **PjCG**    | "The clarifications ... mostly **resolve my previous concerns**. I am **keeping my original score**."                                            | **8**                                              | 25 Nov 2025, 22:22       |
| **fBUi**    | "..., this **addresses our concern**. We have **raised our score**."         | **8** | 27 Nov 2025, 10:49       |
| **SWqj**    | "Thank you ... I have **increased my rating** accordingly."                                  | **8**                     | 26 Nov 2025, 21:20       |
| **kvMY**    | None                                  | **Unknown**                     | None                                         |

Thank you again for your consideration.

Best regards,

Authors of Paper 140

---

### Meta-Review · Area_Chair_zBpn · 2026-01-07

**Summary:**

This paper addresses the ability of injection dynamic evolving multi-modal knowledge into LLMs. The paper introduces MMEVOKE, a large-scale benchmark comprising 9,422 multi-modal samples spanning 159 subtypes, collected from evolving entities and news since 2024. Through experimental results, the paper observe that current knowledge injection methods have poor performance. To this end, the authors further propose knowledge-aware augmentation (versus naive data augmentation) and retention strategies (e.g., replay and MoELoRA). The results demonstrate that knowledge-aware augmentation could improve injection performance, and Data Replay/MoE methods effectively alleviate degradation.

**Reviewer Concerns:**

According to reviews and rebuttals, most concerns have been addressed by the authors, except the follows.

1) How do you ensure that knowledge updates scraped from CNN or Wikipedia do not already exist in the LMM’s parametric knowledge, given the counterexamples mentioned?

2) The mentions that existing studies merely as text-only benchmarks and presenting the concept of "evolving knowledge" as a novel contribution in the first paragraph, are appropriate.

**Reviewer Scores:**

According to the discussions in the rebuttals, two reviewers raised their scores from 6 to 8. So the final scores will be 8 8 8 4.

---

### Decision · Program_Chairs · 2026-01-26

Accept (Poster)